# Ribosome profiling of porcine reproductive and respiratory syndrome virus reveals novel features of viral gene expression

Georgia M Cook[1], Katherine Brown[1], Pengcheng Shang[2†], Yanhua Li[2‡], Lior Soday[1§], Adam M Dinan[1#], Charlotte Tumescheit[1], AP Adrian Mockett[3], Ying Fang[2*¶], Andrew E Firth[1*], Ian Brierley[1*]

[1]Department of Pathology, University of Cambridge, Cambridge, United Kingdom; [2]Department of Diagnostic Medicine and Pathobiology, Kansas State University, Manhattan, United States; [3]Cambivac Ltd, Cambridge, United Kingdom

**\*For correspondence:**
yingf@illinois.edu (YF);
aef24@cam.ac.uk (AEF);
ib103@cam.ac.uk (IB)

**Present address:** [†]Department of Pediatrics, School of Medicine, University of Pittsburgh, Pittsburgh, United States; [‡]College of Veterinary Medicine, Yangzhou University, Yangzhou, China; [§]Department of Chemistry, Molecular Sciences Research Hub, Imperial College London, London, United Kingdom; [#]Department of Medicine, University of Cambridge, Cambridge, United Kingdom; [¶]Department of Pathobiology, University of Illinois at Urbana-Champaign, Champaign, United States

**Abstract** The arterivirus porcine reproductive and respiratory syndrome virus (PRRSV) causes significant economic losses to the swine industry worldwide. Here we apply ribosome profiling (RiboSeq) and parallel RNA sequencing (RNASeq) to characterise the transcriptome and translatome of both species of PRRSV and to analyse the host response to infection. We calculated programmed ribosomal frameshift (PRF) efficiency at both sites on the viral genome. This revealed the nsp2 PRF site as the second known example where temporally regulated frameshifting occurs, with increasing −2 PRF efficiency likely facilitated by accumulation of the PRF-stimulatory viral protein, nsp1β. Surprisingly, we find that PRF efficiency at the canonical ORF1ab frameshift site also increases over time, in contradiction of the common assumption that RNA structure-directed frameshift sites operate at a fixed efficiency. This has potential implications for the numerous other viruses with canonical PRF sites. Furthermore, we discovered several highly translated additional viral ORFs, the translation of which may be facilitated by multiple novel viral transcripts. For example, we found a highly expressed 125-codon ORF overlapping nsp12, which is likely translated from novel subgenomic RNA transcripts that overlap the 3′ end of ORF1b. Similar transcripts were discovered for both PRRSV-1 and PRRSV-2, suggesting a potential conserved mechanism for temporally regulating expression of the 3′-proximal region of ORF1b. We also identified a highly translated, short upstream ORF in the 5′ UTR, the presence of which is highly conserved amongst PRRSV-2 isolates. These findings reveal hidden complexity in the gene expression programmes of these important nidoviruses.

## Editor's evaluation

The article presents a first example of a detailed quantitative study of host and PRRSV gene expression over the time course of infection. The study not only identifies multiple non-canonical mechanisms of PRRSV gene expression regulation, but also shows that the frameshifting efficiency at the canonical ORF1ab frameshifting site changes with time. This finding provides new insights into the viral gene expression and into the regulation of programmed ribosome frameshifting, which has important implications for understanding viral biology and for developing antiviral drugs.

**eLife digest** Viruses have tiny genomes. Rather than carry all the genetic information they need, they rely on the cells they infect. This makes the few genes they do have all the more important. Many viruses store their genes not in DNA, but in a related molecule called RNA. When the virus infects cells, it uses the cells' ribosomes – the machines in the cells that make proteins – to build its own proteins. One of the central ideas in biology is that one molecule of RNA carries the instructions for just one type of protein. But many viruses break this rule.

The ribosomes in cells read RNA instructions in blocks of three: three RNA letters correspond to one protein building block. But certain sequences in the RNA of viruses act as hidden signals that affect how ribosomes read these molecules. These signals make the ribosomes skip backward by one or two letters on the viral RNA, restarting part way through a three-letter block. Scientists call this a 'frameshift', and it is a bit like changing the positions of the spaces in a sentence. The virus causes these frameshifts using proteins or by folding its RNA into a knot-like structure. The frameshifts result in the production of different viral proteins over time. The porcine reproductive and respiratory syndrome virus (PRRSV) uses frameshifts to cause devastating disease in pigs. Besides the sequences in its RNA that allow the ribosomes to skip backwards, the viral enzyme that copies the RNA can also skip forward. This results in shortened copies of its genes, which also changes the proteins they produce.

To find out exactly how PRRSV uses these frameshifting techniques, Cook et al. examined infected cells in the laboratory. They monitored the RNA made by the virus and looked closely at the way the cells read it using a technique called ribosome profiling. This revealed that frameshifting increases over the course of an infection. This is partly because the viral protein that causes frameshifts builds up as infection progresses, but it also happened with frameshifts caused by RNA knots. The reason for this is less clear. Cook et al. also discovered several new RNAs made later in infection, which could also change the proteins the virus makes.

RNA viruses cause disease in humans as well as pigs. Examples include coronaviruses and HIV. Many of these also have frameshift sites in their genomes. A better understanding of how frameshifts change during infection may aid drug development. Future work could help researchers to understand which proteins viruses make at which stage of infection. This could lead to new treatments for viruses like PRRSV.

## Introduction

Porcine reproductive and respiratory syndrome virus (PRRSV) is an enveloped, positive-sense, single-stranded RNA virus in the family *Arteriviridae* (order: *Nidovirales*) (*Meulenberg et al., 1994*; *Cavanagh, 1997*), and the aetiological agent of the disease from which it takes its name: porcine reproductive and respiratory syndrome (PRRS). Attempts to control PRRS by vaccination have had limited success (*Nan et al., 2017*), and it remains one of the most economically devastating diseases of swine, causing reproductive failure in adult sows and respiratory failure in young pigs, at an estimated cost of $664 million a year in the US alone (*Holtkamp et al., 2013*; *Kappes and Faaberg, 2015*). The two lineages of PRRSV, formerly known as 'European' (Type 1) and 'North American' (Type 2) PRRSV, share just ~60% pairwise nucleotide similarity and were recently reclassified as two separate species, *Betaarterivirus suid 1* and *2* (viruses named PRRSV-1 and PRRSV-2) (*Kappes and Faaberg, 2015*; *Collins et al., 1992*; *Wensvoort et al., 1991*). For ease of reference, PRRSV-1 is herein referred to as EU (European) and PRRSV-2 as NA (North American) PRRSV, although both lineages are observed worldwide (*Guo et al., 2018a*). Despite the substantial genetic and antigenic diversity between the two species, the overall clinical symptoms are similar, although there is also considerable (~20%) genetic diversity within each species, rendering this isolate-dependent (*Nan et al., 2017*; *Kappes and Faaberg, 2015*). This is largely due to PRRSV's rapid mutation rate, which leads to relatively frequent emergence of highly pathogenic strains capable of escaping existing immunity, particularly within the NA PRRSV species (*Nan et al., 2017*; *Kappes and Faaberg, 2015*).

The PRRSV genome (14.9–15.5 kb; *Figure 1A*) is 5'-capped, 3'-polyadenylated, and directly translated following release into the cytoplasm (*Snijder et al., 2013*). Like most members of the order *Nidovirales*, PRRSV replication includes the production of a nested set of subgenomic (sg) RNAs by

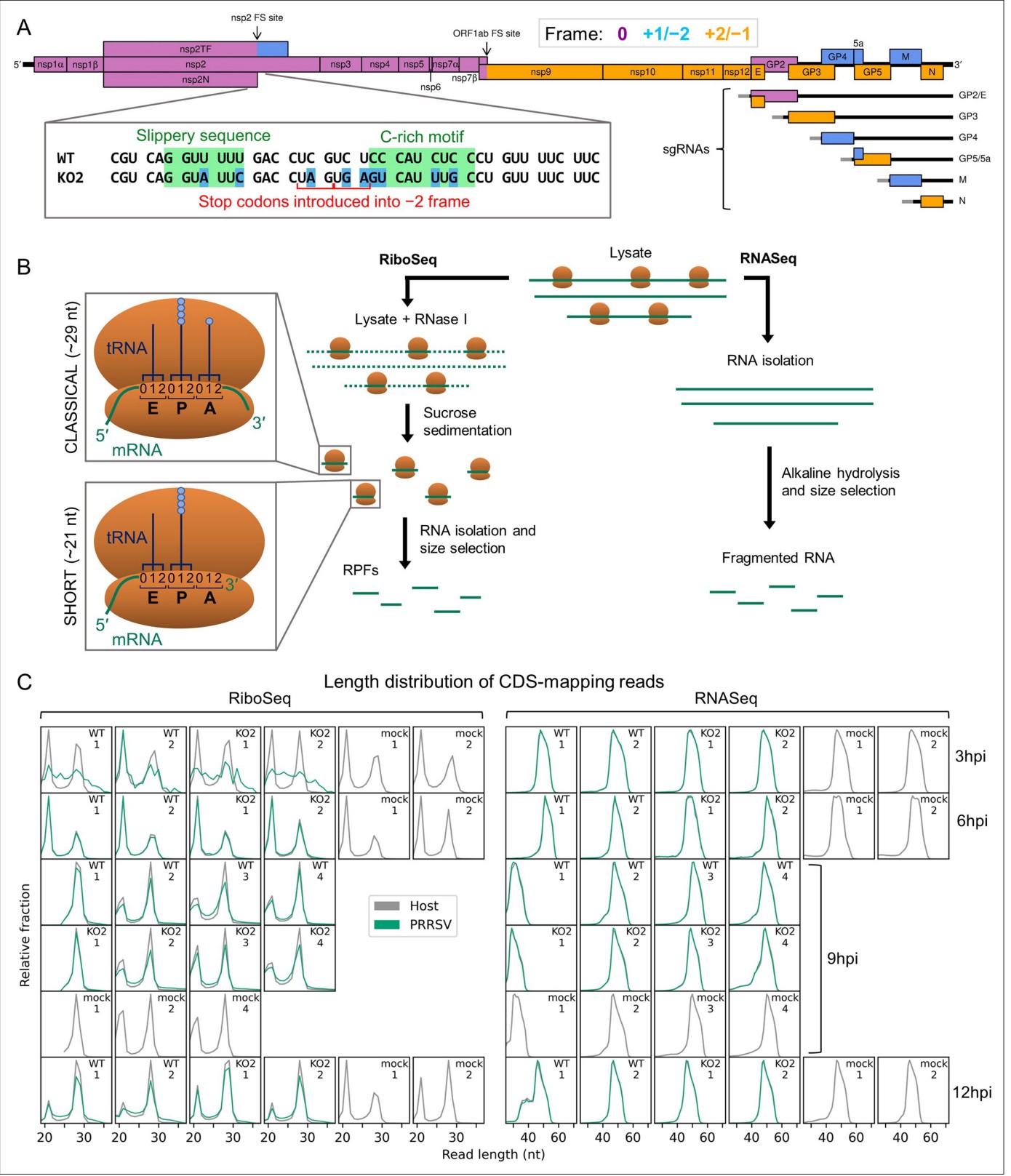

**Figure 1.** An overview of the experimental set-up and the quality of the datasets. (**A**) Genome map of the North American porcine reproductive and respiratory syndrome virus (NA PRRSV) isolate used in this study (SD95-21, GenBank accession KC469618.1). ORFs are coloured and offset on the y-axis according to their frame relative to ORF1a (0: purple, no offset; +1/−2: blue, above axis; +2/−1: yellow, below axis). Subgenomic (sg) RNAs are shown beneath the full-length genomic RNA, with the region of 5′ UTR that is identical to the genomic 5′ UTR shown in grey (known as the 'leader'). 'FS',

*Figure 1 continued on next page*

*Figure 1 continued*

frameshift. ORFs translated from each sgRNA are depicted as coloured boxes and named to the right. The nucleotide sequence at the non-structural protein (nsp)2 programmed ribosomal frameshift (PRF) site of the NA PRRSVs used in this study is shown (boxed), with mutations made to disrupt PRF and/or expression of nsp2TF in the KO2 mutant virus highlighted in blue. All mutations are synonymous with respect to the ORF1a amino acid sequence. (**B**) Key experimental steps in preparation of RiboSeq libraries (left) and parallel RNASeq libraries (right). Schematics of ribosomes protecting classical length ribosome-protected fragments (RPFs) (A site occupied) and short RPFs (A site unoccupied) are shown to the left, with numbers within the decoding centre indicating nucleotide positions within codons. (**C**) Length distribution of positive-sense RiboSeq (left) and RNASeq (right) reads mapping within host (grey) or viral (green, mock excluded) coding sequences (CDSs) in each library. For 9 hr post-infection (hpi) replicate 1 samples (RiboSeq and RNASeq), fragments of 25–34 nt were size-selected during the library preparation; for all other samples, the minimum length selected was 19 nt for RiboSeq and ~45 nt for RNASeq. Note that the RiboSeq library 9 hpi mock replicate 3 was discarded due to poor quality. Further quality control analyses can be found in *Figure 1—figure supplement 1–Figure 1—figure supplement 6*.

The online version of this article includes the following figure supplement(s) for figure 1:

**Figure supplement 1.** Metagene profile showing the average distribution of 5′ ends of host mRNA-mapping reads relative to start and stop codons.

**Figure supplement 2.** Phase composition of CDS-mapping reads.

**Figure supplement 3.** Length distribution of ribosome-protected fragments (RPFs) mapping to specified regions of the viral genome.

**Figure supplement 4.** Number of reads of each length that are attributed to each phase for ribosome-protected fragments (RPFs) mapping to specified regions of the viral genome.

**Figure supplement 5.** Assessment of potential ribonucleoprotein (RNP) contamination of host-mapping RiboSeq reads.

**Figure supplement 6.** Quality control for the European porcine reproductive and respiratory syndrome virus (EU PRRSV) dataset.

discontinuous transcription, where the viral RNA-dependent RNA polymerase (RdRp) jumps between similar sequences in the 3′-proximal region of the genome and the 5′ UTR, known as body and leader transcription regulatory sequences (TRSs), respectively (*Kappes and Faaberg, 2015*; *Posthuma et al., 2017*). These sgRNAs are 5′- and 3′-co-terminal and are translated to express the structural proteins encoded towards the 3′ end of the genome (*Kappes and Faaberg, 2015*; *Posthuma et al., 2017*). The 5′-proximal two-thirds of the genome contains two long ORFs, ORF1a and ORF1b, with a −1 programmed ribosomal frameshift (PRF) site present at the overlap of the two ORFs (*Meulenberg et al., 1993*; *Nelsen et al., 1999*). Ribosomes that frameshift at this site synthesise polyprotein (pp)1ab, while the remainder synthesise pp1a, both of which are cleaved by viral proteases into several non-structural proteins (nsps) (*Kappes and Faaberg, 2015*; *Snijder et al., 1994*). The proteins encoded by ORF1b include the RdRp and the helicase, and frameshifting at this site is thought to set the stoichiometry of these proteins relative to those encoded by ORF1a, a prevalent expression strategy in the *Nidovirales* order (*Gorbalenya et al., 2006*).

Canonical −1 PRF signals are characterised by two main features, a heptanucleotide 'slippery' sequence (SS) which permits re-pairing of the codon:anticodon duplex in the new reading frame, separated by a 5–9 nucleotide (nt) spacer from a downstream RNA structure, often a pseudoknot. This is thought to present a 'roadblock' which impedes ribosome processivity over the slippery sequence and stimulates frameshifting (*Rodnina et al., 2020*; *Atkins et al., 2016*; *Firth and Brierley, 2012*; *Plant and Dinman, 2005*; *Namy et al., 2006*; *Caliskan et al., 2014*). In the PRRSV genome, the ORF1ab frameshift signal comprises a U_UUA_AAC slippery sequence (where underscores delineate codons in the 0 frame) and a pseudoknot beginning 5 nt downstream (*Meulenberg et al., 1993*; *Nelsen et al., 1999*). The efficiency of −1 PRF at the PRRSV ORF1ab site has not been measured in the context of infection, but is thought to be around 15–20% based on assays using reporter constructs (*den Boon et al., 1991*; *Bekaert and Rousset, 2005*).

Recently, the region of the PRRSV genome encoding nsp2 was found to contain a second PRF signal (*Figure 1A*, inset, WT), conserved in all known arteriviruses except equine arteritis virus (EAV) and wobbly possum disease virus (WPDV) (*Fang et al., 2012*; *Li et al., 2014*; *Napthine et al., 2016*; *Li et al., 2019*). This PRF signal is unusual in that it stimulates both −1 and −2 PRF, enabling production of three variants of nsp2 and rendering it the first example of efficient −2 PRF in a eukaryotic system (*Fang et al., 2012*; *Li et al., 2014*). These three proteins share the N-terminal two-thirds of nsp2 (the 0-frame product), which encodes a papain-like protease (PLP) 2 domain – an ovarian tumour domain (OTU) superfamily protease with deubiquitinase (DUB) and deISGylase activity (*Han et al., 2009*; *van Kasteren et al., 2012*; *Frias-Staheli et al., 2007*; *Sun et al., 2010*; *Sun et al., 2012*; *Li et al., 2018*). This has an immune antagonistic effect, and interferon (IFN)-β signalling inhibition has been

demonstrated for all three variants of nsp2, most strongly for the frameshift products (*Li et al., 2018*). After the PRF site, nsp2 contains a multi-spanning transmembrane (TM) domain, thought to promote formation of double-membrane vesicles (DMVs) in the peri-nuclear region and anchor nsp2 to these membranes (*Kappes et al., 2015*; *Snijder et al., 2001*; *Knoops et al., 2012*). Ribosomes which undergo −2 PRF at this site translate 169 codons in the −2 frame to produce nsp2TF. This contains an alternative putative multi-spanning TM domain, thought to be responsible for targeting nsp2TF to the exocytic pathway, where it deubiquitinates the PRRSV structural proteins GP5 and M, preventing their degradation (*Fang et al., 2012*; *Guo et al., 2021*). Nsp2N, the product of −1 PRF, is a truncated form of nsp2, which is generated following termination of translation at a −1-frame stop codon immediately downstream of the slippery sequence, and is predicted to be cytosolic (*Fang et al., 2012*; *Li et al., 2014*).

A second unique feature of the nsp2 PRF site is its non-canonical nature. Rather than an RNA secondary structure, the stimulatory element is a complex of a cellular protein, poly(rC) binding protein (PCBP), and the viral protein nsp1β, bound at a C-rich motif (CCCANCUCC) located 10 nt downstream of the slippery sequence (G_GUU_UUU) (*Fang et al., 2012*; *Li et al., 2014*; *Napthine et al., 2016*; *Li et al., 2019*). How binding of this motif by the protein complex stimulates PRF is uncertain, but it may act as a roadblock analogous to the RNA structures of canonical PRF (*Li et al., 2014*; *Napthine et al., 2016*; *Li et al., 2019*; *Patel et al., 2020*). In contrast to RNA structure-directed PRF sites, which are commonly assumed to operate at a fixed efficiency, the *trans*-acting mechanism of PRF stimulation at the nsp2 site presents a potential mechanism for temporal regulation, as observed for cardioviruses – the only other known example of protein-stimulated PRF (*Loughran et al., 2011*; *Finch et al., 2015*; *Napthine et al., 2017*; *Napthine et al., 2019*; *Hill et al., 2021b*; *Hill et al., 2021a*). Frameshift efficiency in EU PRRSV-infected MARC-145 cells at 24 hpi was calculated as 20% for −2 PRF and 7% for −1 PRF (*Fang et al., 2012*); however, this has not been measured over a timecourse of infection.

In recent years, both low- and high-throughput studies of nidoviruses have highlighted considerably greater complexity in both the transcriptome and translatome than is captured solely by the canonical transcripts and ORFs (*Nelsen et al., 1999*; *Yuan et al., 2000*; *Di et al., 2017*; *Kim et al., 2020*; *Wang et al., 2021*; *Stewart et al., 2018*; *Irigoyen et al., 2016*; *Dinan et al., 2019*; *Finkel et al., 2021b*; *Zhang et al., 2021b*). Here, we use ribosome profiling (RiboSeq), a deep-sequencing-based technique which generates a global snapshot of ongoing translation (*Ingolia et al., 2009*), in parallel with RNASeq, to probe viral and host gene expression over a timecourse of PRRSV infection. Host differential gene expression analysis revealed that many of the transcriptional changes upon infection were counteracted by reductions in translation efficiency (TE), indicating a dampened host response, and highlighting the importance of looking beyond transcription when analysing gene expression. On the viral genome, our studies reveal, for the first time, a significant increase in frameshift efficiency over the course of infection at the nsp2 −2 PRF site, highlighting arteriviruses as the second example of temporally regulated frameshifting during infection. In addition, we identify several novel viral ORFs, including a highly expressed upstream ORF (uORF), the presence of which is conserved amongst NA PRRSV isolates. In both species of PRRSV, related non-canonical sgRNAs overlapping ORF1b were identified and characterised. These likely facilitate the expression of several of the novel ORFs which overlap ORF1b, and the observation of increased ribosome density in the 3′-proximal region of ORF1b suggests they may also function to temporally regulate expression of the C-terminal region of ORF1b itself. This first application of RiboSeq to an arterivirus uncovers hidden layers of complexity in PRRSV gene expression that have implications for other important viruses.

## Results

### Experimental set-up

PRRSV gene expression was investigated using three viruses: an EU PRRSV isolate based on the Porcilis vaccine strain (MSD Animal Health; GenBank accession OK635576.1), NA PRRSV SD95-21 (GenBank accession KC469618.1), and a previously characterised mutant variant (NA PRRSV SD95-21 KO2) which bears silent mutations in the nsp2 PRF site slippery sequence and C-rich motif rendering it unable to bind PCBP, induce −1 or −2 PRF, or produce nsp2N or nsp2TF (*Figure 1A*, inset) (*Fang et al., 2012*; *Li et al., 2014*; *Napthine et al., 2016*; *Li et al., 2018*). MA-104 cells (*Chlorocebus sabaeus*) were infected with EU PRRSV at a multiplicity of infection (MOI) of ~1–3 and harvested

at 8 hr post-infection (hpi) following a 2 min pre-treatment with the translation elongation inhibitor, cycloheximide (CHX). MARC-145 cells (a cell line derived from MA-104) were infected with NA PRRSV (WT or KO2 mutant) at MOI 5 or mock-infected and harvested at 3, 6, 9, or 12 hpi by flash-freezing without CHX pre-treatment. Cell lysates were used for ribosome profiling, in which RNase I was added to digest unprotected regions of RNA and ribosomes were purified to isolate ribosome-protected fragments (RPFs) of RNA (*Figure 1B*), which indicate the positions of ribosomes at the time of harvesting. In parallel, aliquots of the same lysates were subjected to alkaline hydrolysis to generate fragments of RNA for RNASeq. Amplicons were prepared, deep sequenced, and reads aligned to host (*C. sabaeus*) and viral genomes (*Supplementary file 1*) to characterise the transcriptome and translatome of infected cells.

## Data quality analysis

Quality control analyses were performed as described previously (*Irigoyen et al., 2016*; *Figure 1C*, *Figure 1—figure supplements 1–6*), revealing that the overall data quality is good. The length distribution of coding sequence (CDS)-mapping RPFs is observed to peak at ~21 nt (where fragments of this length were purified) and at ~29 nt, with RPFs of these lengths thought to originate from, respectively, ribosomes with an empty A site or an A site occupied by aminoacyl-tRNA (*Figure 1B and C*, *Figure 1—figure supplement 6A*; *Ingolia et al., 2009*; *Steitz, 1969*; *Wolin and Walter, 1988*; *Wu et al., 2019*; *Lareau et al., 2014*). Interestingly, the proportion of 'short' (19–24 nt) RPFs is significantly lower in the NA PRRSV-infected libraries than mock libraries at late timepoints (9 and 12 hpi grouped; p=0.03; see Materials and methods). In yeast, this phenotype has been attributed to stress-induced phosphorylation of eukaryotic elongation factor (eEF) 2, leading to inhibition of translation elongation (*Wu et al., 2019*), which suggests a similar regulatory response may be triggered here by the stress of PRRSV infection. The predominant distance between the 5′ end of an RPF and the P site of the ribosome is 12 nt in these datasets (*Figure 1—figure supplements 1 and 6B*), resulting in CDS-mapping RiboSeq reads showing clear triplet periodicity, known as 'phasing', with the majority of RPF 5′ ends mapping to the first position within the codon, known as phase 0 (*Figure 1—figure supplement 1*, *Figure 1—figure supplement 2*, *Figure 1—figure supplement 6B and C*). Together with the observed characteristic length distribution (*Figure 1C*, *Figure 1—figure supplement 6A*), this indicates that a high proportion of these reads are genuine RPFs. In contrast, the length and 5′ end position of RNASeq reads is determined by alkaline hydrolysis and size selection, leading to a broader length distribution (*Figure 1C*, *Figure 1—figure supplement 6A*) and lack of a clearly dominant phase (*Figure 1—figure supplement 1*, *Figure 1—figure supplement 2*, *Figure 1—figure supplement 6B and C*). Virus CDS-mapping reads show a similar profile to host CDS-mapping reads (*Figure 1C*, *Figure 1—figure supplement 2*, *Figure 1—figure supplement 6A and C*), with the exception of 3 hpi NA PRRSV RiboSeq libraries, in which the background level of non-RPF contamination in the virus-mapping fraction appears to be high relative to the proportion of genuine RPFs, likely due to the low levels of viral translation at this timepoint. These libraries are therefore excluded from all analyses except those in *Figure 2D–F* and *Figure 2—figure supplement 3A*, where they provide an upper bound. The subtle flattening of the length distribution and phase composition of virus-mapping reads compared to host-mapping reads in some NA PRRSV RiboSeq libraries at late timepoints (*Figure 1C*, *Figure 1—figure supplement 2*) suggests that a small proportion of viral reads originate from non-RPF sources, such as protection from RNase I digestion by viral ribonucleoprotein (RNP) complex formation. This non-RPF fraction of the library (henceforth referred to as RNP contamination although it could originate from several sources) is predominantly noticeable among reads mapping to the ORF1b region of the viral genome (*Figure 1—figure supplements 3 and 4*), where the read depth from genuine translation is lowest. RiboSeq read lengths for which a high proportion of reads map to phase 0 were inferred to be least likely to have a high proportion of RNP contamination (*Figure 1—figure supplement 4*) and were selected for all NA PRRSV RiboSeq analyses henceforth, unless specified. RNP contamination is not a relevant concern for RNASeq libraries (as proteins are enzymatically digested before RNA purification), and it does not noticeably affect the EU PRRSV RiboSeq libraries, nor RPFs mapping to the host transcriptome (*Figure 1—figure supplements 5 and 6*). Overall, we inferred that these datasets have a high proportion of RiboSeq reads representing genuine RPFs, and where RNP contamination is evident in lowly translated regions of the viral genome its effects will likely be ameliorated by stratification of read lengths.

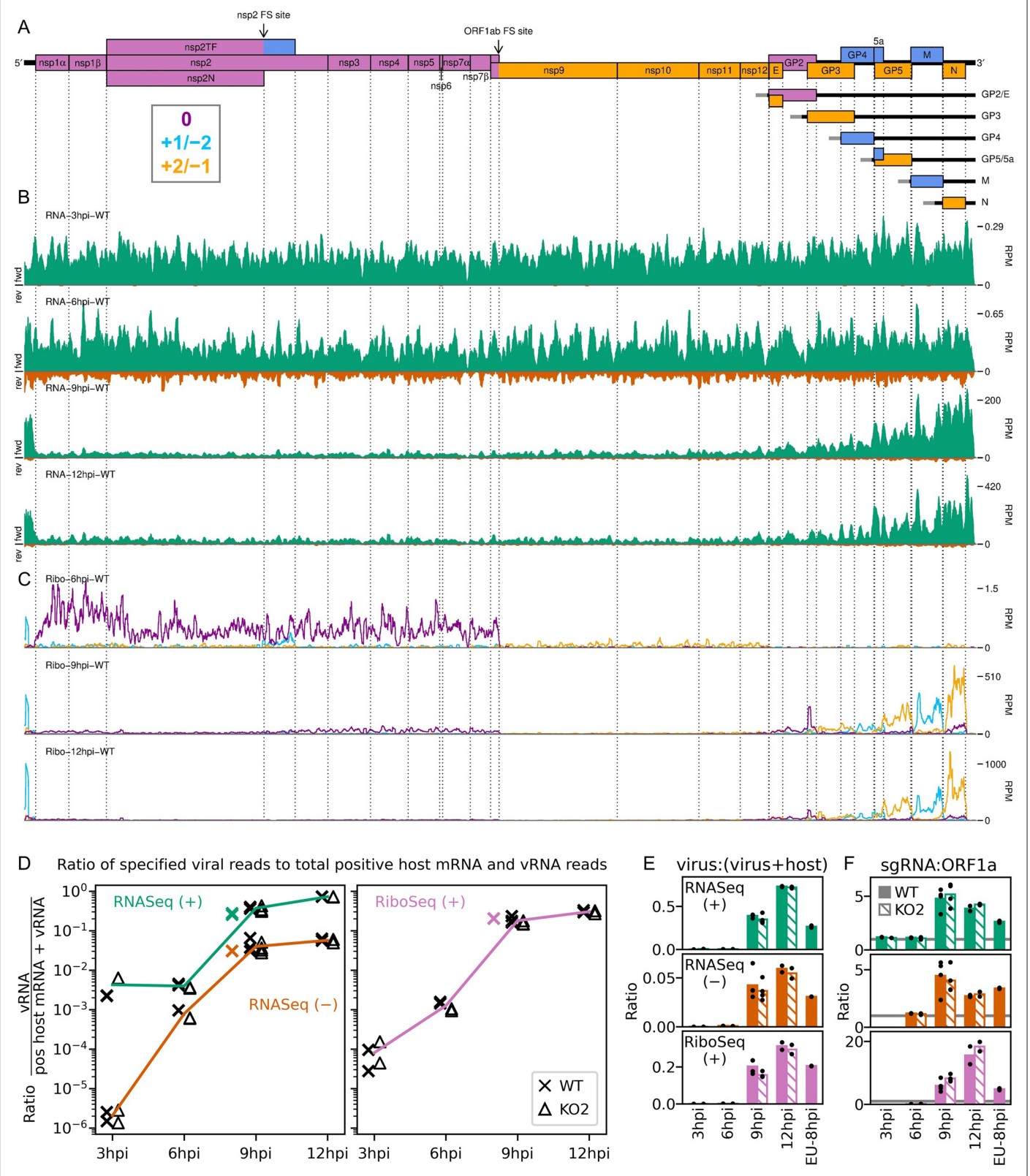

**Figure 2.** An overview of viral transcription and translation over a timecourse of infection. (**A**) Genome map of North American porcine reproductive and respiratory syndrome virus (NA PRRSV), reproduced from *Figure 1A*. (**B**) RNASeq read densities in reads per million mapped reads (RPM) on the WT viral genome, after application of a 45-nt running mean filter, from cells harvested over a timecourse of 3–12 hr post-infection (hpi). Positive-sense reads are plotted in green (above the horizontal axis), negative-sense in orange (below the horizontal axis). The WT libraries with the best RiboSeq

*Figure 2 continued on next page*

*Figure 2 continued*

quality control results were selected for this plot (3 hpi replicate 1, 6 hpi replicate 2, 9 hpi replicate 4, 12 hpi replicate 1), with further replicates and KO2 libraries shown in *Figure 2—figure supplement 1*. (C) RiboSeq read densities on the WT viral genome from the counterpart libraries to (B). Reads were separated according to phase (0: purple; –2/+1: blue; –1/+2: yellow), and densities plotted after application of a 15-codon running mean filter. Further replicates and KO2 libraries are shown in *Figure 2—figure supplement 2*. (D) Ratio of virus-mapping reads to (positive-sense host mRNA- plus positive-sense vRNA-mapping reads). Virus-mapping reads (all read lengths) in the numerator were split into the following categories: positive-sense RNASeq (green), negative-sense RNASeq (orange), and positive-sense RiboSeq (purple). Analysis of negative-sense RiboSeq reads can be found in *Figure 2—figure supplement 3*. The line graphs represent the mean ratios for each category for NA PRRSV (WT and KO2 combined), with individual datapoints for WT (crosses) and KO2 (triangles) overlaid, offset to the left and right, respectively. EU PRRSV (8 hpi) ratios are plotted as individual datapoints represented by crosses in the category colour. The RiboSeq (+) 3 hpi timepoint is plotted here to represent the upper limit of the NA PRRSV fraction at this timepoint, likely inflated by the relatively high proportion of non-ribosome-protected fragment (non-RPF) reads in these libraries. (E) Data from (D) represented on a linear scale. Here, data from WT (solid bars) and KO2 (hatched bars) are plotted separately, and individual datapoints are plotted as black circles. (F) Ratio of the density of subgenomic (sg)RNA-mapping reads to ORF1a-mapping reads. All read lengths were used, and densities were calculated as reads per kilobase per million mapped reads (RPKM) of reads from each category in (E). RiboSeq 3 hpi libraries were excluded, and negative-sense RNASeq was omitted from the plot at 3 hpi due to the number of reads being insufficient for robust assessment of the ratio. Categories arranged and plot constructed as in (E), with a grey line indicating a ratio of 1.

The online version of this article includes the following figure supplement(s) for figure 2:

**Figure supplement 1.** Further replicates of a timecourse of North American porcine reproductive and respiratory syndrome virus (NA PRRSV) viral transcription.

**Figure supplement 2.** Further replicates of a timecourse of North American porcine reproductive and respiratory syndrome virus (NA PRRSV) viral translation.

**Figure supplement 3.** Investigation of negative-sense RiboSeq reads.

## Viral transcription and translation over a timecourse of infection

Having confirmed data quality, we moved on to analyse virus replication over the timecourse by plotting RNASeq and RiboSeq read densities at each position on the viral genome (*Figures 2 and 3*, *Figure 2—figure supplements 1 and 2*). RNASeq plots revealed a predictable pattern of PRRSV replication and transcription, with low read levels at 3 hpi, likely corresponding to input genomes, evidence for genome replication at 6 hpi, with the appearance of negative-sense reads, and high-level synthesis of sg mRNAs at later timepoints (*Figure 2*, *Figure 2—figure supplement 1*). The observed profile of general virus translation was also consistent with expectation (*Figure 2*, *Figure 2—figure supplement 2*). At 3 hpi (plot not shown), a small number of genuine RPFs were observed (see previous paragraph), indicating that translation of the NA PRRSV genome is just beginning to reach the level detectable by RiboSeq under these conditions. At 6 hpi, translation of ORF1ab is robustly detectable and comprises the majority of viral translation (*Figure 2*, *Figure 2—figure supplement 2*; mean sgRNA:ORF1a RPF density ratio 0.08), consistent with the lack of significant sgRNA production at this timepoint. At 9 hpi, translation of sgRNAs dominates the landscape, and viral translation represents a sizeable proportion of ongoing translation in the cell (*Figure 2*, *Figure 2—figure supplement 2*). Consistent with this, viral nsp1β expression at 9 hpi is clearly detectable by Western blotting (*Figure 3D and E*) and other studies have shown robust expression of viral replicase proteins and viral RNA (vRNA) replication at this timepoint (*Li et al., 2012*; *Kreutz and Ackermann, 1996*). Positive-sense vRNA continues to accumulate between 9 and 12 hpi, although accumulation of the negative-sense counterpart appears to reach a plateau and, at both timepoints, production and translation of sgRNAs are highly favoured over gRNA (*Figure 2*, *Figure 2—figure supplements 1 and 2*). This likely represents a transition towards virion formation, for which the main components required are positive-sense gRNA and structural proteins, expressed from sgRNAs. At all timepoints, a large RiboSeq peak in the NA PRRSV 5′ UTR is seen (*Figure 2C*, *Figure 2—figure supplement 2*), which results from translation of a novel upstream ORF, discussed below. With the exception of this highly expressed uORF, the transcriptional and translational profile of EU PRRSV at 8 hpi is similar to that of NA PRRSV at 9 hpi, although the production and translation of sgRNAs relative to ORF1a are slightly lower (*Figures 2 and 3*). In all RiboSeq libraries, we noted a variable proportion of negative-sense reads that mapped to the viral genome; however, they do not display the characteristic length distribution or phasing of genuine RPFs (*Figure 2—figure supplement 3*), suggesting that they originate from other sources (discussed above). They are therefore excluded from plots and analyses hereafter.

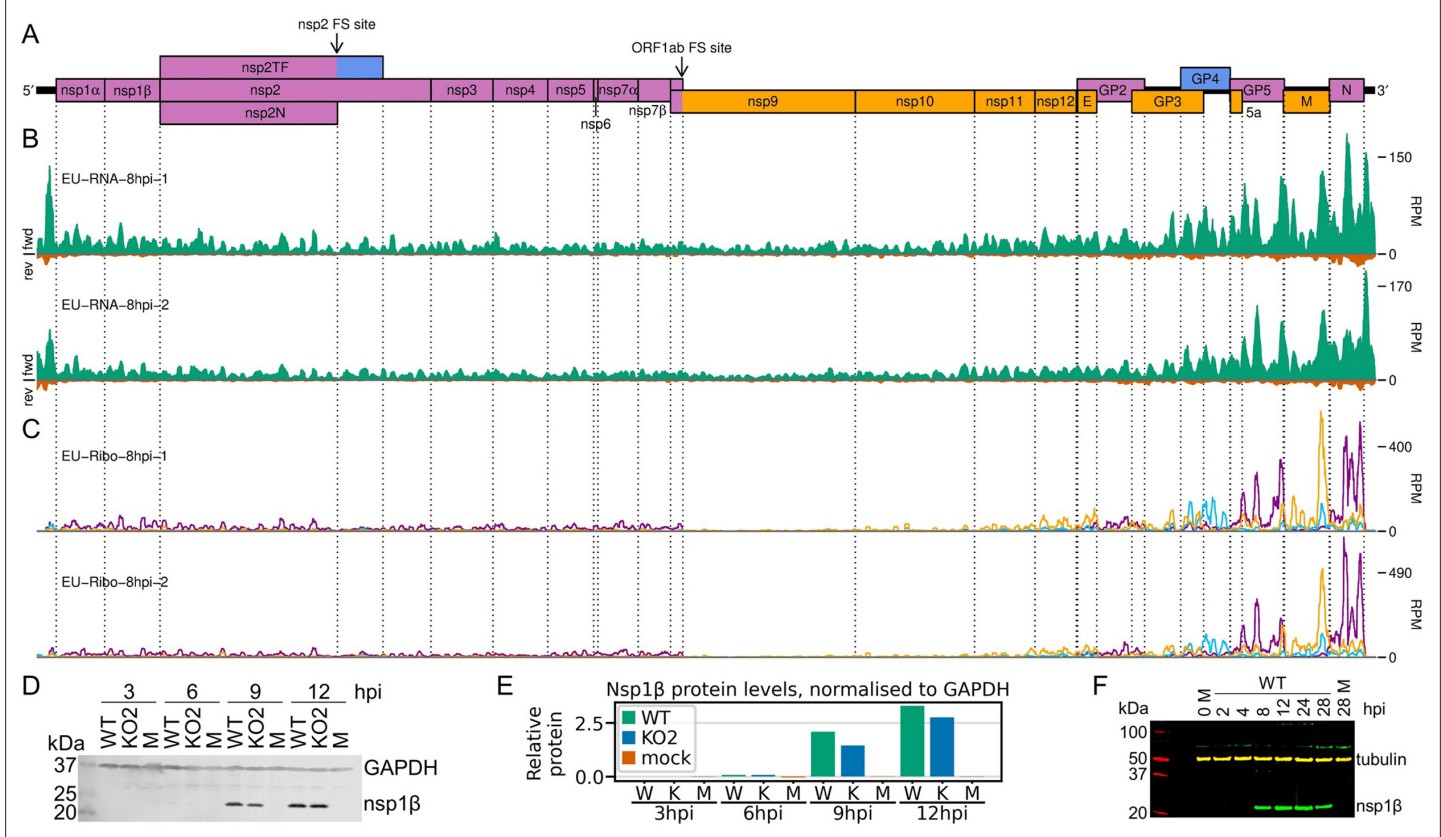

**Figure 3.** Transcription and translation of the European porcine reproductive and respiratory syndrome virus (EU PRRSV) genome and Western blots of non-structural protein 1β (nsp1β). (**A**) Genome map of the EU PRRSV strain used in this study (GenBank accession OK635576.1). Genome map constructed as in *Figure 1A*, with subgenomic RNAs omitted for space considerations. (**B**) RNASeq read densities on the EU PRRSV genome. Plot constructed as in *Figure 2B*. (**C**) RiboSeq read densities on the EU PRRSV genome. Plot constructed as in *Figure 2C*, except for the selection of read lengths to include – in this case, read lengths showing good phasing were selected for inclusion (indicated in *Figure 1—figure supplement 6D*). (**D**) Western blot of lysates used for North American (NA) PRRSV ribosome profiling (replicate 1 samples) with antibodies to viral protein nsp1β (23 kDa) and cellular protein GAPDH (36 kDa) as a loading control. M, mock. (**E**) Quantification of the Western blot from panel (**D**) to determine the level of nsp1β relative to GAPDH. W, WT; K, KO2. (**F**) Western blot of nsp1β expression in MA-104 cells infected with EU PRRSV, harvested over a 28 hr timecourse.

The online version of this article includes the following source data for figure 3:

**Source data 1.** Raw tiff of Western blot in *Figure 3D*.

**Source data 2.** Image of full Western blot in *Figure 3D*, with bands labelled.

**Source data 3.** Quantification of band densities in Western blot in *Figure 3D* and normalised results presented in *Figure 3E*.

**Source data 4.** Raw tiff of Western blot in *Figure 3F*.

**Source data 5.** Image of full Western blot in *Figure 3F*, with bands labelled.

## Characterisation of the PRRSV transcriptome

As described above, discontinuous transcription by the viral RdRp is an integral part of the nidoviral life cycle. Recent RNASeq studies have revealed considerable complexity in nidoviral transcriptomes beyond the canonical transcripts, including the discovery of numerous novel sgRNAs (*Kim et al., 2020*; *Wang et al., 2021*; *Stewart et al., 2018*; *Irigoyen et al., 2016*; *Finkel et al., 2021b*; *Zhang et al., 2021b*; *Viehweger et al., 2019*). We characterise the PRRSV transcriptome here by examining reads which map discontinuously to the viral genome, representing a 'junction' between two regions (*Figure 4A*). Borrowing terminology from the process of splicing, we refer to the 5′-most and 3′-most positions of the omitted region (with respect to the positive-sense genome) as the 'donor' and 'acceptor' sites, respectively (*Figure 4A*). Junctions were located by aligning RNASeq reads to the viral genome using STAR (*Dobin et al., 2013*), and the results were processed (as detailed in Materials and methods) to generate one set of reproducible junctions per timepoint (*Figure 4—source data*

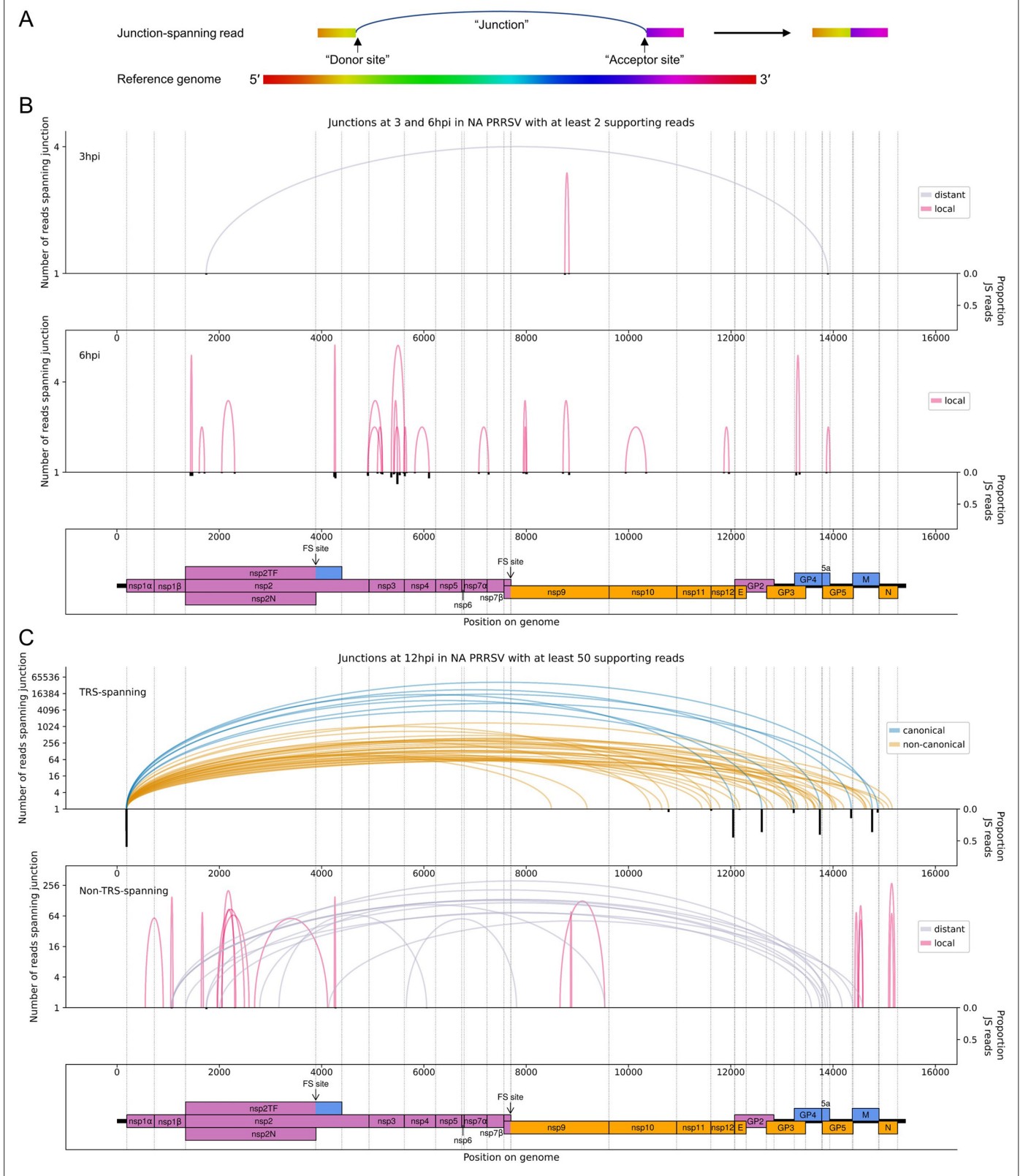

**Figure 4.** The North American porcine reproductive and respiratory syndrome virus (NA PRRSV) transcriptome at 3, 6, and 12 hr post-infection (hpi). (**A**) Illustrative schematic of a junction-spanning (JS) read (not to scale). The blue arc ('Junction') represents a deletion in the read with respect to the reference genome. (**B**) Sashimi plot of junctions in the NA PRRSV dataset at early timepoints during infection (3 and 6 hpi). The number of reads supporting each junction is indicated (on a logarithmic scale) by the highest point of its arc and represents the total number of reads spanning the

*Figure 4 continued on next page*

*Figure 4 continued*

junction in all libraries from the specified timepoint combined. Only junctions for which this number is ≥2 are plotted (which, for these timepoints, does not include any transcription regulatory sequence (TRS)-spanning junctions). Beneath the sashimi plot is an inverted bar chart (black) of the proportion of reads at each donor and acceptor site that span the junction of interest, plotted on a linear scale (see Materials and methods for details). Lists of all junctions from these timepoints, and their associated proportions of JS reads, are given in *Figure 4—source data 1–3*. Internal deletions, in which the donor site does not overlap the leader TRS, are coloured according to whether the deletion is distant (>2000 nt deleted, grey) or local (≤2000 nt deleted, red). (C) Upper: sashimi plot of TRS-spanning junctions at 12 hpi, with the major junction for each canonical subgenomic (sg)RNA shown in blue (including both N-long and N-short for the N sgRNA), and other junctions ('non-canonical') shown in orange. Both upper and lower panels were constructed as in panel (A) except that the threshold for inclusion of junctions was adjusted to ≥50 supporting reads. Lower: sashimi plot of junctions representing internal deletions.

The online version of this article includes the following source data for figure 4:

**Source data 1.** Junctions in the North American porcine reproductive and respiratory syndrome virus (NA PRRSV) transcriptome at 3 hr post-infection (hpi).

**Source data 2.** Junctions in the North American porcine reproductive and respiratory syndrome virus (NA PRRSV) transcriptome at 6 hr post-infection (hpi).

**Source data 3.** Junctions in the North American porcine reproductive and respiratory syndrome virus (NA PRRSV) transcriptome at 12 hr post-infection (hpi).

*1–3*, *Figure 5—source data 1–2*). Junctions for which the donor site overlaps the leader TRS ('TRS-spanning') are expected to give rise to sgRNAs, while the remaining junctions ('non-TRS-spanning') are herein termed 'deletions' (unless specified).

Consistent with the trends identified in the general transcriptome analysis (*Figure 2*), junction-spanning reads attributed to sgRNAs do not pass the filters for detection at early timepoints (*Figure 4B*), but are abundant at 9 and 12 hpi, where they make up the vast majority of viral junction-spanning reads (*Figures 4C and 5A and B*). Canonical sgRNAs are the most abundant transcripts, although reasonably abundant transcript variants are present, which differ only in the length of 5′ UTR between the acceptor site and the CDS start, and are expected to produce the same protein (*Figures 4C and 5*). A study on another arterivirus, simian haemorrhagic fever virus (SHFV), suggests such transcripts may contribute to refining the overall stoichiometry of structural proteins (*Di et al., 2017*). For the N transcript, NA PRRSV isolates VR-2332 and tw91 have each been shown to have a (different) abundant secondary transcript variant (*Nelsen et al., 1999*; *Lin et al., 2002*). Both of these are observed in our NA PRRSV dataset (*Figures 4C and 5A*, *Figure 4—source data 3*, *Figure 5—source data 1*), although the VR-2332-like transcript was much more abundant than the tw91-like transcript, consistent with the fact that SD95-21 is more closely related to VR-2332. This more abundant secondary transcript, herein termed N-short, has a 5′ UTR 114 nt shorter than that of the NA PRRSV primary transcript (herein termed N-long), presenting a potential opportunity for differential translation regulation. If such regulation exists, it is unlikely to be temporal as the ratio of N-long to N-short remains constant, at approximately 6:1, between 9 and 12 hpi (*Figure 4—source data 3*, *Figure 5—source data 1*). Any such regulation would also likely be isolate-dependent as the N-short body TRS is not completely conserved amongst NA PRRSV isolates (*Figure 5C*). Further, it would likely be restricted to NA PRRSV as the N-long body TRS is neither highly conserved nor highly utilised in EU PRRSV (*Figure 5C*), for which N-short is ~60-fold more abundant than any other N transcript (*Figure 5B*, *Figure 5—source data 2*) and its body TRS is absolutely conserved (*Figure 5C*).

In addition to the numerous novel sgRNAs predicted to express full-length structural proteins, we found that most canonical sgRNAs have transcript variants with body TRSs downstream of the start codon, which are expected to express truncated forms of the structural proteins (*Figures 4C and 5A and B*). One of these was also observed for VR-2332 PRRSV: the '5-1' transcript variant (*Nelsen et al., 1999*), which is thought to express a truncated form of GP5, and is present in our NA PRRSV dataset at ~1.7% of the abundance of the primary GP5 transcript (based on the number of junction-spanning reads at the donor site). Similar GP5 transcript variants were observed in SHFV, and mutagenesis studies suggest that the truncated GP5 may be beneficial for viral fitness (*Di et al., 2017*), raising the possibility that the putative truncated forms of this and other PRRSV structural proteins could be functional.

In addition to the transcript variants for the structural proteins, a small number of non-canonical sgRNAs were discovered in both NA and EU PRRSV which have acceptor sites within ORF1b, herein

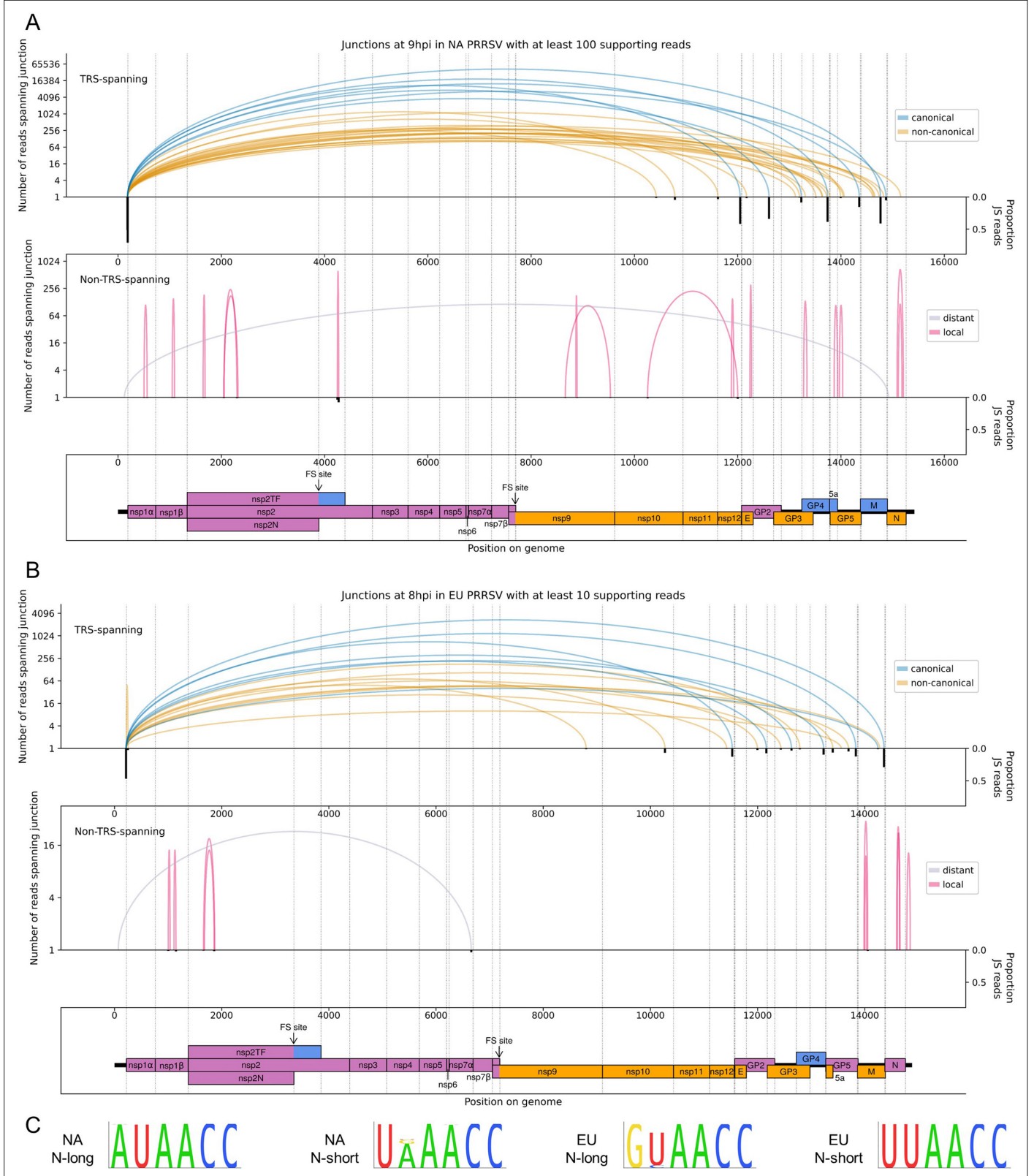

**Figure 5.** The North American (NA) and European (EU) porcine reproductive and respiratory syndrome virus (PRRSV) transcriptomes at 9 and 8 hr post-infection (hpi), respectively. (**A**) Sashimi plots of junctions for NA PRRSV at 9 hpi. Plots constructed as in *Figure 4C*, but with the threshold for inclusion of junctions adjusted to ≥100 junction-spanning (JS) reads in total from all 9 hpi libraries (as eight libraries were analysed at this timepoint compared to four at other timepoints). (**B**) Sashimi plots of junctions for EU PRRSV at 8 hpi. Plots constructed as in *Figure 4C*, but with the threshold for inclusion

*Figure 5 continued on next page*

*Figure 5 continued*

of junctions adjusted to ≥10 supporting reads (as only two libraries were analysed and shorter read lengths are expected to lead to fewer identifiably JS reads). Lists of all junctions from these timepoints, and their associated proportion of JS reads, are given in **Figure 5—source data 1 and 2**. (**C**) Conservation of the body transcription regulatory sequences (TRSs) for N-long and N-short in both species of PRRSV, based on all available full-genome sequences.

The online version of this article includes the following source data for figure 5:

**Source data 1.** Junctions in the North American porcine reproductive and respiratory syndrome virus (NA PRRSV) transcriptome at 9 hr post-infection (hpi).

**Source data 2.** Junctions in the European porcine reproductive and respiratory syndrome virus (EU PRRSV) transcriptome at 8 hr post-infection (hpi).

termed ORF1b sgRNAs (**Figures 4C and 5A and B**). Despite their low abundance relative to the canonical sgRNAs, further analysis suggests that these make a significant contribution to viral translation, discussed later.

Deletions (in which the junction donor site is not the leader TRS) tend to have fewer junction-spanning reads than sgRNAs, but nonetheless may influence gene expression (**Figures 4 and 5**, non-TRS-spanning panels). Many of these likely represent defective interfering (DI) RNAs; however, several of the long-range deletions in the NA PRRSV 12 hpi dataset (**Figure 4C**, grey arcs) bear similarity to 'heteroclite' sgRNAs, a family of non-canonical transcripts found in several NA PRRSV isolates (**Yuan et al., 2000**; **Zhang et al., 2021b**; **Yuan et al., 2004**). Heteroclite sgRNA formation is thought to be directed by short (2–12 nt) regions of similarity between the donor site, located within ORF1a, and the acceptor site, usually located within the ORFs encoding structural proteins (**Yuan et al., 2000**; **Yuan et al., 2004**; **Xiao et al., 2011**). These transcripts can be packaged into virions but, unlike classical DI RNAs, they do not appear to interfere with canonical gRNA or sgRNA production, and are present in a wide range of conditions, including low MOI passage and samples directly isolated from the field (**Yuan et al., 2000**; **Yuan et al., 2004**). In our datasets, the most abundant deletion at 12 hpi (**Figure 4C**, non-TRS-spanning) is identical to the junction that forms the 'S-2' heteroclite sgRNA for VR-2332 PRRSV, from which a fusion of the first 520 amino acids of ORF1a (nsp1α, nsp1β and part of nsp2) and the last 11 amino acids of 5a is thought to be expressed (**Yuan et al., 2000**; **Yuan et al., 2004**). Although this junction is not present above the limit of detection at 6 hpi, it is observed at 9 hpi (**Figure 5—source data 1**; total read counts below the threshold for inclusion in **Figure 5A**) and at 3 hpi (**Figure 4B**, upper), consistent with this transcript being packaged into virions (**Yuan et al., 2000**; **Yuan et al., 2004**). No transcripts resembling heteroclite sgRNAs were detected for EU PRRSV (**Figure 5B**), although it is possible such transcripts might be observed if a later timepoint was sampled and/or longer RNASeq inserts were generated, as the shorter read lengths purified for these libraries (and NA PRRSV 9 hpi replicate 1; see **Figure 1C** and **Figure 1—figure supplement 6A**) are less amenable to detection of junctions.

The numerous novel transcripts described in this section not only present opportunities for regulation of the known PRRSV proteins, but also highlight considerable flexibility in the transcriptome, which provides a platform for expression of truncated protein variants and novel ORFs. Nonetheless, it is likely that many of the lowly abundant novel transcripts are simply an unavoidable consequence of a viral replication complex that has evolved to facilitate discontinuous transcription as an essential component of the viral life cycle.

## Characterisation of the PRRSV translatome

To characterise the viral translatome, RiboSeq reads were mapped to the host and viral genomes using STAR, which formed the input for PRICE (**Erhard et al., 2018**). PRICE detected 14 novel NA PRRSV ORFs and 8 novel EU PRRSV ORFs (**Figure 6**, **Figure 6—figure supplements 1 and 2**, **Figure 6—source data 1**). An additional NA PRRSV library (**Figure 6—figure supplement 3**), which had been harvested after CHX pre-treatment, was also inspected as CHX pre-treatment may emphasise initiation peaks (albeit less efficiently than specific initiation inhibitors such as harringtonine) (**Ingolia et al., 2009**; **Ingolia et al., 2012**). For NA PRRSV (**Figure 6A**), four of the novel ORFs overlap the ORFs encoding the structural proteins and may be expressed from the array of non-canonical sgRNAs discovered in this part of the genome. Most of the other novel ORFs overlap ORF1b and are likely expressed from the novel ORF1b sgRNAs described above, consistent with the fact that their translation is predominantly observed at late timepoints (**Figure 6B**, **Figure 6—figure supplement 1**). Some

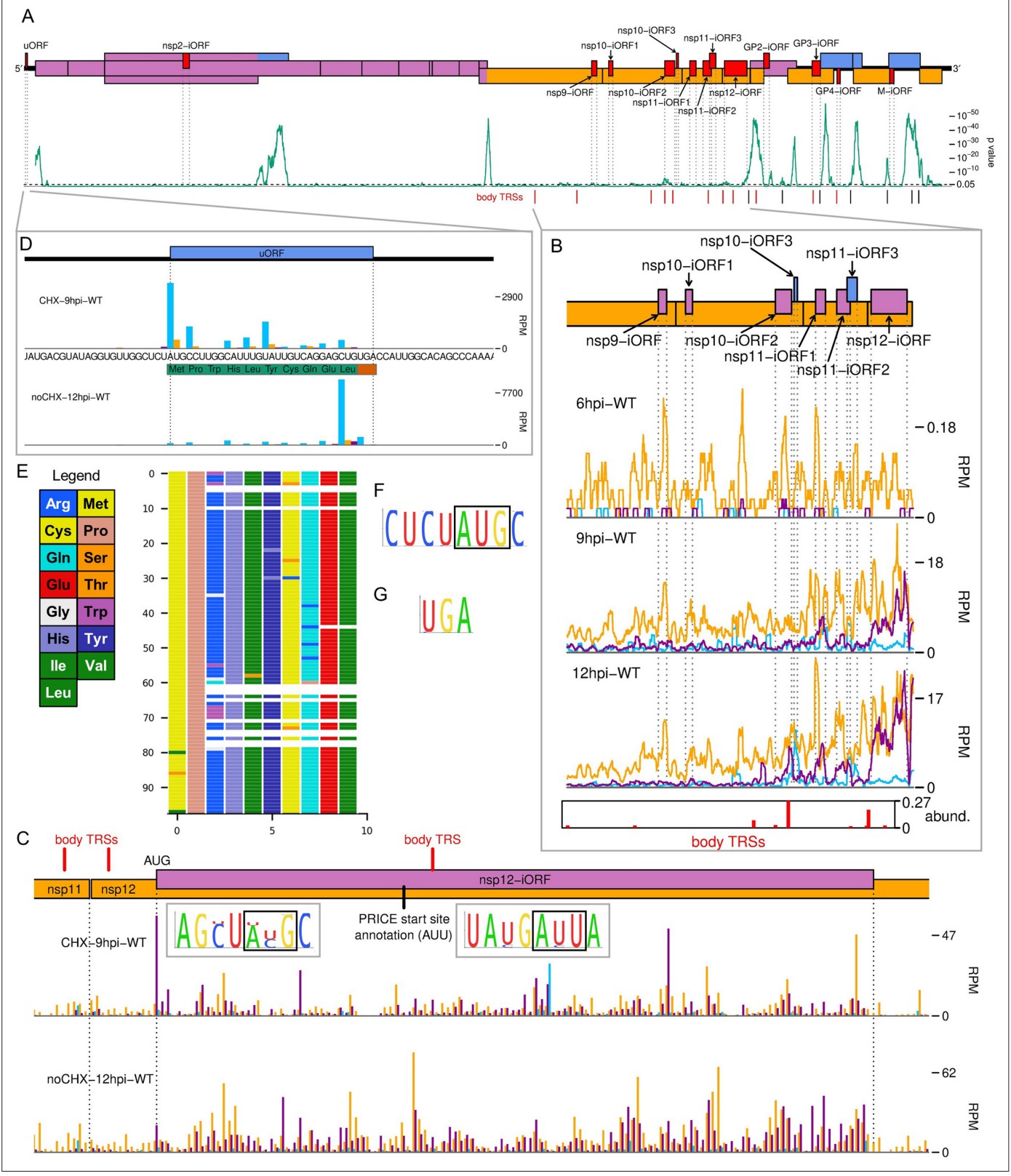

**Figure 6.** The North American porcine reproductive and respiratory syndrome virus (NA PRRSV) translatome. (**A**) Locations of novel ORFs (red) in the NA PRRSV genome, offset on the y-axis according to frame relative to ORF1a. Below this is a SYNPLOT2 (*Firth, 2014*) analysis of synonymous site conservation in the canonical protein-coding regions, based on 137 NA PRRSV genomes representative of NA PRRSV diversity (see Materials and methods). The green line represents the probability (over a 25-codon sliding window) that the observed conservation could occur under a null model

*Figure 6 continued on next page*

*Figure 6 continued*

of neutral evolution at synonymous sites; with peaks indicative of overlapping functional elements (such as ORFs). Locations of known (black) and selected novel (red) body transcription regulatory sequences (TRSs) are indicated below. Our analysis of the European (EU) PRRSV translatome can be found in *Figure 6—figure supplement 2*. (**B**) Translation of novel ORFs overlapping ORF1b. Reads mapping to the NA PRRSV genome between the ORF1b sgRNA 1 body TRS and the end of ORF1b are shown, separated according to phase and plotted after application of a 15-codon running mean filter. Novel ORFs on the genome map are coloured according to frame relative to ORF1a. Positions of moderately frequently used body TRSs (at least 44 junction-spanning reads) are indicated by red bars at the bottom of the plot, with the height of each bar scaled according to transcript abundance ('abund.'; see Materials and methods for details). The libraries displayed are those in *Figure 2C*, with remaining replicates and KO2 libraries in *Figure 6—figure supplement 1*. (**C**) Main: ribosome-protected fragment (RPF) distribution on the region of the NA PRRSV genome predicted to contain non-structural protein (nsp)12-iORF. RPFs are coloured according to phase and plotted without application of a sliding window. Quality control analyses for CHX-9hpi-WT are in *Figure 6—figure supplement 3*, and noCHX-12hpi-WT is replicate 1. The positions of body TRSs with ≥50 junction-spanning reads at 12 hr post-infection (hpi) are indicated by a red bar at the top of the genome map. The final initiation codon predicted by PRICE is an AUU codon, indicated by a black line. However, the observed RPF profiles are more consistent with the N-terminally extended ORF annotated in this plot, for which ribosomes would initiate at the upstream AUG. Insets: conservation of the context of the upstream AUG (left) and PRICE-predicted AUU (right), based on 661 available sequences for full NA PRRSV genomes. The putative initiator codons are indicated by black boxes. (**D**) Distribution of RPFs mapping to the region of the NA PRRSV 5′ UTR containing the upstream ORF (uORF). Plot constructed as in panel (**C**), with the genome sequence in this region, and the uORF amino acid sequence, underlaid. Note that cycloheximide (CHX) pre-treated libraries typically have heightened initiation peaks, while libraries harvested without CHX pre-treatment have heightened termination peaks. (**E**) Predicted amino acid sequences of the uORF from 98 PRRSV genomes representative of NA PRRSV diversity. Visualisation made using CIAlign (*Tumescheit et al., 2022*), with each row representing one sequence, each coloured rectangle representing an amino acid, and gaps indicating translation termination due to a stop codon. (**F, G**) Conservation of (**F**) the initiation context and (**G**) the stop codon for the NA PRRSV uORF. The initiator AUG is indicated by a black box. The initiation context of this ORF is weak, as defined by the absence of a G at position +4 or a A/G at position –3 relative to the A of the AUG, but the sequence is highly conserved.

The online version of this article includes the following source data and figure supplement(s) for figure 6:

**Source data 1.** ORFs in the porcine reproductive and respiratory syndrome virus (PRRSV) translatome detected by PRICE.

**Figure supplement 1.** Further replicates of ribosome-protected fragments (RPFs) mapping to novel ORFs overlapping ORF1b.

**Figure supplement 2.** The European porcine reproductive and respiratory syndrome virus (EU PRRSV) translatome.

**Figure supplement 3.** Quality control for North American porcine reproductive and respiratory syndrome virus (NA PRRSV) cycloheximide (CHX) dataset.

of these ORFs are highly translated – for example, the 125-codon NA PRRSV nsp12-iORF is translated at a level similar to nsp12 at 12 hpi (*Figure 6C*). To test whether these novel ORFs in either virus are subject to purifying selection (an indicator of functionality), we analysed synonymous site conservation within the known functional viral ORFs (*Figure 6A*, *Figure 6—figure supplement 2A*). Overlapping functional elements are expected to place additional constraints on evolution at synonymous sites, leading to local peaks in synonymous site conservation. While such peaks were observed in the regions where the known viral ORFs overlap (and also within the M ORF and at the 5′ end of ORF1a), no large conservation peaks were observed in the vicinity of the novel, translated overlapping ORFs, indicating their functional relevance is debatable (*Figure 6A*, *Figure 6—figure supplement 2A*).

As mentioned earlier, we also identified a uORF in the NA PRRSV 5′ UTR (*Figure 6D*), which is highly expressed at all timepoints (*Figure 2C*, blue peak). At only 10 amino acids, the peptide expressed from this uORF is unlikely to be functional, and the ORF is truncated or extended in a small proportion of isolates (*Figure 6E*). However, the presence of a uORF in this position is highly conserved in NA PRRSV (*Figure 6E–G*), with the initiator AUG present in 558/564 available sequences, and relatively efficient (*Kearse and Wilusz, 2017*) non-canonical initiation codons (GUG, AUA, or ACG) in the remainder. This suggests the uORF may have advantages for viral fitness, for example, by modulating translation of other ORFs.

## Quantification of viral gene expression

Next, we quantified viral transcription and translation to better understand PRRSV gene expression profiles and determine the contribution of the novel transcripts and ORFs. RiboSeq read density (in reads per kilobase per million mapped reads [RPKM]) was calculated using the PRICE output, and transcript abundance was quantified based on the number of junction-spanning reads (in RPM). ORFs were paired with the transcripts from which they are most likely expressed (*Figure 7*), and translation efficiency calculated as the RiboSeq read density divided by the transcript abundance.

Consistent with the results shown in *Figure 2*, gRNA is by far the most abundant viral transcript at 6 hpi, after which there is a marked shift towards sgRNA production at 9 hpi (*Figure 8A and B*,

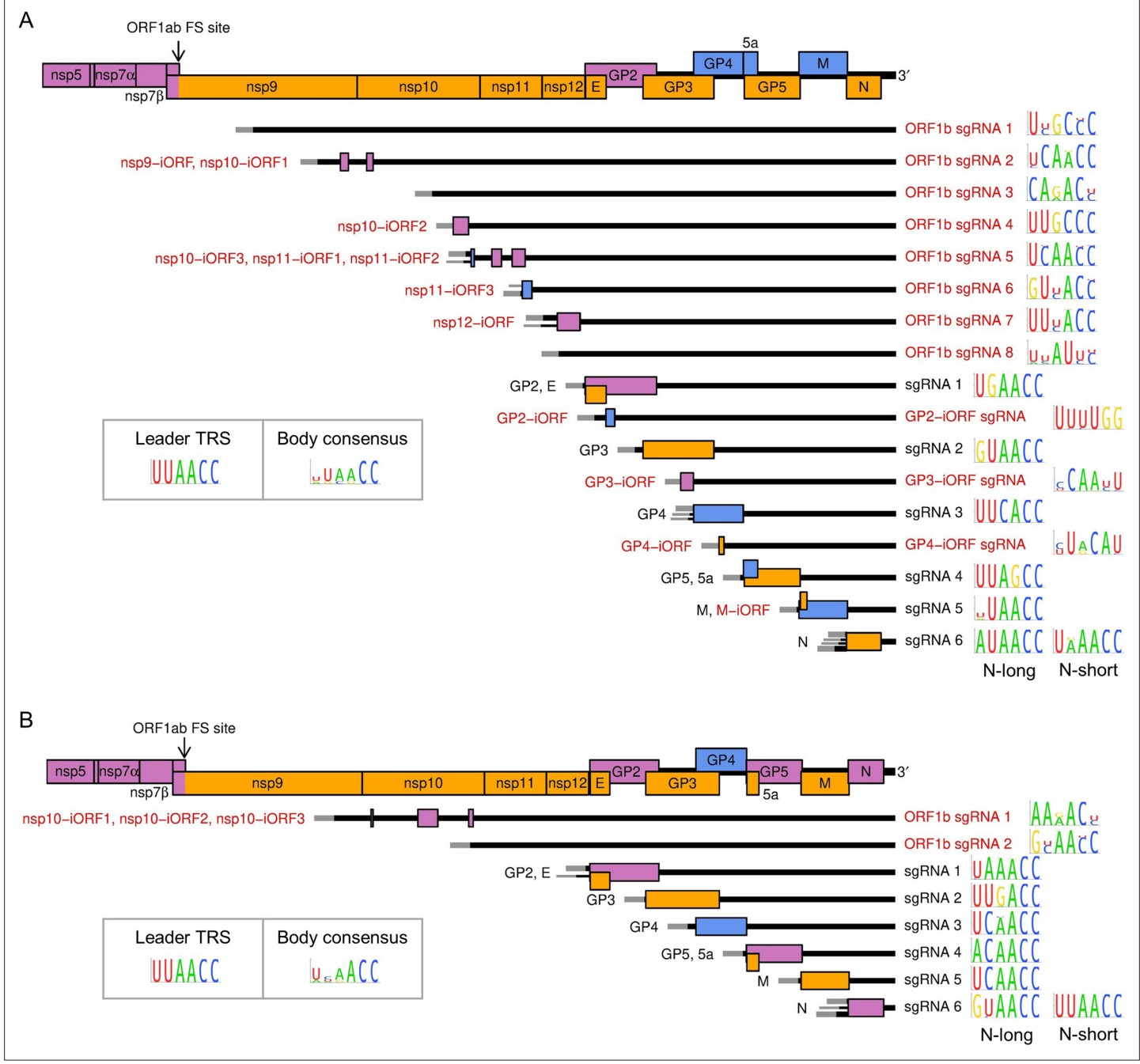

**Figure 7.** Subgenomic (sg) mRNA transcripts and ORFs included in viral gene expression analysis of (**A**) North American porcine reproductive and respiratory syndrome virus (NA PRRSV) and (**B**) European PRRSV (EU PRRSV). Canonical transcripts and ORFs are labelled in black, novel ones in red. The genome map from nsp5 onwards is reproduced above for comparison. The leader (grey) is treated as a separate transcript for the purposes of these analyses (see Materials and methods), and the NA PRRSV upstream ORF (uORF) putatively expressed from it was omitted from these plots for clarity. Where more than one 5' UTR is depicted for some mRNAs, this indicates that multiple merged junctions were detected that likely give rise to transcripts from which the same ORF(s) are translated, in which case junction-spanning read counts for these junctions were combined. To the right of each transcript, the consensus sequence of the body transcription regulatory sequence (TRS) used to generate the major transcript variant (indicated by the thicker UTR) is plotted, based on (**A**) 661 NA PRRSV or (**B**) 120 EU PRRSV genome sequences. For ease of identification, both N-long and N-short are depicted as major transcripts for N. The body TRS consensus (inset) is based on a combination of the consensus sequences of all the canonical body TRSs (omitting N-short for NA PRRSV and N-long for EU PRRSV). In addition to these sgRNAs and ORFs, ORF1a and all novel ORFs not depicted here were included in the analysis and designated as expressed from the gRNA transcript.

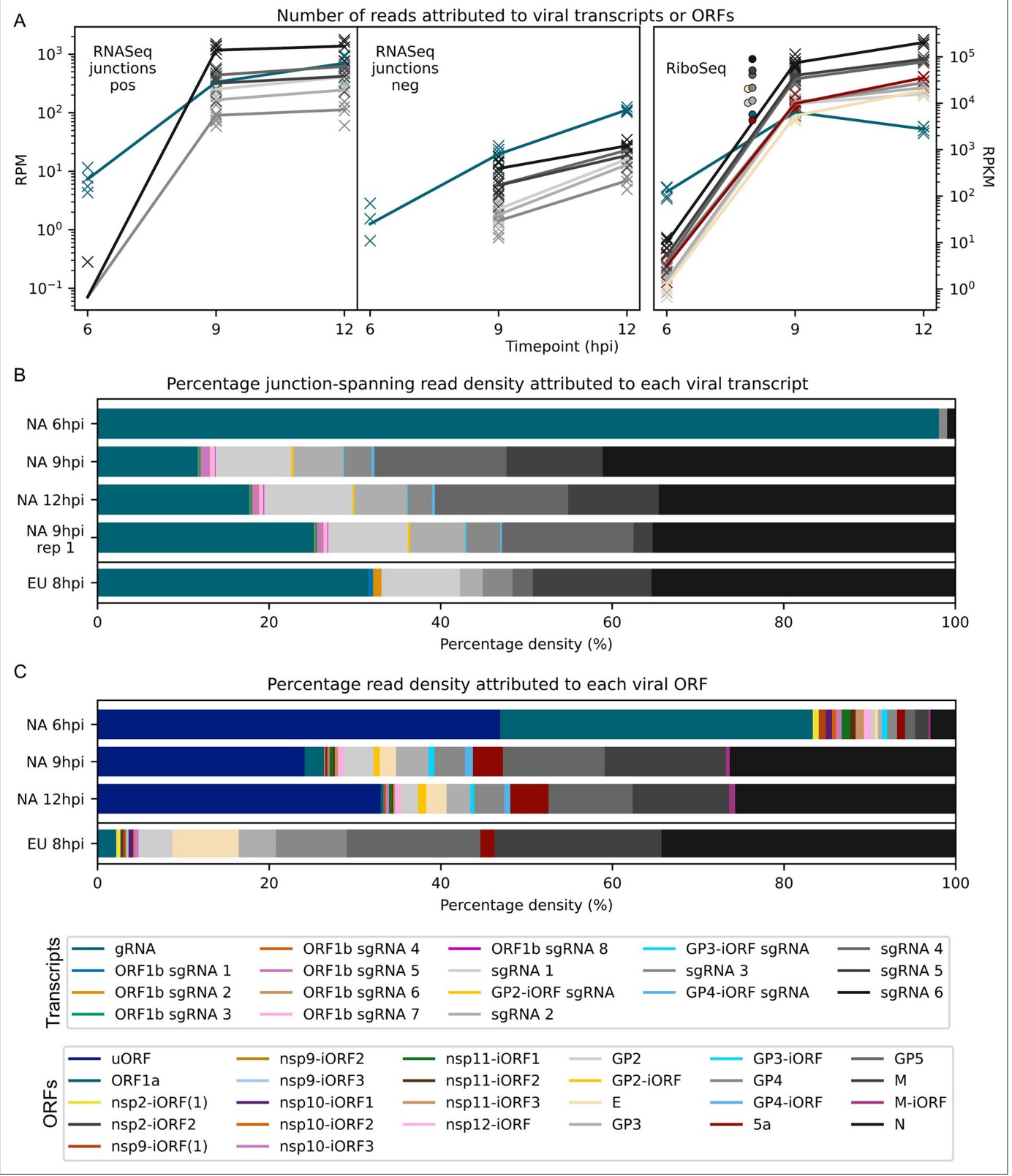

**Figure 8.** Viral transcript abundance and total translation of viral ORFs. (**A**) Left: positive-sense (pos) and negative-sense (neg) junction-spanning read density attributed to canonical viral transcripts. Mean values are indicated by the line graph, with individual datapoints plotted as crosses. Due to the shorter RNASeq fragment lengths, North American porcine reproductive and respiratory syndrome virus (NA PRRSV) 9 hr post-infection (hpi) replicate 1 libraries and European (EU) PRRSV libraries are not comparable to the remaining NA PRRSV libraries in this analysis, so are omitted from this plot and

*Figure 8 continued on next page*

*Figure 8 continued*

shown separately in *Figure 8—figure supplement 1*. Right: RiboSeq read density attributed to canonical viral ORFs, based on the PRICE read count values. Mean values for EU PRRSV are plotted as filled circles, with individual datapoints omitted for clarity and some circles offset on the x-axis to aid visualisation. ORF1b is omitted from this analysis and investigated separately in *Figure 9A–C*. The legend for colours in all panels is displayed beneath panel (**C**). Analysis of non-canonical ORFs can be found in *Figure 8—figure supplement 2*, and translation efficiency results in *Figure 8—figure supplement 3*. The source data for this figure is the same as the source data for *Figures 4–6*. (**B**) Percentage of the viral transcriptome represented by each transcript (leader omitted; see Materials and methods for details). (**C**) Percentage of the viral translatome represented by each ORF. Note that the novel ORFs detected on the EU PRRSV genome were named according to the same convention as for NA PRRSV novel ORFs, but equivalent names does not indicate that they are equivalent ORFs.

The online version of this article includes the following figure supplement(s) for figure 8:

**Figure supplement 1.** Number of junction-spanning reads attributed to viral transcripts.

**Figure supplement 2.** RiboSeq density attributed to each viral ORF.

**Figure supplement 3.** Translation efficiency (TE) of viral ORFs.

*Figure 8—figure supplement 1*). Between 9 and 12 hpi, the proportion of gRNA increases slightly; however, this may be partly related to changes in abundance of heteroclite sgRNAs, which are not discriminated from gRNA in this analysis, and are investigated separately in the next section. The relative abundance of each sgRNA remains fairly stable between 9 and 12 hpi (*Figure 8A and B*, *Figure 8—figure supplement 1*), consistent with findings for SHFV and MHV (*Di et al., 2017*; *Irigoyen et al., 2016*), and non-canonical sgRNAs make up a relatively small proportion of the viral transcriptome (*Figure 8B*, *Figure 8—figure supplement 1*). The results for negative-sense transcripts broadly mirror the positive-sense results, although negative-sense gRNA is proportionally more abundant (*Figure 8A*, *Figure 8—figure supplement 1*). The overall transcriptional profile of EU PRRSV resembles that of NA PRRSV at 9 hpi, although gRNA is more abundant and there are some differences in the relative proportions of canonical sgRNAs (*Figure 8B*, *Figure 8—figure supplement 1*).

Analysis of RiboSeq datasets revealed a similar trend to the RNASeq analysis of NA PRRSV, with ORF1a translation predominating at 6 hpi, while sgRNA translation dominates at 9 hpi (*Figure 8A and C*, *Figure 8—figure supplement 2*). ORF1a translation declines by 12 hpi, despite the increase in transcript abundance, perhaps reflecting the sequestration of gRNA through genome packaging, reducing the pool available for translation. Consistent with this, ORF1a, which has relatively low TE throughout infection, is the only canonical NA PRRSV ORF for which TE decreases over time, decreasing from ~20 at 6 and 9 hpi to 4.4 at 12 hpi (*Figure 8—figure supplement 3*; explored in more detail below).

Strikingly, the uORF is among the most highly translated NA PRRSV ORFs at all timepoints (*Figure 8C*, *Figure 8—figure supplement 2*), although this may be somewhat inflated by the heightened termination peak having a proportionally greater effect on RPKM for a small ORF such as this. Its high expression at 6 hpi indicates that the uORF is expressed from gRNA, as this is by far the most abundant viral transcript at this timepoint (*Figure 8B*). The increase in the ratio of uORF compared to ORF1a translation at later timepoints (*Figure 8C*, *Figure 8—figure supplement 2*), when sgRNAs become abundant, suggests that the uORF is also translated from the sgRNAs.

Except for the absence of a uORF, the relative translation levels of EU PRRSV ORFs are similar to those in NA PRRSV, although with less translation of 5a (*Figure 8C*, *Figure 8—figure supplement 2*). This may reflect the different relative arrangements of GP5 and 5a for these two isolates, with 5a beginning 5 nt downstream of the beginning of GP5 for EU PRRSV and 10 nt upstream for NA PRRSV. TE values for EU PRRSV are slightly higher than those for NA PRRSV (*Figure 8—figure supplement 3*); however, this may be influenced by reduced accuracy of transcript abundance quantification due to the shorter read lengths of the EU libraries.

Novel ORFs make up a relatively small proportion of total viral translation (*Figure 8C*). Nonetheless, they may represent a significant contribution to the viral proteome – for example, the novel ORFs overlapping the end of ORF1b have a similar density of ribosomes as ORF1a at 12 hpi (*Figure 8C*, *Figure 8—figure supplement 2*). As described above (*Figure 6A*, *Figure 6—figure supplement 2A*), these overlapping ORFs are not subject to noticeable purifying selection, indicating that they are unlikely to produce functional proteins. This raises the possibility that their translation is tolerated as a side effect of ORF1b sgRNA production, which may primarily function to regulate expression of ORF1b itself (explored in the following section).

## Examining the potential for non-canonical transcripts to modulate non-structural protein stoichiometry

After characterising the translation of novel ORFs present on some of the transcripts discovered in our junction-spanning read analysis, we wanted to determine whether non-canonical transcripts had the potential to modulate expression of canonical PRRSV proteins. Transcripts, such as the ORF1b sgRNAs and the heteroclite sgRNAs, which might permit expression of a portion of ORF1a or ORF1b could potentially result in modulation of the stoichiometry of the nsps which make up the polyprotein. Although these non-canonical sgRNAs are less abundant than gRNA, from which ORF1ab is canonically expressed, they are likely to be translated more efficiently. As described above, gRNA has poor TE, likely due to containing signals targeting it for replication and/or packaging, meaning only a fraction of the transcript pool is available for translation. The non-canonical sgRNAs may lack some of these signals and therefore be more available for translation, meaning they could have a considerable effect on the expression of polyprotein products, despite relatively low transcript abundance.

First, we investigated the possibility that C-terminal portions of ORF1b are translated from the ORF1b sgRNAs, inspired by the increased ORF1b-phase RiboSeq density in the 3′-proximal region of ORF1b at late timepoints (*Figure 6B*, *Figure 6—figure supplements 1 and 2B*). Correlating these density changes with the positions of ORF1b sgRNA body TRSs supported this conclusion, revealing step increases in ORF1b-phase ribosome density after some of the ORF1b sgRNA body TRSs (*Figure 9A and B*, *Figure 9—figure supplement 1*). This was confirmed by quantification of density in the regions between these body TRSs (*Figure 9C*). At 6 hpi, when no ORF1b sgRNAs are detected, read density remains reasonably constant throughout ORF1b, while at later timepoints, as ORF1b sgRNA expression increases, a pattern of increasing density towards the 3′ end of ORF1b emerges, with the 3′-most regions more highly translated than ORF1a (*Figure 9C*).

For NA PRRSV, the greatest step increases are observed after the ORF1b sgRNA 2, 5, and 7 body TRSs (*Figure 9A and C*) – the only non-canonical sgRNAs in *Figure 7* which have just a single mismatch in the body TRS compared to the leader TRS. These body TRSs are also well-conserved, particularly the final two Cs, identified as the most highly conserved part of the canonical sgRNA body TRS consensus in this and other studies (*Nelsen et al., 1999*; *Zhang et al., 2021b*; *Lin et al., 2002*; *Figure 7*). This raises the likelihood that such body TRSs may also produce ORF1b sgRNAs in other isolates of NA PRRSV. Indeed, while this article was under review, a study was published in which junction-spanning reads were analysed at 12 hpi in NA-PRRSV-infected porcine alveolar macrophages (*Zhang et al., 2021b*). These authors found similar ORF1b sgRNAs, despite using isolates from divergent areas of the NA PRRSV phylogeny (XM-2020 and GD from lineages 1 and 8, respectively, compared to SD95-21 from lineage 5 in the present work). The similarity was most pronounced for ORF1b sgRNA 5, for which the GD isolate was found to use a body TRS in precisely the same genomic location as SD95-21, and the XM-2020 isolate used one just 24 nt upstream (*Zhang et al., 2021b*). Furthermore, although the body TRSs for the EU PRRSV ORF1b sgRNAs are less well-conserved within the species (*Figure 7*), they are located at very similar positions on the genome compared to the NA PRRSV ORF1b sgRNA 2 and 5 body TRSs (which correlate with two of the greatest increases in ORF1b-phase RiboSeq read density for NA PRRSV). Strikingly, the resemblance is again greatest for NA PRRSV ORF1b sgRNA 5, which has a body TRS in a genomic location exactly equivalent to EU PRRSV ORF1b sgRNA 2, and both body TRSs have only a single mismatch compared to the leader TRS, despite this not being a requirement for maintaining the amino acid identities at this position. The conservation of these features of ORF1b sgRNAs between these highly divergent arterivirus isolates suggests that there may be a selective advantage in their production, which could result from temporal modulation of the stoichiometry of nsps 10–12.

Similarly, the heteroclite sgRNAs have the potential to modulate the stoichiometry of ORF1a. To examine this, the read density in ORF1ab was partitioned between gRNA and heteroclite sgRNAs (a distinction not made in the previous analyses) using a 'decumulation' procedure introduced in *Irigoyen et al., 2016* (*Figure 9D*). This revealed evidence of heteroclite sgRNA translation at all timepoints, with the highest ratio of heteroclite:gRNA translation being reached at 12 hpi, consistent with the increased ratio of heteroclite:gRNA RNASeq density at this timepoint (*Figure 9D*). This supports the hypothesis that the N-terminal region of ORF1a can be independently translated from heteroclite sgRNAs (besides from gRNA as part of pp1a/ab) during infection, which could function to increase the ratio of nsp1α and nsp1β compared to the other nsps (*Figure 9D*). Consistent with the

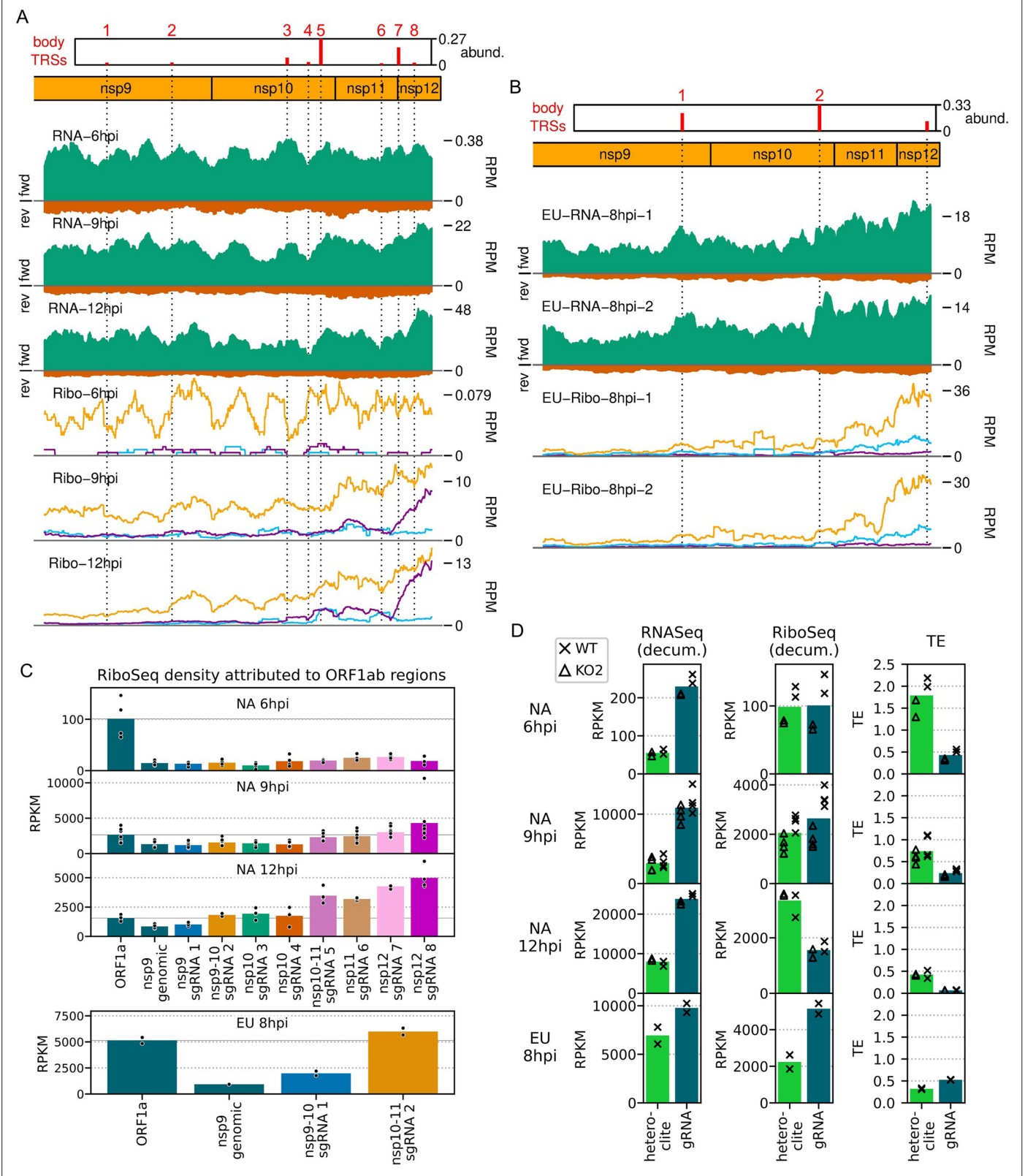

**Figure 9.** Expression of specific regions of ORF1a and ORF1b. (**A**) Distribution of RNASeq (upper) and RiboSeq (lower) reads mapping to the ORF1b region of the North American porcine reproductive and respiratory syndrome virus (NA PRRSV) genome. Plots constructed as in *Figures 2B and 6B*, respectively, with the application of a 213-nt running mean filter. Dotted lines indicate body transcription regulatory sequence (TRS) positions, with junction-spanning read abundances supporting body TRSs reproduced from *Figure 6B*, and the designated ORF1b subgenomic (sg)RNA number

*Figure 9 continued on next page*

*Figure 9 continued*

indicated above in red. The libraries displayed are those in **Figure 2B and C**, with remaining replicates and KO2 libraries in **Figure 9—figure supplement 1**. (**B**) Distribution of RNASeq (upper) and RiboSeq (lower) reads mapping to the ORF1b region of the European (EU) PRRSV genome. Plot constructed as in panel (**A**) using RiboSeq read lengths with good phasing and with junction-spanning read abundances supporting body TRSs reproduced from **Figure 6—figure supplement 2B**. The body TRS annotated at the end of non-structural protein (nsp)12 does not represent an ORF1b sgRNA, but is expected to produce an alternative transcript for GP2. (**C**) RiboSeq read density attributed to different regions of ORF1b. ORF1b was divided into regions based on the positions of the ORF1b sgRNA body TRSs, and RiboSeq density of reads in-phase with ORF1b was determined. All sgRNA numbers in the x-axis labels refer to ORF1b sgRNAs. RiboSeq density in ORF1a (the 'gRNA' region from panel **D**) is included for comparison, and its mean value is indicated by a solid grey line. Plot constructed as in **Figure 8—figure supplement 1** using a linear scale. (**D**) Gene expression of the heteroclite sgRNAs compared to gRNA. Transcript abundance and RiboSeq density for gRNA was distinguished from that of the heteroclite sgRNAs by decumulation ('decum.'), as described in Materials and methods. Although no junctions were detected for putative heteroclite sgRNAs in the EU dataset, regions were designated analogously to NA PRRSV, for comparison. Plot constructed as in **Figure 8—figure supplement 1** using a linear scale and with WT and KO2 values indicated by crosses and triangles, respectively.

The online version of this article includes the following figure supplement(s) for figure 9:

**Figure supplement 1.** Distribution of (**A**) RNASeq and (**B**) RiboSeq reads mapping to the ORF1b region of the North American porcine reproductive and respiratory syndrome virus (NA PRRSV) genome, further replicates.

previous analysis (**Figure 8—figure supplement 3**), the TE of ORF1a decreases over time, on both the gRNA and the heteroclite sgRNAs (**Figure 9D**). For NA PRRSV, the TE of the heteroclite sgRNAs is higher than that of gRNA at all timepoints (**Figure 9D**), potentially due to differences such as the extent to which these transcripts are sequestered for replication. Despite the absence of detectable EU PRRSV heteroclite sgRNAs in the junction-spanning read analysis (**Figure 5B**), analogous calculations were performed to investigate heteroclite sgRNA and ORF1a expression in EU PRRSV (**Figure 9D**), revealing RNASeq and RiboSeq outcomes consistent with the presence of translated heteroclite sgRNAs. These transcripts could potentially be present below the threshold of detection for the junction-spanning read analysis pipeline. Taken together, these results demonstrate that the non-canonical transcripts discovered in this study provide a potential mechanism to temporally regulate the stoichiometry of the polyprotein components, which may reflect changing requirements for the different nsps throughout infection.

## Investigation of PRF on the viral genome

Another key mechanism by which the stoichiometry of the polyprotein components is controlled is PRF. The ORF1ab frameshift site facilitates a reduction in the ratio of nsp9–12 compared to the upstream proteins (**Meulenberg et al., 1993**; **Nelsen et al., 1999**), whereas frameshifting at the nsp2 site produces three variants of nsp2 and causes a proportion of ribosomes to terminate before reaching nsp3 (**Fang et al., 2012**; **Li et al., 2014**; **Napthine et al., 2016**; **Li et al., 2018**). The occurrence of both frameshift events is evident on the WT NA and EU PRRSV genomes from the changes in phasing after the PRF sites (**Figure 10A and B**, **Figure 10—figure supplements 1–4**).

We began by quantifying the efficiency of frameshifting at the nsp2 site. Commonly, from profiling data, frameshift efficiency is calculated using the ratio of the read density upstream of the PRF site compared to downstream, where density is expected to be lower due to termination of either the 0-frame or the transframe ORF (**Hill et al., 2021a**; **Irigoyen et al., 2016**; **Dinan et al., 2019**; **Finkel et al., 2021b**). However, at the NA PRRSV nsp2 PRF site, ribosome drop-off at the end of nsp2N and nsp2TF is not evident, and instead there is an increase in RiboSeq read density after the frameshift site at 9 and 12 hpi (**Figure 10A**, **Figure 10—figure supplement 1**). The reason for this is unclear; perhaps it is a consequence of expressed non-canonical transcripts below the threshold of detection, or biological and/or technical biases. This increase is not seen in the counterpart RNASeq libraries (**Figure 10—figure supplement 5**), and is not related to frameshifting, as it also occurs in the KO2 mutant, in which nsp2 frameshifting is prevented (**Figure 10A**, **Figure 10—figure supplement 1**). Initially, drawing on our previous work on cardioviruses (**Napthine et al., 2017**; **Hill et al., 2021a**), we attempted to normalise potential biasing effects such as these density changes by dividing the RiboSeq profile for the WT virus by that of the KO2 mutant. However, calculations of nsp2-site PRF efficiencies using these normalised read densities were quite variable (**Figure 10—figure supplement 6A and B**; see Materials and methods for details). This may be due to the modest level of frameshifting at this site (see below), meaning ribosomal drop-off is low relative to the level of non-frameshift translation.

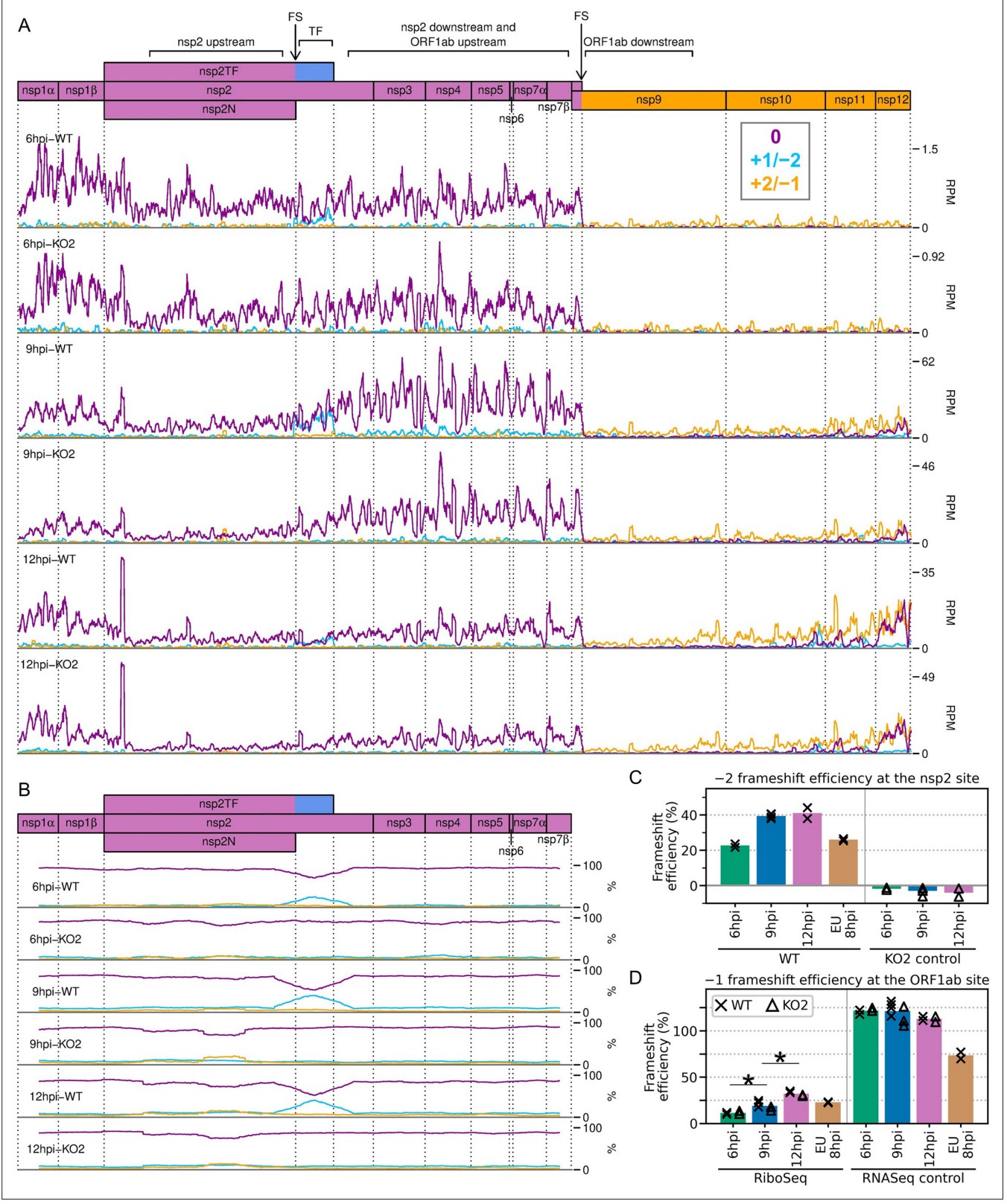

**Figure 10.** Frameshifting on the porcine reproductive and respiratory syndrome virus (PRRSV) genome. (**A**) Distribution of RiboSeq reads in each phase in the ORF1ab region of the North American (NA) PRRSV genome. Plot constructed as in *Figure 2C*. Regions used in the frameshift efficiency calculations for the non-structural protein (nsp)2 and ORF1ab sites are annotated above the genome map ('TF' = transframe). Replicates shown are noCHX-Ribo-6hpi-WT-2, noCHX-Ribo-6hpi-KO2-2, noCHX-Ribo-9hpi-WT-4, noCHX-Ribo-9hpi-KO2-3, noCHX-Ribo-12hpi-WT-1, and noCHX-Ribo-

*Figure 10 continued on next page*

*Figure 10 continued*

12hpi-KO2-1, with remaining replicates in *Figure 10—figure supplement 1*. The heightened peak shortly after the beginning of nsp2 corresponds to ribosomes with proline codons, which are known to be associated with ribosomal pausing (*Ingolia et al., 2011*; *Artieri and Fraser, 2014*; *Pavlov et al., 2008*), in both the P and A sites (P site genomic coordinates 1583–1585). The region upstream of this peak is the 'heteroclite' region analysed in *Figure 9D*, with the increased ribosome density in this region likely corresponding to heteroclite sgRNA translation. (**B**) Percentage of RiboSeq reads in each phase across the ORF1a region of the NA PRRSV genome, plotted using a 183-codon running mean filter (see Materials and methods). Replicates shown are those from panel (**A**), with remaining replicates in *Figure 10—figure supplement 2*. (**C**) Bar chart of −2 programmed ribosomal frameshift (PRF) efficiency at the nsp2 site, calculated based on the differences in phasing in the upstream and transframe regions (data from *Figure 10—figure supplement 4*). Bars represent the mean results for each group, with individual datapoints overlaid as crosses (WT) and triangles (KO2). The KO2 libraries provide a negative control (expected value ~0%). (**D**) Percentage frameshift efficiency at the ORF1ab site, calculated based on differences in read density upstream and downstream of the frameshift site. Plot constructed as in panel (**C**), with WT and KO2 scatter points offset on the x-axis to aid visualisation. The right-hand panel shows the results of applying these calculations to RNASeq reads as a control, for which the expected result is ~100%. Further plots related to PRF analysis can be found in *Figure 10—figure supplement 3*–*Figure 10—figure supplement 7*.

The online version of this article includes the following figure supplement(s) for figure 10:

**Figure supplement 1.** Distribution of RiboSeq reads in each phase in the ORF1ab region of the North American porcine reproductive and respiratory syndrome virus (NA PRRSV) genome, further replicates.

**Figure supplement 2.** Percentage of RiboSeq reads in each phase across the ORF1a region of the North American porcine reproductive and respiratory syndrome virus (NA PRRSV) genome, further replicates.

**Figure supplement 3.** Distribution of reads in the ORF1ab region of the European porcine reproductive and respiratory syndrome virus (EU PRRSV) genome.

**Figure supplement 4.** Bar charts of the percentage of reads in each phase in the specified regions of the viral genome.

**Figure supplement 5.** Distribution of RNASeq reads in the ORF1ab region of the North American porcine reproductive and respiratory syndrome virus (NA PRRSV) genome.

**Figure supplement 6.** Bootstrapping and control analyses for non-structural protein (nsp)2 programmed ribosomal frameshift (PRF) efficiency.

**Figure supplement 7.** RiboSeq read densities around the non-structural protein (nsp)2 frameshift site.

This is in contrast to cardioviruses, where the frameshift efficiencies reach ~80% (*Napthine et al., 2017*; *Hill et al., 2021a*), and to frameshifting at the PRRSV ORF1ab site, where it is only frameshifted ribosomes (rather than non-frameshifted ribosomes) that contribute to downstream RiboSeq density.

Therefore, we instead quantified −2 PRF efficiency at the nsp2 site by comparing the proportion of reads in each phase in the upstream and transframe regions (see Materials and methods for details). This led to much greater reproducibility between replicates and revealed that −2 PRF efficiency increases significantly, from 23% at 6 hpi to 39% at 9 hpi, at which point it reaches a plateau (*Figure 10C*, *Figure 10—figure supplement 6*; $p<0.0005$ based on bootstrap resampling). This is only the second known example of temporally regulated PRF (after cardioviruses; *Napthine et al., 2017*) and supports a model of increasing −2 PRF efficiency as nsp1β, the viral protein responsible for stimulating PRF at this site, accumulates and then similarly starts to plateau at 9 hpi (*Figure 3D and E*). The −2 PRF efficiency on the EU PRRSV genome at 8 hpi was estimated to be 26% (*Figure 10C*, *Figure 10—figure supplement 6*), which is similar to the 20% value determined by $^{35}$S-Met radio-labelling of MARC-145 cells infected with the EU PRRSV isolate SD01-08 and harvested at 24 hpi (MOI 0.1) (*Fang et al., 2012*). The efficiency of EU PRRSV −2 PRF at 8 hpi (26%) is significantly lower than the NA PRRSV efficiency at 9 hpi (39%; $p<0.0005$ based on bootstrap resampling). This likely reflects differences between the two viruses as opposed to the difference in timepoints, as EU nsp1β has already accumulated by 8 hpi (*Figure 3F*), and gene expression analyses suggest the 8 hpi EU PRRSV samples and 9 hpi NA PRRSV samples have progressed to a similar stage of infection (e.g. see *Figure 2*). Although these higher levels (~39%) of −2 PRF have not previously been measured in the context of viral infection, nsp2-site frameshift efficiencies of up to ~50% have been previously recorded in various reporter systems (*Fang et al., 2012*; *Li et al., 2014*; *Patel et al., 2020*), confirming that this site is capable of facilitating the highly efficient −2 PRF observed here.

Frameshift efficiency at the arterivirus ORF1ab site has not previously been determined in the context of infection, although studies using transfected reporter constructs for PRRSV (*Bekaert and Rousset, 2005*) (in yeast) and EAV (*den Boon et al., 1991*) (in HeLa cells) estimated −1 PRF efficiency as 16% and 15–20%, respectively. We set out to quantify its efficiency in the context of PRRSV infection. Ribosomal drop-off is clearly evident at the ORF1ab −1 PRF site for both NA and EU PRRSV, corresponding to ribosomes which do not frameshift encountering the ORF1a stop codon shortly

downstream of the frameshift site (*Figure 10A*, *Figure 10—figure supplement 1*, *Figure 10—figure supplement 3A*). We quantified the ratio of RiboSeq read density in the region downstream of the PRF site compared to that upstream to calculate frameshift efficiency (*Figure 10D*). PRF efficiency at RNA structure-directed sites is commonly assumed to be fixed; however, surprisingly, −1 PRF efficiency at this site also increased over the course of infection, from 11% at 6 hpi to 19% at 9 hpi (p-value from two-tailed Mann–Whitney *U* test = $8.5 \times 10^{-3}$), and further increased to 32% at 12 hpi (p = $8.5 \times 10^{-3}$). The same trend was not observed in the RNASeq libraries (*Figure 10D*), which were processed as a negative control, indicating that it does not result from shared technical biases or an increase in non-canonical transcripts facilitating translation of ORF1b (note that all detected ORF1b sgRNAs are excluded from the regions used). The ORF1ab −1 PRF efficiency on the EU PRRSV genome at 8 hpi was 23%, which is similar to the calculated efficiency for NA PRRSV at 9 hpi (*Figure 10D*). This is consistent with the replicase components being required at similar stoichiometries at this stage of infection for these two viruses.

Ribosomal pausing over the slippery sequence is considered to be an important mechanistic feature of PRF (*Plant and Dinman, 2005*; *Namy et al., 2006*; *Brierley et al., 1992*; *Choi et al., 2020*), although it has been difficult to detect robustly on WT slippery sequences using ribosome profiling (*Irigoyen et al., 2016*; *Dinan et al., 2019*; *Finkel et al., 2021b*). To determine whether ribosomal pausing occurs over the nsp2 slippery sequence, we plotted the RPF distribution on the WT genome in this region and compared this to the KO2 genome to control for shared biases (*Figure 10—figure supplement 7A*). This revealed a peak on the WT genome, derived predominantly from 21-nt reads and corresponding to ribosomes paused with P site over the slippery sequence (G-G̲U̲U̲-U̲U̲U, P site pause location underlined, hyphens delineate 0-frame codons) (*Figure 10—figure supplement 7A–C*). The peak is not present on the KO2 genome, nor does it overlap the point mutations of KO2, indicating that the differences between the WT and KO2 profiles do not result from technical biases (*Figure 10—figure supplement 7D*); however, its origin is unclear. The frameshift-associated pause is thought to occur at a late stage of the translocation event which begins with the GUU codon in the P site (*Caliskan et al., 2014*; *Choi et al., 2020*; *Caliskan et al., 2017*). The positioning of the short-read peak 1 nt downstream of this could be due to an unusual frameshift-intermediate conformation (e.g. a hyper-rotated state; *Qin et al., 2014*; *Chen et al., 2014*) protecting a shorter region of mRNA in the exit tunnel. However, short RPFs are thought to originate from post-translocation ribosomes without an aminoacyl-tRNA in the A site (*Wu et al., 2019*; *Lareau et al., 2014*), suggesting that this peak could instead represent ribosomes pausing while decoding the first codon of nsp2TF, which would not be translated on the KO2 genome. This UUG (Leu) codon is normally not expected to be slow to decode, as it is well-adapted to the cellular tRNA pool (*Figure 10—figure supplement 7A*, heatmap); however, other factors, such as incomplete dissociation of the PCBP/nsp1β complex shortly downstream, could hinder decoding after frameshifting. We found no convincing evidence of ribosomal pausing at the ORF1ab −1 PRF site, and it will be interesting to see whether future ribosome profiling studies which capture the population of short RPFs find similar pauses at other sites.

## Host differential gene expression at 12 hpi

Finally, we interrogated our datasets to investigate the host transcriptional and translational response to NA PRRSV infection. Although several analyses of host differential transcription have been performed previously (*Li et al., 2018*; *Dong et al., 2021*; *Lim et al., 2020*; *Pröll et al., 2017*; *Wilkinson et al., 2016*; *Zeng et al., 2018*; *Zhang et al., 2017*; *Zhang et al., 2019a*), changes in the host translatome in response to PRRSV infection have not been determined. Here, we characterise infection-induced changes in host transcription and TE by analysing our 12 hpi libraries using xtail (*Xiao et al., 2016*) and DESeq2 (*Love et al., 2014*).

First, we compared the WT libraries against mock (*Figure 11A*, *Figure 11—source data 1*) and the KO2 libraries against mock (*Figure 11B*, *Figure 11—source data 2*). Similarly to other studies (*Pröll et al., 2017*; *Badaoui et al., 2013*; *Crisci et al., 2020*), we found transcription of genes related to regulation of the cell cycle (amongst other GO terms) to be perturbed by WT PRRSV infection (*Figure 11—source data 1*, sheet: 'GO_TS_up'; GO term GO:0051726 ~26-fold enriched in transcriptionally upregulated genes). However, a comparison of transcriptional fold changes with those of TE reveals that the majority of transcriptionally upregulated genes in WT or KO2 compared to mock are down-regulated in terms of TE (*Figure 11A and B*, top-left quadrants of right-column panels). Such an

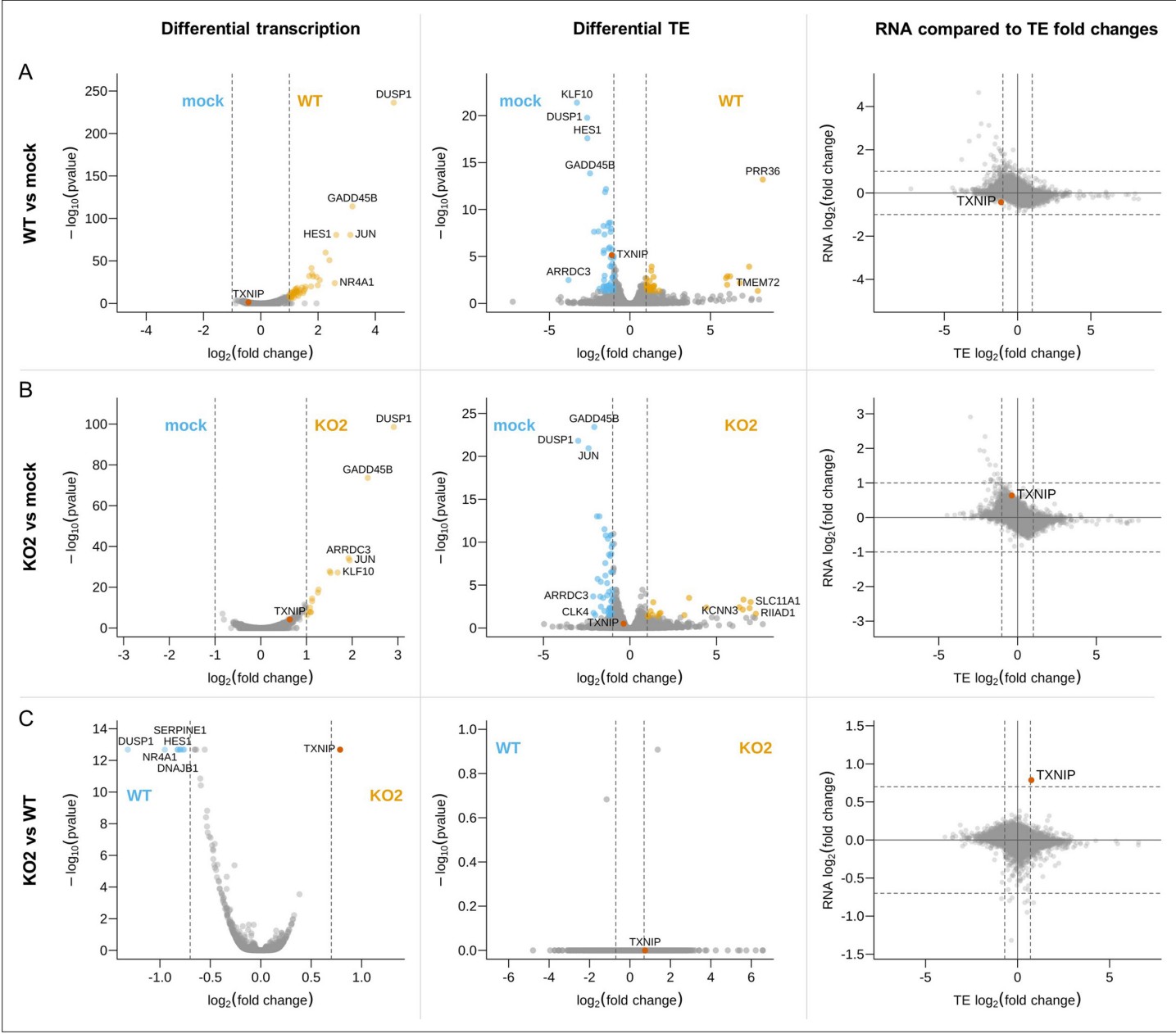

**Figure 11.** Host differential gene expression at 12 hr post-infection (hpi). Differences in transcription (left) and translation efficiency (centre) were determined using DESeq2 and xtail, respectively. Volcano plots show relative changes in pair-wise comparisons between the 12 hpi libraries ($n = 2$ biological replicates per condition): (**A**) WT and mock, (**B**) KO2 and mock, or (**C**) KO2 and WT. The y-axis shows the false discovery rate (FDR)-corrected p-values. Genes with FDR-corrected p-values≤0.05 and log$_2$(fold change) magnitudes greater than 1 (WT vs. mock and KO2 vs. mock) or 0.7 (KO2 vs. WT; thresholds in each case indicated by grey dashed lines) were considered differentially expressed and are coloured orange (upregulated) or blue (downregulated) in the volcano plots. Where gene names were available, those of the top five significantly up- or downregulated genes with the greatest fold changes are annotated, and TXNIP is annotated in red on all plots. For all genes where both RNA and translation efficiency (TE) fold changes were determinable, these were compared (right), irrespective of p-value. The full results of these differential expression analyses, including lists of GO terms enriched in each set of differentially expressed genes, are in *Figure 11—source data 1–3*.

The online version of this article includes the following source data for figure 11:

**Source data 1.** Host differential gene expression in WT vs. mock at 12 hr post-infection (hpi).

**Source data 2.** Host differential gene expression in KO2 vs. mock at 12 hr post-infection (hpi).

**Source data 3.** Host differential gene expression in KO2 vs. WT at 12 hr post-infection (hpi).

effect has previously been described as 'translational buffering' and is expected to result in little to no change in protein abundance (*Kusnadi et al., 2022*), suggesting that many of the observed transcriptional changes make only a minor contribution to the host response to infection. This is consistent with the observation that many of the GO terms enriched amongst the transcriptionally upregulated genes are also enriched in the translationally downregulated genes (*Figure 11—source data 1*, 'GO_TE_down' sheet). Comparisons between RNA and TE fold changes further reveal many genes with large fold changes of TE and little to no change at the transcriptional level (*Figure 11A and B*, right-column panels, points in centre-left and centre-right regions), suggesting that translational regulation may be a greater contributor to the host response than transcriptional changes. This is supported by the fact that several GO terms (such as those related to lipid binding and the extracellular matrix) are enriched amongst the lists of translationally regulated genes and not in the genes transcriptionally regulated in the opposing direction (*Figure 11—source data 1 and 2*).

We moved on to compare the host response to infection with WT PRRSV to that of KO2 PRRSV to investigate the effects of the nsp2 frameshift products, nsp2TF and nsp2N (*Figure 11C*, *Figure 11—source data 3*). As we expected relatively small differences in the gene expression programmes activated by the two viruses, we lowered the magnitude of the $\log_2$(fold change) required to qualify as a 'differentially expressed' gene (DEG) from 1 (in comparisons of infected vs. mock) to 0.7 to increase sensitivity. Many of the DEGs in the comparison between WT and KO2 also appear in the list of DEGs in the WT vs. mock comparison (*Figure 11A and B*, *Figure 11—source data 1 and 2*). These genes have fold changes in the same direction in both the WT vs. mock and KO2 vs. mock analyses, but those in the KO2 analysis are of a lower magnitude, likely representing the slightly slower replication kinetics of the mutant virus (*Fang et al., 2012*; *Li et al., 2018*) as opposed to meaningful differences in the host response. An exception to this is thioredoxin interacting protein (TXNIP; *Figure 11*, red). TXNIP is significantly more highly transcribed in KO2 than WT infection [$\log_2$(fold change) = 0.79, p=2.1 × 10$^{-13}$; *Figure 11C*, left column] and, although it is below our thresholds for qualification as a DEG in these analyses, it is transcriptionally regulated in opposing directions in the WT vs. mock [$\log_2$(fold change) = −0.43, p=0.055] and KO2 vs. mock [$\log_2$(fold change) = 0.64, p=7.0 × 10$^{-5}$] comparisons (*Figure 11A and B*, left column), suggesting the difference is not related to replication kinetics. No genes generated significant p-values for the KO2 vs. WT TE analysis (*Figure 11C*, middle column), in which the distribution of p-values was conservative, likely due to the similarity between the two datasets. Nonetheless, TXNIP clearly stands out in the comparison of RNA and TE fold changes as it is both more highly transcribed and more efficiently translated in KO2 than WT (*Figure 11C*, right column), further supporting the conclusion that increased TXNIP expression is a notable feature of KO2 infection.

The mechanism by which the presence of nsp2TF/nsp2N could lead to reduced TXNIP expression in WT infection is unclear. The frameshift products share a PLP2 protease domain with the 0-frame product, nsp2, although the DUB/deISGylase activity of this domain is most potent in nsp2N (*Li et al., 2018*). The frameshift proteins also have different sub-cellular distributions to nsp2 (*Fang et al., 2012*; *Guo et al., 2021*), which may grant them access to proteins involved in the transcriptional activation of TXNIP, allowing them to interfere with this signalling pathway (e.g. by de-ubiquitinating its components). While the mechanism remains elusive, there are several reasons why this downregulation may be beneficial to PRRSV. TXNIP is a key protein in metabolism and redox homeostasis (*Kim et al., 2007*; *Alhawiti et al., 2017*), regulates cell survival/apoptosis via apoptosis signal regulating kinase 1 (ASK1) (*Xiang et al., 2005*), and triggers NLRP3 inflammasome activation in monocytes and innate immune cells (*Zhou et al., 2010*; *Shalev, 2014*). TXNIP largely exerts its functions by binding and inhibiting thioredoxin (*Nishiyama et al., 1999*), an antioxidant which is central to redox signalling and homeostasis (*Lu and Holmgren, 2014*; *Matsuzawa, 2017*), and reduced cellular levels of TXNIP lead to lower concentrations of reactive oxygen species (ROS) (*Song et al., 2003*; *Lee et al., 2005*). ROS are known to be induced by PRRSV infection (*Guo et al., 2018b*; *Lee and Kleiboeker, 2007*) and lead to apoptosis (*Lee and Kleiboeker, 2007*; *Redza-Dutordoir and Averill-Bates, 2016*), which would likely be detrimental to viral replication. Further, ROS, and by extension thioredoxin and TXNIP, have particular significance for the physiology and function of macrophages (*Herb and Schramm, 2021*; *Forman and Torres, 2001*), the primary target for PRRSV in vivo (*Morgan et al., 2016*). As well as being integral to phagocytosis (*Herb and Schramm, 2021*; *Forman and Torres, 2001*), ROS exert a complex, context-dependent effect on macrophage polarisation (*Rendra et al., 2019*; *Tan et al.,*

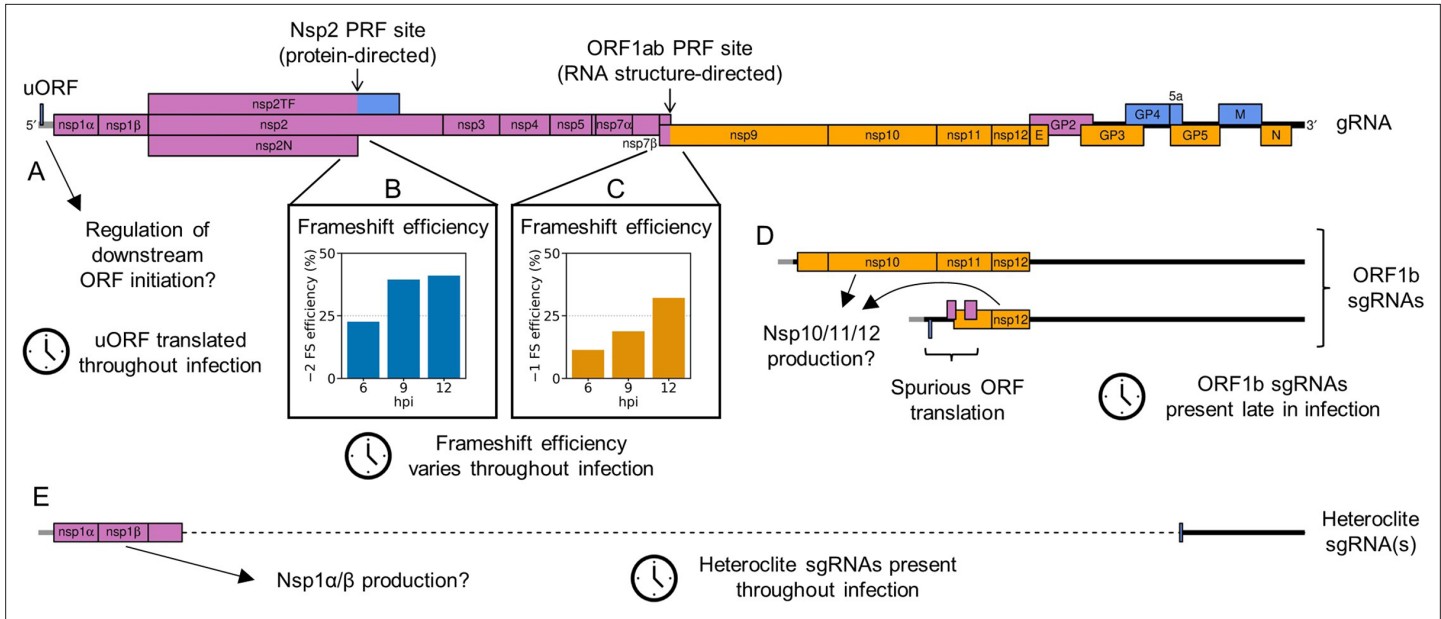

**Figure 12.** Schematic summary of non-canonical mechanisms of porcine reproductive and respiratory syndrome virus (PRRSV) gene expression regulation supported by this study. (**A**) A short upstream ORF (uORF) was discovered in the North American (NA) PRRSV 5′ leader (grey), which is shared by the gRNA and subgenomic (sg)RNAs. The uORF is highly translated throughout infection and could potentially regulate initiation of the canonical ORFs downstream. (**B**) At the protein-directed non-structural protein (nsp)2 frameshift site, −2 programmed ribosomal frameshift (PRF) efficiency significantly increases between 6 and 9 hr post-infection (hpi), at which point it reaches a plateau. This likely reflects the accumulation of the viral frameshift-stimulatory protein, nsp1β. (**C**) At the RNA structure-directed ORF1ab frameshift site, −1 PRF efficiency also increases over time, despite the lack of an obvious *trans*-acting regulatory factor. The changes in frameshift efficiency at this and the nsp2 site likely modulate the stoichiometry of the polyprotein components over the course of infection. (**D**) Non-canonical ORF1b sgRNAs are synthesised at late timepoints, from which novel ORFs overlapping ORF1b are translated. Although the overlapping ORFs are unlikely to be functional, these ORF1b sgRNAs may also provide templates for translation of ORF1b itself, contributing to temporal regulation of nsp10, 11, and 12. (**E**) Heteroclite sgRNAs, in which the beginning of ORF1a is fused to ORFs in the 3′-proximal region of the genome, are present throughout infection. These may provide a source of nsp1α and nsp1β to supplement that provided by the genomic RNA.

2016), and high levels of thioredoxin have been found to favour polarisation of M2 macrophages (*El Hadri et al., 2012*). These produce less antiviral cytokines and are less suppressive of PRRSV replication than their M1 counterparts (*Wang et al., 2017*). Therefore, reducing cellular TXNIP concentrations during WT PRRSV infection may be favourable by allowing thioredoxin to function uninhibited and preventing excessive ROS concentrations, with potential implications for macrophage physiology and polarisation. Indeed, it has already been suggested that PRRSV infection induces a skew towards M2 polarisation, although the mechanism was uncharacterised (*Wang et al., 2017*).

## Discussion

Here, we describe a high-resolution analysis of PRRSV replication through ribosome profiling and RNASeq. In addition to confirming and extending the findings of previous transcriptomic analyses, we define the PRRSV translatome and identify strategies of gene expression that may permit the virus to exert translational control during the replication cycle (*Figure 12*).

### The host transcriptional response to infection is dampened at the level of translation

Global analysis of the host response to PRRSV infection at 12 hpi reveals that many of the observed changes in transcript abundance are offset at the translational level, indicating that changes in TE of host mRNAs may play a dominant role in the response to infection. This phenomenon of host transcriptional responses being counteracted by opposing changes in TE has also been observed in response to SARS-CoV-2 infection, where it was attributed to inhibition of mRNA export from the nucleus, preventing translation (*Finkel et al., 2021a*; *Alexander et al., 2021*). This activity is known

to be associated with coronavirus nsp1, which inhibits nuclear export by interacting with the nuclear export factor NXF1 in SARS-CoV or the nuclear pore complex component Nup93 in SARS-CoV-2 (*Gomez et al., 2019*; *Zhang et al., 2021a*). Similarly, PRRSV nsp1β causes imprisonment of host mRNAs in the nucleus by binding Nup62 to cause the disintegration of the nuclear pore complex (*Han et al., 2017*; *Ke et al., 2019*). This inhibition of host mRNA nuclear export has been found to reduce synthesis of interferon-stimulated genes (*Ke et al., 2019*), and mutations in nsp1β which ablate this activity lead to reduced viral load and increased neutralising antibody titres in pigs (*Ke et al., 2018*). These findings suggest that nsp1β-mediated nuclear export inhibition may be responsible for the translational repression seen in our host differential gene expression analyses, which, analogously to conclusions drawn for SARS-CoV-2 (*Finkel et al., 2021a*; *Alexander et al., 2021*), may be a key mechanism by which PRRSV evades the host response to infection.

## The NA PRRSV 5′ UTR contains a highly expressed uORF

On the viral genome, numerous translated novel ORFs were discovered, including a short but highly expressed uORF in the NA PRRSV 5′ leader, which is shared by the gRNA and sgRNAs (*Figure 12A*). The presence of this uORF is very well-conserved amongst NA PRRSV isolates, with an AUG in this position in 558/564 available NA PRRSV sequences, and a non-canonical initiation codon (GUG/AUA/ACG) present in the remainder. In many contexts, uORFs have been shown to regulate translation of the main ORF downstream (*Zhang et al., 2019b*) and, interestingly, uORFs have been found in many nidovirus genomes. For example, recent ribosome profiling studies of the coronaviruses MHV, IBV, and SARS-CoV-2 revealed translation of uORFs initiating within the 5′ leader (*Irigoyen et al., 2016*; *Dinan et al., 2019*; *Finkel et al., 2021b*; *Kim et al., 2021*). Intriguingly, the extent of translation initiation on the SARS-CoV-2 leader uORF differs on different viral transcripts (*Finkel et al., 2021b*; *Kim et al., 2021*), suggesting that the same could be true of the NA PRRSV uORF, and presenting a potential mechanism by which the uORF could differentially regulate translation of the canonical ORFs downstream. Looking to the *Arteriviridae* family, EAV has a highly conserved AUG within the leader (present in 94/95 available genome sequences), at genomic coordinates 14–16, predicted to permit expression of a 37-amino acid leader peptide (*Kheyar et al., 1996*). In vitro translation of the N transcript demonstrated that the uORF is translated and found it downregulated translation of the downstream N ORF (*Archambault et al., 2006*). Mutations which disrupted this AUG or altered its predicted initiation efficiency were detrimental to viral fitness and led to rapid reversion to the WT sequence (*Archambault et al., 2006*), although the uORF was not essential for virus viability (*Archambault et al., 2006*; *Molenkamp et al., 2000*). Upstream ORFs in other arteriviruses have not been characterised, although it was noted that the SHFV 5′ leader contains a putative 13-codon uORF at genomic coordinates 35–73 (*Snijder and Meulenberg, 1998*), which is conserved in all 37 available full genome sequences. This SHFV uORF is of a similar length and position to the NA PRRSV uORF (10 codons; genomic coordinates 24–36), with the uORFs in these two viruses ending, respectively, 126 nt and 128 nt upstream of the leader TRS. The similarity between these two putative uORFs suggests a possible conserved function, for example, conferring resistance to eIF2α-phosphorylation-mediated translation inhibition (as observed for some cellular uORFs; *Andreev et al., 2015*) or affecting re-initiation efficiency on the downstream ORF. The latter could help to modulate the ratio of overlapping ORFs (GP2:E and/or GP5:5a) in which the downstream AUG is thought to be accessed by leaky scanning. However, there is little evidence that EU PRRSV encodes a similar uORF as only 3 out of the 100 EU PRRSV genome sequences with any 5′ UTR have AUGs other than at the extreme 5′ end of the leader (although translation could potentially initiate from near-cognate start codons; *Firth and Brierley, 2012*). Further, we do not detect robust uORF translation in the EU PRRSV isolate used in this study, indicating that any putative function of the uORF is not conserved across the entire family.

## Frameshift efficiency at the protein-directed nsp2 site increases over time

Our analysis of canonical PRRSV ORFs over a 12 hr timecourse revealed that the expression of many of these ORFs is controlled by additional mechanisms at both the transcriptional and translational levels beyond what was previously appreciated. A key observation is that −2 PRF at the NA PRRSV nsp2 PRF site is both highly efficient and temporally regulated (*Figure 12B*). At 6 hpi its efficiency is 23%, and this increases to ~40% at 9 and 12 hpi, likely due to accumulation of the frameshift-stimulatory viral

protein, nsp1β. Such regulation may be a selective advantage for PRRSV by directing ribosomes to translate proteins which are most beneficial at each stage of infection, optimising the use of cellular resources. At early timepoints, lower nsp2 frameshift efficiency means more ribosomes continue to translate the remainder of pp1a or pp1ab, which encode components of the replication and transcription complex (RTC), which may be more important for establishing infection than translation of the accessory protein nsp2TF. Later in the replication cycle, higher −2 PRF efficiency likely corresponds to an increased requirement for nsp2TF to prevent degradation of GP5 and M (*Guo et al., 2021*), which are expressed from ~8 hpi and are essential for virion assembly (*Guo et al., 2021*; *Wissink et al., 2005*). Further, nsp2TF is a more potent innate immune suppressor than nsp2, and downregulates expression of swine leukocyte antigen class I (swine MHC class I) (*Li et al., 2018*; *Cao et al., 2016*), which may become critical later in infection as viral proteins and double-stranded RNA accumulate due to viral translation and replication. The only other known example of temporally regulated frameshifting is provided by cardioviruses (*Napthine et al., 2017*), which encode the only other known protein-stimulated PRF site (*Loughran et al., 2011*; *Finch et al., 2015*; *Napthine et al., 2017*; *Napthine et al., 2019*; *Hill et al., 2021b*; *Hill et al., 2021a*). This suggests that temporal regulation may emerge as a common feature of *trans*-activated PRF sites as more non-canonical PRF sites are discovered in future.

## Frameshift efficiency at the RNA structure-directed ORF1ab site increases over time

RNA structure-directed frameshift sites are commonly assumed to operate at a fixed ratio due to the lack of *trans*-acting factors involved; however, we found that −1 PRF efficiency at the ORF1ab site increased over time, from 11% to 32% (*Figure 12C*). As opposed to representing specific 'regulation' of PRF, we suggest that this is due to changes in gRNA translation conditions as infection progresses. Such changes could result from activation of pathways that globally regulate translation, such as the unfolded protein response (which is known to be activated by PRRSV infection; *Huo et al., 2013*) or potential phosphorylation of eEF2 (discussed above). Additionally, changes in the localisation or availability of gRNA for translation could result in changing ribosome density as infection progresses, and decreases in ribosome load have been shown to increase −1 PRF efficiency in some studies (*Lopinski et al., 2000*; *Smith et al., 2019*). The mechanism responsible for this effect is not well characterised, although it has been suggested that frameshift-stimulatory RNA structures are more likely to have time to re-fold in between ribosomes if the transcript is more sparsely occupied (*Lopinski et al., 2000*; *Smith et al., 2019*). Consistent with this hypothesis, we find a trend of decreasing gRNA TE over time, although this analysis may be confounded by, for example, the inability to discern translatable gRNA from that undergoing packaging. Whether the observed changes in PRF efficiency represent a selective advantage for PRRSV, or whether they are simply incidental, is unclear. The expected result is an increase in the ratio of ORF1b products to ORF1a products over time. This could be advantageous for PRRSV, for example, if there is a greater requirement for nsp2 and nsp3, which promote DMV formation (*Snijder et al., 2001*), early in infection to establish a protective environment for viral replication, followed by a later preference for producing more of the RdRp (nsp9) and helicase (nsp10) to promote replication itself. Alternatively, this could simply reflect that PRRSV can tolerate a reasonably wide range of ORF1a:ORF1b stoichiometries. Nonetheless, the finding that changes in the translational landscape during infection affect PRF efficiency is relevant to many other RNA viruses, and future studies may reveal temporal changes in PRF efficiency at other frameshift sites, such as those in coronaviruses and HIV. For MHV and SARS-CoV-2, previous results tentatively suggest that ORF1ab frameshift efficiency may increase over time (*Irigoyen et al., 2016*; *Kim et al., 2021*; *Puray-Chavez et al., 2021*), but temporal dependence was not assessed in detail, nor statistical significance determined, highlighting this as an interesting area for future coronavirus research.

## Non-canonical ORF1b sgRNAs may modulate expression of ORF1b

In addition to changes in the ratio of ORF1b to ORF1a translation, we observed temporal changes in the relative translation of different regions within ORF1b, with increasing translation of the 3'-proximal region as infection progresses. This may result from translation of non-canonical sgRNAs, which we term ORF1b sgRNAs, in which the body TRS is within ORF1b (*Figure 12D*). If the putative translated proteins are processed to produce functional nsps, this would be expected to increase the

stoichiometry of nsps 10–12 compared to nsp9, and alter the relative stoichiometries of nsp10, nsp11, and nsp12. There are several possible reasons this could be beneficial to viral fitness. Although the stoichiometry of the arterivirus RTC is unknown, there is some evidence that the stoichiometry of the coronavirus replication complex varies, containing either one or two copies of the helicase for each copy of the holo-RdRp (*Chen et al., 2020*). This highlights the possibility that the composition of the PRRSV RTC could change over time, for example, if extra copies of nsp10, 11, or 12 are supplied from ORF1b sgRNA translation (as well as the potential contribution of increased ORF1ab frameshift efficiency). This provides a potential mechanism of regulating viral replication, for example, by altering the ratio of gRNA to sgRNA production (as observed in *Figures 2 and 8*), as both nsp10 (the helicase) and nsp12 are thought to be involved in promoting sgRNA transcription (*Chen et al., 2020*; *van Dinten et al., 1997*; *Tang et al., 2020*; *Song et al., 2018*; *Nan et al., 2018*; *Wang et al., 2019b*). Nsp11 (NendoU) is an endoribonuclease found in many nidoviruses, which has broad substrate specificity in vitro (*Nedialkova et al., 2009*), and is also an innate immune antagonist (*Wang et al., 2015*; *Sun et al., 2016*; *Su et al., 2018*; *Wang et al., 2019a*; *Yang et al., 2019*). Its expression outside the context of infection is highly toxic (*Wang et al., 2019a*; *Shi et al., 2016*), leading to the suggestion that its restricted perinuclear localisation during infection is important to prevent its expression becoming 'suicidal' for the virus (*Snijder et al., 2013*; *Sun et al., 2016*). Therefore, it may be beneficial to maintain relatively low levels of nsp11 early in infection and increase production after the optimal microenvironment for its localisation has formed. However, such possibilities are clearly speculative at present.

Interestingly, ORF1b sgRNAs have been found in a number of other nidoviruses. Our results are highly consistent with a previous study on SHFV, in which several ORF1b sgRNAs were detected, which were predicted to produce in-frame portions of the ORF1b polyprotein, or in one case a novel overlapping ORF (*Di et al., 2017*). Quantitative mass spectrometry provided support for translation of both categories of ORF1b sgRNA and showed that peptides from nsp11 and nsp12 were 1.2- and 3.1-fold more abundant (respectively) than those from ORF1a-encoded nsp8 (*Di et al., 2017*). ORF1b sgRNAs were also found in lactate dehydrogenase-elevating virus (predicted to express the C-terminal 200 amino acids of ORF1b) (*Chen et al., 1993*), SARS-CoV-2, HCoV-229E, and equine torovirus (*Kim et al., 2020*; *Stewart et al., 2018*; *Finkel et al., 2021b*; *Viehweger et al., 2019*). Whether this has evolved by virtue of conferring a selective advantage, or whether it is a neutral consequence of the promiscuous discontinuous transcription mechanism, this suggests that ORF1b sgRNAs are a conserved feature of the nidovirus transcriptome. Further characterisation of these non-canonical transcripts would be highly informative to determine potential initiation sites and ascertain whether any in-frame products are functional.

## Non-canonical heteroclite sgRNAs may modulate expression of ORF1a

Another group of non-canonical transcripts with the potential to modulate polyprotein stoichiometry comprises those termed 'heteroclite sgRNAs' by *Yuan et al., 2000*; *Yuan et al., 2004* (*Figure 12E*). These transcripts result from large internal deletions between regions of sequence similarity in nsp1β/nsp2 and the canonical sgRNA ORFs (*Yuan et al., 2000*; *Yuan et al., 2004*), and have been found in several isolates of NA PRRSV (*Yuan et al., 2000*; *Zhang et al., 2021b*; *Yuan et al., 2004*) (including in the present work), with similar types of transcripts found in coronaviruses (*Kim et al., 2020*; *Wang et al., 2021*; *Viehweger et al., 2019*; *Taiaroa et al., 2020*). Translation of these transcripts is supported by our ribosome profiling results (*Figure 9D*), in vitro experiments using PRRSV reporter constructs (*Yuan et al., 2004*), and ribosome profiling and mass spectrometry studies of SARS-CoV-2-infected cells (*Wang et al., 2021*; *Finkel et al., 2021b*; *Davidson et al., 2020*; *St-Germain et al., 2020*; *Grenga et al., 2020*). However, it remains to be determined whether the resultant proteins are appropriately cleaved to generate functional nsps. We detected heteroclite sgRNAs as early as 3 hpi, consistent with the finding that they are packaged into PRRSV virions (*Yuan et al., 2000*; *Yuan et al., 2004*), and our results suggest they are present throughout infection. If they do produce functional proteins, this could serve as a mechanism to increase the levels of nsp1α and nsp1β, generally considered the most potent innate immune suppressors encoded by PRRSV (*Han et al., 2017*; *Ke et al., 2019*; *Ke et al., 2018*; *Han et al., 2013*; *Han et al., 2014*; *Kim et al., 2010*; *Chen et al., 2016*; *Song et al., 2010*), from early timepoints onwards to evade immune activity.

## Conclusion

This work is the first application of ribosome profiling to an arterivirus and has revealed a complex complement of PRRSV gene expression strategies, several of which permit stoichiometric modulation of the polyprotein proteins. At the level of translational control, the nsp2 −2 PRF site was found to be a rare example of temporally regulated frameshifting, while the finding that −1 PRF efficiency at the ORF1a/1b overlap also increased over time challenges the paradigm that RNA structure-directed frameshift sites operate at a fixed efficiency. At the transcriptional level, numerous non-canonical transcripts were characterised, some of which bear similarities to those found in other nidoviruses. Among these transcripts, the ORF1b sgRNAs and the heteroclite sgRNAs encode portions of the polyprotein and may provide an additional method of regulating the stoichiometry of its components. Further, some ORF1b sgRNAs likely facilitate the surprisingly high levels of translation we observed for several novel ORFs overlapping ORF1b. Although there is no evidence that these overlapping ORFs produce functional proteins, the lability in the translatome that is afforded by the heterogeneous transcriptome potentially paves the way for similar ORFs to gain functions and become fixed in the viral population. Further expanding the PRRSV translatome, a short but highly translated uORF was discovered in the NA PRRSV 5′ UTR, the presence of which is highly conserved. This presents another opportunity for regulation of viral translation, potentially allowing adaptation in response to infection-induced cellular stress. This is the most comprehensive analysis of PRRSV gene expression to date and presents new paradigms for understanding arterivirus gene expression and the wider field of PRF, with potential ramifications for a range of viruses.

# Materials and methods

## Key resources table

| Reagent type (species) or resource | Designation | Source or reference | Identifiers | Additional information |
|---|---|---|---|---|
| Strain, strain background (*PRRSV-2*) | WT NA PRRSV | PMID:23761406 | SD95-21; GenBank:KC469618.1 | |
| Strain, strain background (*PRRSV-2*) | KO2 NA PRRSV | PMID:23043113 | SD95-21 KO2 | |
| Strain, strain background (*PRRSV-1*) | EU PRRSV | This paper | BH1; GenBank:OK635576 | Based on Porcilis vaccine strain (MDS Animal Health), passaged in tissue culture |
| Cell line (*Chlorocebus sabaeus*) | MA-104 | ECACC | RRID:CVCL_3845; ECACC 85102918 | |
| Cell line (*C. sabaeus*) | MARC-145 | ATCC | RRID:CVCL_4540; CRL-12231 | |
| Antibody | Anti-NA-nsp1β (mouse monoclonal) | PMID:24825891 | | (1/1000) |
| Antibody | Anti-EU-nsp1β (mouse monoclonal) | PMID:22258855 | | (1/500) |
| Commercial assay or kit | Ribo-Zero Gold rRNA removal kit (human/mouse/rat) | Illumina | | This kit is now discontinued, replaced by Illumina Stranded Total RNA Prep with Ribo-Zero Plus kit |
| Chemical compound, drug | Cycloheximide; CHX | Merck | 239763 | |
| Other | RNase I | Ambion | AM2294 | |
| Software, algorithm | bowtie; bowtie1 | PMID:19261174 | Version 1.2.3; RRID:SCR_005476 | Aligner for reads in core pipeline |
| Software, algorithm | STAR | PMID:23104886 | Version 2.7.3a; RRID:SCR_004463 | Aligner for junction-spanning read analysis, PRICE, and differential gene expression |
| Software, algorithm | PRICE | PMID:29529017 | Version 1.0.3b | Detection of novel ORFs and quantification of translation |
| Software, algorithm | htseq-count | PMID:25260700 | Version 0.13.5 | Counting reads attributed to each gene for differential gene expression |
| Software, algorithm | DESeq2 | PMID:25516281 | Version 1.30.1; RRID:SCR_015687 | Performing differential gene expression analysis for RNASeq, and quality filtering and normalisation for differential TE analysis |
| software, algorithm | xtail | PMID:27041671 | Version 1.1.5 | Performing differential TE analysis |

## Cells and viruses

For EU PRRSV infections, MA-104 cells were infected with a PRRSV strain derived from the Porcilis vaccine strain (MSD Animal Health; GenBank accession OK635576.1). Confluent 6 cm$^2$ dishes of MA-104 cells were infected at a MOI within the range 1–3. At 0 (mock), 2, 4, 8, 12, 24, and 28 hpi (infected and mock), cycloheximide (Sigma-Aldrich) was added to the medium (final concentration 100 µg/ml) and incubated for 2 min. Cells were rinsed with 5 ml of ice-cold PBS, placed on ice, and 400 µl lysis buffer (20 mM Tris pH 7.5, 150 mM NaCl, 5 mM MgCl$_2$, 1 mM DTT, 1% Triton X-100, 100 µg/ml CHX, and 25 U/ml TURBO DNase [Life Technologies]) added drop-wise. Cells were scraped off the plate and triturated 10 times with a 26G needle, before cell debris was removed by centrifugation (13,000 × $g$, 4°C, 20 min) and the supernatant harvested and stored at −70°C.

For NA PRRSV infections, confluent 10 cm$^2$ dishes of MARC-145 cells were infected with NA PRRSV isolate SD95-21 (GenBank accession KC469618.1), a previously characterised mutant (KO2) based on this background (*Fang et al., 2012*; *Li et al., 2014*; *Li et al., 2018*), or mock-infected. At the time of harvesting for the timecourse samples, cells were washed with warm PBS and snap-frozen in liquid nitrogen. For the CHX pre-treated library (CHX-9hpi-WT), an additional CHX pre-treatment step (100 µg/ml, 2 min) was included directly prior to harvesting, and cells were washed with ice-cold PBS containing 100 µg/ml CHX before snap-freezing. Snap-frozen dishes were transferred to dry ice and 400 µl lysis buffer added. The dish was transferred to ice to defrost, and cells were scraped and processed as described above. For both NA and EU PRRSV, all replicates described in the 'Results' are biological replicates, with one dish of cells representing one replicate. Cells were verified as mycoplasma-free by PCR (Universal Mycoplasma Detection Kit, ATCC) and deep sequencing.

## Western blotting

Samples were resolved by 15% SDS-PAGE (NA PRRSV) or 10–20% Tris-tricine (EU PRRSV; Invitrogen) and transferred to 0.2 µm nitrocellulose membranes. Membranes were blocked with 5% Marvel milk powder (milk) dissolved in PBS (1 hr, 25°C). Primary antibodies were diluted in 5% milk, PBS, 0.1% Tween-20, and incubated with membranes (1 hr, 25°C). After three washes in PBS, 0.1% Tween-20, membranes were incubated with IRDye fluorescent antibodies in 5% milk, PBS, 0.1% Tween-20 (1 hr, 25°C). Membranes were washed three times in PBS, 0.1% Tween-20, and rinsed in PBS prior to fluorescent imaging with an Odyssey CLx platform (LI-COR). Antibodies used were mouse monoclonal anti-NA-nsp1β (1/1000) and anti-EU-nsp1β (1/500), as previously described (*Li et al., 2014*; *Li et al., 2012*), mouse IgM monoclonal anti-GAPDH (1/20,000, clone G8795, Sigma-Aldrich), goat monoclonal anti-tubulin (1/1000), goat anti-mouse IRDye 800 CW (1/10,000, LI-COR), rat anti-goat IRDye 800 CW (1/10,000, LI-COR), and goat anti-mouse IgM (µ chain specific) IRDye 680RD (1/10,000, LI-COR).

## Ribosome profiling library preparation

Ribosome profiling and RNASeq libraries were prepared as described in *Irigoyen et al., 2016* using a protocol derived from *Ingolia et al., 2009*, *Ingolia et al., 2012*, *Guo et al., 2010*, and *Chung et al., 2015*, with the following modifications. For RiboSeq libraries, RNase I (Ambion) was added to final concentration 2.7, 3, 5, 4.17, 3.33, 2.5, or 2.5 U/µl for 3 hpi, 6 hpi, and 9 hpi replicate 1, 9 hpi replicates 2–4, 12 hpi, CHX-9hpi-WT, and EU libraries, respectively, and SUPERase-In RNase inhibitor (Invitrogen) scaled accordingly (amounts adjusted to improve phasing). The range of fragment sizes selected during the first polyacrylamide gel purification was 25–34 nt (9 hpi replicate 1 RiboSeq and RNASeq libraries, all EU libraries, CHX-9hpi-WT), 19–80 nt (9 hpi replicate 2 RiboSeq libraries), 19–34 nt (all other RiboSeq libraries), or ~50 nt (all other RNASeq libraries), and subsequent gel slices were adjusted accordingly. The greater length range of 9 hpi replicate 2 RiboSeq libraries was selected to investigate the potential presence of long fragments protected by disomes or stable structures/complexes at frameshift sites (analyses not included herein due to inconclusive results). Ribosomal RNA depletion was carried out solely using the Ribo-Zero Gold Human/Mouse/Rat kit (Illumina). Adapter sequences were based on the TruSeq small RNA sequences (Illumina) and, for most libraries, additional randomised bases were added to the end destined for ligation: 7 randomised bases on both adapters for all NA PRRSV libraries, no randomised bases on EU PRRSV replicate 1, and 14 randomised bases on the 3′ adapter for EU PRRSV replicate 2. Randomised bases allow identification of PCR duplicates and are expected to reduce technical biases. Libraries were deep sequenced as a

single-end run on the NextSeq 500 platform (Illumina) or a paired-end run using a Mid Output v2 kit (150 cycles: 2 × 75) for 9 hpi replicate 2 RiboSeq libraries.

## Core analysis pipeline

For single-end libraries, fastx_clipper (FASTX Toolkit version 0.0.14, parameters: '-Q33 -l 33 -c -n -v') was used to trim the universal adapter sequence from reads and to discard adapter-only reads, non-clipped reads, and 'too-short' reads (inferred original fragment lengths shorter than the minimum intended length experimentally purified – see library preparation description for lengths). Adapter dimers were counted using the grep command line utility and added to the adapter-only read count. For paired-end libraries, adapter trimming, read pair merging, and removal of adapter-only reads were carried out using LeeHom (*Renaud et al., 2014*) (v.1.1.5) with the --ancient-dna option specified (as the expected fragment lengths of such DNA are in the same range as ours). Pairs of reads which LeeHom was unable to merge were put in the 'non-clipped' category for the purposes of library composition analysis, and 'too-short' reads were removed using awk. For libraries prepared using adapters with randomised bases, PCR duplicates were removed using awk, and seqtk (version 1.3) was used to trim the randomised bases from the reads. Bowtie1 (version 1.2.3) (*Langmead et al., 2009*) was used to map reads to host and viral genomes using parameters '-v n_mismatches --best', where n_mismatches was 1 for RiboSeq and 2 for RNASeq libraries. Reads were mapped to each of the following databases in order, and only reads that failed to align were mapped to the next database: ribosomal RNA (rRNA), virus RNA (vRNA), mRNA, non-coding RNA (ncRNA), genomic DNA (gDNA). Viral genome sequences were verified by de novo genome assembly using Trinity (version 2.9.1). The rRNA database comprised the following GenBank accessions: NR_003287.2, NR_023379.1, NR_003285.2, NR_003286.2, AY603036.1, AF420058.1, AF420040.1, AY633510.1, AF352382.1, L35185.1, DQ983926.1, KJ193255.1, M30951.1, M30950.1, M30952.1, KJ193272.1, KJ193259.1, KJ193258.1, KJ193256.1, KJ193255.1, KJ193045.1, KJ193042.1, KJ193044.1, KJ193041.1, KJ193019.1, KJ193018.1, KJ193017.1, and AF420040.1. The mRNA database was compiled from the available *C. sabaeus* RefSeq mRNAs after removing transcripts with annotated changes in frame. The ncRNA database was Chlorocebus_sabaeus.ChlSab1.1.ncrna.fa, and the gDNA database was the ChlSab1.1 genome assembly, both from Ensembl. The position of vRNA within the database mapping order was altered to confirm that significant numbers of viral reads were not erroneously mapping to host databases or vice versa. All analyses were carried out using reads mapped by bowtie as described above, except for running PRICE, analyses using junction-spanning reads, or host differential gene expression.

Quality control plots and analyses were performed as described in *Irigoyen et al., 2016*, modified for the timecourse libraries to account for the longer RNASeq reads, so that a 3′ UTR of at least 90 nt (as opposed to 60 nt) was required for inclusion of transcripts in the metagene analysis of 5′ end positions relative to start and stop codons (*Figure 1—figure supplement 1*). For quality control analyses of read length and phasing, reads mapping to ORF1ab (all phases, nsp2TF excluded) were used for the virus versus host analyses (e.g. *Figure 1C*, *Figure 1—figure supplement 2*), while for analyses of specific regions (e.g. *Figure 1—figure supplements 3 and 4*), overlapping regions of ORFs were permitted for the length distribution but not phasing analyses. For these phasing plots, phase 0 was designated independently for each region, relative to the first nucleotide of the ORF in that region. For the negative-sense read analysis, reads mapping to anywhere on the viral genome were used, and phase was determined using the 5′ end of the read (the 5′ end of the reverse complement reported by bowtie plus the read length). The coordinates of the regions of the viral genome used for all analyses are given in *Supplementary file 2*. All plots and analyses use RiboSeq read lengths identified as having minimal RNP contamination (*Figure 1—figure supplement 4*, *Figure 6—figure supplement 3F*) unless otherwise specified, and all plots and analyses using RNASeq reads use all read lengths.

## Plots of read distributions on viral genomes

Read densities were plotted at the inferred ribosomal P site position, obtained by applying a static +12 nt offset to the 5′ end coordinate of the read (applied to both RiboSeq and RNASeq for comparability, although ribosomal P site is not relevant to RNASeq reads). For *Figure 10—figure supplement 7*, species-specific tRNA adaptation index (stAI) values for *Macaca mulatta* (in the absence of *C. sabaeus* data) were obtained from STADIUM (*Yoon et al., 2018*) on 4 October 2020,

and a heatmap constructed using the minimum and maximum values for *M. mulatta* codons as the most extreme colours available in the gradient. For plots of the percentage of reads in each phase across ORF1a, positive-sense RiboSeq reads were separated according to phase and a 183-codon running mean filter applied to avoid any instances of zero across ORF1a (excluding the halfwindow at each end) after sliding window application. From this, the percentage of reads in each phase at each codon in-frame with ORF1a was calculated.

## Significance testing for proportion of host RPFs which are short

RiboSeq libraries were grouped into early (3 and 6 hpi) and late (9 and 12 hpi) timepoints to provide enough replicates in each group to perform a two-tailed *t*-test (WT and KO2 were treated as equivalent). Positive-sense RPFs mapping to host mRNA were used, and short reads were defined as 19–24 nt long, with the denominator formed by 19–34 nt long reads. The early timepoint group was used as a control, for which there was no significant difference in the percentage of short RPFs in infected cells compared to mock-infected cells (p=0.52).

## Junction-spanning read analysis for novel transcript discovery

Reads which did not map to any of the host or viral databases (rRNA, vRNA, mRNA, ncRNA, or gDNA) in the core pipeline (described above) were used as input for mapping using STAR (*Dobin et al., 2013*), version 2.7.3a. Mapping parameters were selected based on those suggested in *Kim et al., 2020* to switch off penalties for non-canonical splice junctions:

```
'--runMode alignReads --outSAMtype BAM SortedByCoordinate
--outFilterType BySJout --outFilterMultimapNmax 2 --alignSJoverhangMin
12 --outSJfilterOverhangMin 12 12 12 12 --outSJfilterCountUniqueMin 1 1
1 1 --outSJfilterCountTotalMin 1 1 1 1 --outSJfilterDistToOtherSJmin 0
0 0 0 --outFilterMismatchNmax 2 --scoreGapNoncan -4 --scoreGapATAC -4
--chimOutType Junctions --chimScoreJunctionNonGTAG 0 --alignEndsType
EndToEnd --alignSJstitchMismatchNmax -1 -1 -1 -1 --alignIntronMin 20
--alignIntronMax 1000000 --outSAMattributes NH HI AS nM jM jI'
```

First, junctions were processed within each library. To avoid junction clusters becoming inflated, late timepoint libraries (8 hpi onwards) were filtered to remove junctions with fewer than four supporting reads. Reads were split into two categories, TRS-spanning and non-TRS-spanning, according to whether the donor site of the junction overlapped the leader TRS (genomic coordinates in *Supplementary file 2*). Junctions were clustered so that all junctions within a cluster had acceptor coordinates within 7 (for TRS-spanning junctions) or 2 (for non-TRS-spanning junctions) nucleotides of at least one other junction in the cluster, with the same requirement applied to donor coordinates. This was to group highly similar junctions together and account for the fact that the precise location of a junction is ambiguous in cases where there is similarity between the donor and acceptor sites (such as between the 6-nt leader and body TRSs). The junctions within each cluster were merged, with donor and acceptor sites defined as the midpoints of the ranges of coordinates observed in the cluster, and the number of reads supporting the merged junction defined as the sum of the supporting read counts for all the input junctions in the cluster.

Then, junctions were filtered to keep only those present in multiple libraries and merged to generate one dataset per timepoint. Merged junctions from individual libraries were filtered so that only junctions which were present in more than one replicate (considering WT and KO2 as one group) passed the filter. Junctions were defined as matching if the ranges of the donor and acceptor coordinates for the junction in one library overlap with those of a junction in a second library. Matching junctions from all replicates were merged as described above to make the final merged junction. For the NA PRRSV M junction, there is a stretch of six bases that is identical upstream of the leader TRS and the body TRS, leading to separation of the two alternative junction position assignments by a distance greater than the seven bases required to combine TRS-spanning junctions into clusters. The two junction clusters that are assigned either side of this stretch of identical bases were specifically selected and merged at this stage. To ensure this merging strategy did not lead to clusters spanning overly

wide regions, widths of merged junction donor and acceptor sites were assessed, and the mean and median junction width for all analyses was found to be <3 nt (maximum width 17 nt, for the NA PRRSV M junction). TRS-spanning junctions were designated as 'known' junctions if they were the major junction responsible for one of the known canonical sgRNAs of PRRSV. Non-TRS-spanning junctions were filtered according to whether they represent local (≤2000 nt) or distant (>2000 nt) deletions.

The proportion of junction-spanning reads at donor and acceptor sites was calculated as junction-spanning/(junction-spanning + continuously aligned to reference genome). The number of non-junction-spanning reads at the junction site was defined as the number of bowtie-aligned reads (from the core pipeline) spanning at least the region 12 nt either side of the midpoint position of the donor or acceptor site (note that for sgRNA acceptor sites the denominator will include not only gRNA but also sgRNAs with body TRSs upstream). For all TRS-spanning junctions, the donor midpoint was set according to the known leader TRS sequence (genomic coordinate 188 for NA PRRSV and 219 for EU PRRSV).

## Scaling ORF1b sgRNA transcript abundance for presentation alongside RiboSeq density plots

To enable easier comparison of ORF1b sgRNA transcript abundance with changes in ribosome profile in RiboSeq density plots (*Figures 6B, 9A and B*, *Figure 6—figure supplements 1 and 2B*, *Figure 9—figure supplement 1*), the usage frequency of the indicated body TRS for each ORF1b sgRNA was compared to that of the major body TRS for the canonical sgRNA with the fewest junction-spanning reads (GP4 for NA PRRSV and GP3 for EU PRRSV). The total number of reads, from all 12 hpi libraries combined (NA PRRSV) or all libraries combined (EU PRRSV), spanning the ORF1b sgRNA body TRS was divided by the number spanning the canonical sgRNA body TRS to give the relative abundance of the non-canonical transcript compared to the canonical transcript (denoted as 'abund.' on plots).

## Detection of novel viral ORFs

Novel ORF discovery was performed using PRICE (*Erhard et al., 2018*) (version 1.0.3b). A custom gtf file was made for each virus, with only the gRNA transcript and ORF1b CDS annotated. Other known viral ORFs were not annotated and served as positive controls. The custom viral gtf files were each individually concatenated with the host gtf file (ChlSab1.1.101, downloaded from Ensembl) to make the input of known ORFs (treating the viral genome as an additional host chromosome). The reference fasta files for the host and viruses were similarly concatenated to make the input reference sequences. RiboSeq reads were mapped to these combined references using STAR (*Dobin et al., 2013*), version 2.7.3a, with parameters as described for novel transcript discovery but with the following changes: '--outFilterType Normal --outFilterMultimapNmax 10 --outFilterMismatchNmax 1 --chimOutType WithinBAM --outSAMattributes MD NH HI AS nM jM jl'.

The STAR alignments were used as input for PRICE, and p-values were corrected for multiple testing using the Benjamini–Hochberg method before filtering results to select significant viral ORFs. For noCHX NA PRRSV libraries, read lengths with minimal RNP contamination were used, while for EU PRRSV libraries all read lengths were used. The CHX NA PRRSV library was not used for PRICE as CHX pre-treatment can artefactually increase uORF translation (*Gerashchenko and Gladyshev, 2014*).

## Analysis of sequence conservation

Sequences were selected for inclusion in the alignment based on a requirement for ≥50% amino acid identity (across all ORFs excluding overlapping regions), ≥70% nucleotide identity, and ≥95% coverage compared with the prototype NA (NC_001961) or EU (NC_043487) PRRSV reference genomes, resulting in 661 and 120 sequences (downloaded on 29 August 2017 and 3 February 2021), respectively. For analyses of NA PRRSV genomes 'representative of NA PRRSV diversity', the NA PRRSV sequences were clustered using CD-HIT (*Li and Godzik, 2006*) (version 4.8.1) based on the whole genome and with a nucleotide similarity threshold of 95% (all other parameters set to default), and one representative sequence from each cluster was selected to make a sequence alignment of 137 sequences. For the uORF amino acid conservation plot, the nucleotide sequence of the CDS was extracted from a multiple sequence alignment and frame 0 was translated, with a 28-codon extension in KY348852 omitted from the plot for space considerations. Logo plots and mini-alignment plots were generated using CIAlign (*Tumescheit et al., 2022*) and, for the uORF analyses, genome

sequences which began partway through the ORF were excluded, as was KT257963 which has a likely sequencing artefact in the 5' UTR. For analysis of uORF start and stop codon conservation, sequences were filtered to take only those spanning the entire feature of interest without gaps, leaving 564 and 598 sequences, respectively, as input for the logo plots in *Figure 6F and G*. Synonymous site conservation was analysed, for the representative NA PRRSV sequences or for all EU sequences, using SYNPLOT2 (*Firth, 2014*) and p-values plotted after application of a 25-codon running mean filter.

## Transcript abundance, total translation, and translation efficiency analyses

For the main analysis, RiboSeq RPKM values were calculated using the read counts and ORF 'Locations' from the PRICE output (*Figure 6—source data 1*) using the same library size normalisation factors as the core pipeline (where positive-sense virus- and host-mRNA-mapping reads from the bowtie output are the denominator). Each ORF was paired with the transcript(s) most likely to facilitate its expression (see schematic in *Figure 7* and junction coordinates in *Supplementary file 2*, *Figure 4—source data 3* and *Figure 5—source data 1–2*). For some ORFs (NA PRRSV nsp10-iORF2, nsp11-iORF3, GP3-iORF, and GP4-iORF), this included transcripts which are expected to produce slightly N-terminally truncated ORFs compared to the PRICE designation. ORFs overlapping ORF1ab for which there were no novel transcripts expected to facilitate expression were paired with gRNA. All ORF1b sgRNAs, defined as sgRNAs with body TRSs within ORF1b and ≥50 or ≥ 10 junction-spanning reads at 12 hpi (NA PRRSV) or 8 hpi (EU PRRSV), were included in the transcript abundance analysis regardless of whether they are expected to result in expression of a novel ORF.

To estimate transcript abundance, reads aligned to the viral genome by STAR (see junction-spanning read analysis pipeline) were normalised by library size using the same library size normalisation factors as the core pipeline. In cases where multiple body TRSs are expected to give rise to two different forms of a transcript that express the same ORF(s), these were treated as a single transcript for the purposes of this analysis, and read counts for all junctions were combined. Abundance of the gRNA transcript was defined as the number of bowtie-aligned reads which span 12 nt either side of the midpoint of the leader TRS (genomic coordinate 188 for NA PRRSV and 291 for EU PRRSV). This is analogous to the 12-nt overhang required either side of a junction to qualify for mapping by STAR; however, these reads are not junction-spanning and map specifically to gRNA (and a small proportion of non-canonical transcripts such as heteroclite sgRNAs). Leader abundance was defined as the total number of reads for all other transcripts in the analysis combined, as the leader is present on all sgRNA species and the gRNA. TE was calculated by dividing RiboSeq RPKM values by RNASeq junction-spanning read RPM values, excluding conditions where the denominator was zero.

For plots with logarithmic axes, datapoints with a value of zero were excluded from the plot, but not from mean calculations. WT and KO2 were treated as equivalent unless specified. For libraries with shorter read lengths (EU libraries and NA 9 hpi replicate 1 libraries) junction-spanning read counts are lower (and also subject to greater inaccuracies as a result of less dilution of possible read start- and end-point-specific ligation biases) due to the requirement for a 12-nt overhang either side of the junction effectively representing a much larger proportion of the total read length. As such, these libraries are not directly comparable to the remaining NA PRRSV libraries and were plotted separately and not included in NA PRRSV mean calculations.

To calculate the percentage of the viral transcriptome represented by each transcript (*Figure 8B*), transcript abundances (estimated from junction-spanning read RPM values) within each library were converted to a percentage of the total RPM of all transcripts plotted for that library. Mean percentages for each group were calculated, treating WT and KO2 as equivalent. A similar process was used to calculate the percentage of the viral translatome represented by each ORF (*Figure 8C*), converting the RPKM value of each ORF (calculated from the PRICE output) to a percentage of the total sum of RPKM values of all ORFs plotted.

For the estimation of translation of different regions of ORF1b, sections were designated as the regions between the downstream-most body TRS of one ORF1b sgRNA and the upstream-most body TRS of the next (all region coordinates given in *Supplementary file 2*). Bowtie-aligned RiboSeq reads (from the core pipeline) which mapped in-phase with ORF1b in the designated regions were counted using only read lengths with minimal RNP contamination (NA PRRSV) or good phasing (EU PRRSV). Total read counts were normalised by library size and region length to give RPKM. The same process

was applied to the region of ORF1a between the major heteroclite junction and the nsp2 PRF site for comparison, counting reads mapping in-phase with ORF1a.

For investigation of gene expression in different regions of ORF1a, transcript abundance for the heteroclite sgRNAs was calculated by 'decumulation': subtracting the gRNA RNASeq RPKM (measured in the region between the major [S-2] heteroclite sgRNA junction and the first ORF1b sgRNA body TRS) from the RPKM in the region between the leader TRS and the major heteroclite sgRNA junction ('heteroclite'). This provides an averaged result for all heteroclite sgRNAs, although it does not take into account the reduced transcript length for the minor heteroclite sgRNAs compared to S-2. RiboSeq read density (calculated as described above for sections of ORF1b) was similarly decumulated to assess ribosome occupancy of the heteroclite sgRNAs by subtracting the RiboSeq RPKM in the gRNA-only region (between the major heteroclite junction and the nsp2 PRF site) from that in the heteroclite region (all coordinates in *Supplementary file 2*). For TE of the heteroclite region, the decumulated RiboSeq RPKM was divided by the decumulated RNASeq RPKM, whereas for the gRNA TE the raw RPKM values were used (as no decumulation is required). These TE values are intended for internal comparison with other TE values in *Figure 9D* – due to the different denominators (RPKM versus RPM), they are not comparable to TE values presented in *Figure 8—figure supplement 3*.

## nsp2 site PRF efficiency calculations based on phasing

The proportion of reads in each phase in the upstream and transframe regions (coordinates in *Supplementary file 2*) was calculated, where in both regions phase is taken relative to the ORF1a reading frame. It was assumed that all ribosomes in the upstream region were translating in the 0 frame, and the phase distribution in this region was used to estimate what proportion of 0-frame ribosomes generate reads attributed to the 0 phase ($upstream_0$) and the −2 phase ($upstream_{-2}$). This was extrapolated to determine what proportion of reads are expected to be in the −2 phase in the transframe region ($transframe_{-2}$) in the absence of frameshifting (which is expected to be the same as in the upstream region). A proportion (FS_proportion) of ribosomes undergoing −2 PRF is expected to mean that, between the upstream and downstream region, FS_proportion of phase 0 reads change from the 0 to the −2 phase and FS_proportion of −2 phase reads move to the −1 phase (leaving 1 − FS_proportion in the −2 phase). These concepts were combined to make the equation

$$transframe_{-2} = (FS\_proportion \times upstream_0) + (1 − FS\_proportion) \times upstream_{-2}$$

This was rearranged to calculate percentage frameshift efficiency (which is FS_proportion expressed as a percentage):

$$FS\_efficiency = 100 \times (transframe_{-2} − upstream_{-2})/(upstream_0 − upstream_{-2})$$

This phasing-based method of calculating frameshift efficiency should theoretically be unaffected by RNP contamination, provided the RNP footprints are equally distributed between the three phases. Let R be the proportion of total reads that are RNPs, and let $P_0$ and $P_{-2}$ be the proportion of total RNPs that are attributed to the 0 and −2 phases, respectively. The phasing of reads originating from RNPs is not expected to change due to frameshifting. Therefore, the equation for calculating the fraction of reads that change from the 0 to −2 phase becomes

$$FS\_proportion \times (upstream_0 − RP_0)$$

and the equation for calculating the fraction of reads that remain in the −2 phase becomes

$$RP_{-2} + [(1 − FS\_proportion) \times (upstream_{-2} − RP_{-2})]$$

Combining these makes the equation

$$transframe_{-2} = \quad FS\_proportion \times [upstream_0 − RP_0] + RP_{-2}$$
$$+[(1 − FS\_proportion) \times (upstream_{-2} − R_{-2})]$$

This rearranges to

$$FS\_proportion = (upstream_{-2} − transframe_{-2})/(RP_0 − upstream_0 − RP_{-2} + upstream_{-2})$$

If $P_0 = P_{-2}$ (e.g. if RNPs are equally distributed between all three phases), then this causes both terms involving R to cancel out of the equation, meaning RNPs would not affect the result. The same would hold for any other form of uniform non-phased contamination.

For NA PRRSV libraries, read lengths identified as having minimal RNP contamination (indicated in *Figure 1—figure supplement 4*) were used, and, for EU PRRSV, read lengths with good phasing (*Figure 1—figure supplement 6D*) were used.

### Nsp2 site PRF efficiency calculations based on KO2-normalised read density

The density normalisation-based method calculates combined −2/−1 frameshift efficiency as 100 × [1−(downstream/upstream)], where downstream and upstream represent the RPKM values for the respective regions (coordinates in *Supplementary file 2*) after normalisation of WT density by the density in its KO2 counterpart to factor out differences in translation speed and/or biases introduced during library preparation. WT and KO2 libraries were paired first according to processing batches, and within each batch (if there were multiple replicates) the WT library with the higher ratio of virus:host RiboSeq reads was paired with the KO2 library with the higher ratio. This resulted in libraries with the same replicate number being paired, except 9 hpi WT-3 was paired with KO2-4, and WT-4 with KO2-3. For RiboSeq libraries, read lengths identified as having minimal RNP contamination (indicated in *Figure 1—figure supplement 4*) were used, whereas for RNASeq-negative control libraries all read lengths were used. This method could not be applied to EU PRRSV libraries as no KO2 libraries were made.

### ORF1ab site PRF efficiency calculations

Frameshift efficiency was calculated as 100 × (downstream/upstream), where downstream and upstream represent the RPKM values for the respective regions. Mutation of this frameshift site prevents viral replication, so normalisation by a frameshift-defective mutant was not possible. KO2 and WT libraries were treated as equivalent for the calculations and statistical tests. For NA PRRSV RiboSeq libraries, read lengths identified as having minimal RNP contamination (indicated in *Figure 1—figure supplement 4*) were used, and for EU PRRSV RiboSeq libraries read lengths with good phasing (*Figure 1—figure supplement 6D*) were used. For all RNASeq negative control libraries, all read lengths were used. A two-tailed Mann–Whitney *U* test was employed to assess the statistical significance of differences between groups of observed values.

### Bootstrap resampling for phasing-based nsp2 −2 PRF efficiency calculations

100,000 randomised resamplings of codons in each respective region were performed. Each WT library was paired with its corresponding KO2 library (as described for the KO2-normalised strategy above), with matched codons selected for the two libraries in each resampling, and reads with 5′ ends mapping to these codons were used as input for the PRF efficiency calculations. Calculation of nsp2 −2 PRF efficiency for each resampling was performed using the phasing-based method as described above, with the results of individual libraries recorded, and then the mean of all libraries in each group calculated. Regions and bounding coordinates used were as described above, with an additional 147-codon downstream region (the same length as the region of nsp2TF used) added as a negative control (coordinates in *Supplementary file 2*). For all resamplings, *n* codons were sampled with replacement, where *n* is the total number of codons in the region undergoing resampling. Bootstrap resamplings were used to empirically determine p-values. Confidence intervals for each bootstrap distribution were calculated using the bias-corrected accelerated (BCa) method, implemented through the R package coxed (version 0.3.3). This was performed for 95, 99.5, and 99.95% confidence intervals, and pairs of groups were considered as significantly different with p<0.05, 0.005, or 0.005, respectively, if the mean of the 'group 1' bootstrap distribution was not within the confidence intervals of 'group 2' and vice versa.

### Host differential gene expression

After basic processing and removal of rRNA- and vRNA-mapping reads using bowtie as described in the core analysis pipeline, remaining reads were aligned to the host genome (fasta and gtf from genome

assembly ChlSab1.1) using STAR (*Dobin et al., 2013*) (version 2.7.3a) with the following parameters: '--runMode alignReads --outSAMtype BAM SortedByCoordinate --outFilterMismatchNmax n_mismatches --outFilterIntronMotifs RemoveNoncanonicalUnannotated --outMultimapperOrder Random' (where n_mismatches was 1 for RiboSeq libraries and 2 for RNASeq libraries). Reads were tabulated using htseq-count (*Anders et al., 2015*) (version 0.13.5), with parameters '-a 0 -m union -s yes -t gene' (covering the whole gene) for the differential transcription and '-a 0 -m intersection-strict -s yes -t CDS' (covering only the CDS) for the differential TE. Genes with fewer than 10 reads between all libraries in the analysis combined were excluded, and quality control was performed according to the recommendations in the DESeq2 (*Love et al., 2014*) user guide, with all replicates deemed to be of sufficient quality. Read counts were normalised for differences in library size using DESeq2 (version 1.30.1), providing the input for differential transcription using DESeq2 (default parameters) or differential TE using xtail (*Xiao et al., 2016*) (version 1.1.5; parameters: 'normalize = FALSE'). Shrinkage was applied to the DESeq2 output using lfcShrink (parameters: 'type = 'normal''). Where necessary (i.e. for the KO2 vs. WT comparison), fdrtool was used to correct conservative p-values for differential transcription (version 1.2.16; parameters: 'statistic = 'normal''), in addition to the Benjamini–Hochberg correction for multiple testing. This correction could not be applied to the xtail results as the test statistic is not included in the xtail output. Genes were considered significantly differentially expressed if they had FDR-corrected p-value≤0.05 and $\log_2$(fold change) of magnitude >1 (for comparisons to mock) or >0.7 (for KO2 vs. WT comparison). GO terms associated with lists of significantly differentially expressed genes were retrieved and tested using DAVID (*Huang et al., 2009*) (version 6.8, functional annotation chart report, default parameters) for enrichment against a background of GO terms associated with all genes that passed the threshold for inclusion in that differential expression analysis.

## Data availability

Data are available on ArrayExpress under accession numbers E-MTAB-10621, E-MTAB-10622, and E-MTAB-10623. Code for the core pipeline and differential gene expression analyses was based on the pipeline available on GitHub at https://github.com/firth-lab/RiboSeq-Analysis, (copy archived at swh:1:rev:bd9e04d2d11a4ec27106af57aa8270db2e35b500; *Cook, 2017*).

## Acknowledgements

We thank Dr David Brown, and all members of the Brierley, Firth and Fang laboratories, for supporting discussions. For the purpose of Open Access, the author has applied a CC BY public copyright licence to any Author Accepted Manuscript version arising from this submission.

## Additional information

### Competing interests

AP Adrian Mockett: Dr A.P. Adrian Mockett is a Director of and Consultant for Cambivac Ltd. He declares no financial or non-financial competing interests regarding the manuscript. Ying Fang, Andrew E Firth: is listed as an inventor for the following patent. Patent Title: Novel arterivirus protein and expression mechanisms; No: WO2014015116A2. The other authors declare that no competing interests exist.

### Funding

| Funder | Grant reference number | Author |
| --- | --- | --- |
| Wellcome Trust | 203864/Z/16/Z | Georgia M Cook |
| Wellcome Trust | 102163/Z/13/Z | Lior Soday |
| Wellcome Trust | 106207/Z/14/Z | Andrew E Firth |
| Wellcome Trust | 202797/Z/16/Z | Ian Brierley |
| H2020 European Research Council | 646891 | Andrew E Firth |

| Funder | Grant reference number | Author |
| --- | --- | --- |
| National Institute of Food and Agriculture | 2015-67015-22969 | Ying Fang |

The funders had no role in study design, data collection and interpretation, or the decision to submit the work for publication.

## Author contributions

Georgia M Cook, Conceptualization, Data curation, Formal analysis, Funding acquisition, Investigation, Methodology, Resources, Software, Visualization, Writing – original draft, Writing – review and editing; Katherine Brown, Data curation, Methodology, Resources, Software, Supervision, Writing – review and editing; Pengcheng Shang, Yanhua Li, Investigation, Writing – review and editing; Lior Soday, Conceptualization, Formal analysis, Funding acquisition, Investigation, Software, Writing – review and editing; Adam M Dinan, Data curation, Resources, Software, Writing – review and editing; Charlotte Tumescheit, Formal analysis, Resources, Writing – review and editing; AP Adrian Mockett, Investigation, Resources, Writing – review and editing; Ying Fang, Conceptualization, Funding acquisition, Methodology, Resources, Writing – review and editing; Andrew E Firth, Conceptualization, Data curation, Funding acquisition, Methodology, Resources, Software, Supervision, Writing – original draft, Writing – review and editing; Ian Brierley, Conceptualization, Funding acquisition, Investigation, Methodology, Resources, Supervision, Writing – original draft, Writing – review and editing

## Author ORCIDs

Georgia M Cook http://orcid.org/0000-0003-1577-735X
Charlotte Tumescheit http://orcid.org/0000-0002-7563-5575
Andrew E Firth http://orcid.org/0000-0002-7986-9520
Ian Brierley http://orcid.org/0000-0003-3965-4370

## Decision letter and Author response

Decision letter https://doi.org/10.7554/eLife.75668.sa1
Author response https://doi.org/10.7554/eLife.75668.sa2

# Additional files

## Supplementary files

• Supplementary file 1. Composition of libraries. Number of reads assigned to each category. Reads were classified as 'too short' if their inferred original fragment length was shorter than the minimum intended length experimentally purified (25 nt for all RNASeq libraries; for RiboSeq libraries: 25 nt for European porcine reproductive and respiratory syndrome virus (EU PRRSV) libraries and North American (NA) PRRSV 9 hr post-infection (hpi) replicate 1 libraries, 19 nt for all other libraries).

• Supplementary file 2. Coordinates of regions of the North American (NA) and European porcine reproductive and respiratory syndrome virus (PRRSV) genomes used for each analysis. All coordinates denote the regions within which the 5′ end of the reads must map to qualify for inclusion in the analysis. Sheet 1: general plots and analyses (other than quantification of viral gene expression). Sheet 2: transcript abundance estimation. Sheet 3: translation level estimation.

• Transparent reporting form

## Data availability

Sequencing data are available on ArrayExpress under accession numbers E-MTAB-10621, E-MTAB-10622 and E-MTAB-10623. Code for the core pipeline and differential gene expression analyses was based on the pipeline available on Github at https://github.com/firth-lab/RiboSeq-Analysis, (copy archived at swh:1:rev:bd9e04d2d11a4ec27106af57aa8270db2e35b500). For Figure 3 (panels D-F), the raw western blots and quantification values are provided in Figure 3-source data 1-5. For Figures 4 and 5, the source data can be found in Figure 4-source data 1-3 and Figure 5-source data 1-2. The data in these tables was also used to generate Figures 7 and 8 and Figure 8-figure supplements 1 and 3. For Figure 6, the source data used to annotate the ORFs is provided in Figure 6-source data 1. The data in this table was similarly used to annotate the ORFs in Figure 6-figure supplements 1 and 2 and Figure 7. The expression data in this table was used to generate some of the plots in Figure 8 and

Figure 8-figure supplements 2 and 3. For Figure 11, the source data can be found in Figure 11-source data 1-3.

The following datasets were generated:

| Author(s) | Year | Dataset title | Dataset URL | Database and Identifier |
|---|---|---|---|---|
| Cook GM, Shang P, Li Y, Soday L, Mockett APA, Fang Y, Firth AE, Brierley I | 2021 | Ribosome profiling of MARC-145 or MA-104 cells infected with porcine reproductive and respiratory syndrome virus (PRRSV)-1 or PRRSV-2, harvested at 3-12 hpi | https://www.ebi.ac.uk/arrayexpress/experiments/E-MTAB-10621/ | ArrayExpress, E-MTAB-10621 |
| Cook GM, Shang P, Li Y, Fang Y, Firth AE, Brierley I | 2021 | Paired-end ribosome profiling of MARC-145 cells infected with porcine reproductive and respiratory syndrome virus (PRRSV)-2, harvested at 9 hpi | https://www.ebi.ac.uk/arrayexpress/experiments/E-MTAB-10622/ | ArrayExpress, E-MTAB-10622 |
| Cook GM, Shang P, Li Y, Soday L, Mockett APA, Fang Y, Firth AE, Brierley I | 2021 | RNASeq of MARC-145 or MA-104 cells infected with porcine reproductive and respiratory syndrome virus (PRRSV)-1 or PRRSV-2, harvested at 3-12 hpi | https://www.ebi.ac.uk/arrayexpress/experiments/E-MTAB-10623/ | ArrayExpress, E-MTAB-10623 |

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
