## [Editor Report]

The article presents a first example of a detailed quantitative study of host and PRRSV gene expression over the time course of infection. The study not only identifies multiple non-canonical mechanisms of PRRSV gene expression regulation, but also shows that the frameshifting efficiency at the canonical ORF1ab frameshifting site changes with time. This finding provides new insights into the viral gene expression and into the regulation of programmed ribosome frameshifting, which has important implications for understanding viral biology and for developing antiviral drugs.

---

## [Decision Letter]

**Decision letter after peer review:**

Thank you for submitting your article "Ribosome profiling of porcine reproductive and respiratory syndrome virus reveals novel features of viral gene expression" for consideration by *eLife*. Your article has been reviewed by 3 peer reviewers, one of whom is a member of our Board of Reviewing Editors, and the evaluation has been overseen by a Reviewing Editor and Miles Davenport as the Senior Editor. The reviewers have opted to remain anonymous.

Essential revisions:

Overall, the study is well designed, carefully executed and is accompanied by rigorous and exceptionally meticulous data analysis. The data generated in this work may be potentially used beyond the scope of this study and are of importance for a broad audience of virologists and those interested in gene expression. One potential weaknesses of the paper is that it is hard to get through simply due to its length and a lot of data, which makes it difficult to follow, in particular for the non-specialists. Concerning potential technical issues, the ribosome run-offs in the libraries prepared without CHX may affect the distribution of reads within ORFs, which would require some caution with the data interpretation.

Specific points:

1. The libraries obtained in the absence of CHX clearly show ribosome run-off, as seen in the metagene profiles (Figure S1) and in the translation profiles of most viral ORFs, especially at the late time points (Figure 2B and S8). This raises the question of whether the reported distribution of reads within ORFs is biased and calls for caution when comparing footprint densities between different regions of the same ORF. This may have implications for several analyses in the paper, including the frameshift efficiency calculations of ORF1ab, and translation from ORF1b body TRS sgRNAs and from heteroclite sgRNAs. This issue should be clarified and the conclusions toned down where appropriate.

2. The non-canonical sgRNA would require extremely high translation efficiency to be detected above translation from the gRNA. This is surprising and the authors should provide more explanations to clarify this point.

3. To test the potential mechanism of changing PRF efficiency, can the authors correlate the PRF efficiency at different time points with the expression of the tRNA isoacceptors (or isodecoders) that are required to read the slippery sequence?

4. The paper is written in a dense technical language that limits its readability and is likely to reduce the impact of the study. As not all findings are equally important, the authors should revise the manuscript for clarity, preferably by shortening the main text and leaving less important observations in the Supplements. In addition, the following specific points should be addressed:

a. Figure 12, which should summarize the findings of the paper, is very difficult to understand. It should be revised for clarity and explained in Figure legend. All elements of the scheme should be clearly labeled.

b. All Figure legends should be shortened by moving non-essential parts into Methods.

c. The Discussion should be better structured, e.g. by introducing subheadings.

d. The figures are not mentioned in the text in order (e.g. Figure 9 may be mentioned before Figure 7). Since there are many figures, this is confusing, and changing the order of the figures or the text will help clarity. In addition, text links to the figure are missing in several places throughout the manuscript (for example, line 312 Figure 5B, line 380 RNA read length distribution).

e. Whenever conservation data regarding a specific nucleotide is used to claim it is conserved, it will be useful to present background conservation to demonstrate whether the position is conserved more than expected.

f. Two viral species are analyzed in the paper, but their similarities and differences could be further discussed in the results or in the introduction.

g. In several cases, comparisons between data presented on two separate graphs are performed. In order to allow the reader to properly compare, for example, RNA levels of viral transcripts at different time points, it would be helpful to present them together on the same plot.

h. Multiple sequence alignment in Figure 6E are not very useful, as the color code for amino acids is not assigned; at the minimum, clarify the color code. In Figure 6D, the nucleotide sequence is somewhat sandwiched and is hard to read. The issue is easily solvable by giving it more space on top and the bottom. Sequence logos in Figures6C, F, G and Figure 7 contain text (on axes) that is of unintelligible size. If it cannot be enlarged, it could be simply removed since the logos are easily interpretable without it.

i. Please add the canonical body TRS sequence for comparison in Figure 7.

j. The western blot in Figure 3F is missing a loading control.

k. Figure S23B, the green 6 hpi lines seem to be colored blue.

l. In Figure S23B and C, the read length distribution in the specific position should be compared to the overall library read length distribution.

m. Which infected time points are analyzed in Figure 11? Showing a time course of the genes of interest may be useful.

n. From the analysis of RNA and TE in Figure 11, it is concluded that translation of transcriptionally upregulated genes does not go up. The footprint data can be used to directly show the change in translation, and even an increase in translation that is lower than expected from the increase in RNA levels can point to translation inhibition.

o. Line 911, no AUG codons were found in the 5'UTR of these viruses, but translation may initiate from near-cognate start codons.

p. I think References 49,117 and 128 do not support the claim that frameshift efficiency changes with time of infection.

*Reviewer #1 (Recommendations for the authors):*

My main recommendation is to re-write the paper to remove the overly technical language, identify the most important points and group the data into essential (which should be explained for a non-specialist reader) and additional (can be kept more technical for the specialists). Figure 12 should be revised for clarity and explained in Figure legend. Figure legends should be shortened by moving non-essential parts into methods. The Discussion should be better structured, e.g. by introducing sub-headings.

Question:

Can the authors correlate the PRF efficiency with the expression of the tRNA isoacceptors (or isodecoders) that are required to read the slippery sequence?

*Reviewer #2 (Recommendations for the authors):*

Overall, the manuscript is very well written and presented, the level of detail is unprecedented! My only concern is the design of a few figure panels which could be improved in my opinion before the final publication. Specifically, in Figure 6D, the nucleotide sequence is somewhat sandwiched and is hard to read. The issue is easily solvable by giving it more space on top and the bottom. I didn't find the multiple sequence alignment in Figure 6E to be very useful. Since we don't know what colours are assigned to what amino acids, the only information that could be obtained from this is on the degree of conservation, but I am not sure that this specific representation is particularly useful for the assessment of conservation. Sequence logos in Figures6C, F, G and Figure 7 contain text (on axes) that is of unintelligible size. If it cannot be enlarged, it could be simply removed since the logos are easily interpretable without it.

*Reviewer #3 (Recommendations for the authors):*

Most of the ribosome profiling experiments in this study were performed in the absence of CHX, which is normally used to freeze ribosomes in place and prevent their run-off toward the 3' of the CDS. The ribosome run-off in these libraries is clearly reflected in the metagene profiles presented (Figure S1) and in the translation profiles of most viral ORFs, especially in the late time points samples (seen in Figure 2B and S8), meaning the distribution of reads within ORFs is unlikely to represent true biology but rather a technical artifact. As a result, comparison of footprint densities between different regions of the same ORF should be done with great caution. This may have implications for several analyses in the paper, including the frame-shift efficiency calculations of ORF1ab, and translation from ORF1b body TRS sgRNAs and from heteroclite sgRNAs.

Furthermore, detection of translation from these non-canonical sgRNA as presented in the manuscript seems less likely, as it would require extremely high translation-efficiency to be detected above translation from the gRNA.

General comments:

– The figures are not mentioned in the text in order (e.g. Figure 9 may be mentioned before Figure 7). Since there are many figures it is confusing to the reader, and changing the order of the figures or the text will help clarity. In addition, text links to the figure is missing in several places throughout the manuscript (for example, line 312 Figure 5B, line 380 RNA read length distribution).

– Whenever conservation data regarding a specific nucleotide is used to claim it is conserved, it will be useful to present background conservation to demonstrate whether the position is conserved more than expected.

– Two viral species are analyzed in the paper, but their similarities and differences could be further discussed in the results or in the introduction.

– In several cases, comparisons between data presented on two separate graphs are performed. In order to allow the reader to properly compare, for example, RNA levels of viral transcripts at different time points, it will helpful to present them together on the same plot.

---

## [Author Response]

Essential revisions:Overall, the study is well designed, carefully executed and is accompanied by rigorous and exceptionally meticulous data analysis. The data generated in this work may be potentially used beyond the scope of this study and are of importance for a broad audience of virologists and those interested in gene expression. One potential weaknesses of the paper is that it is hard to get through simply due to its length and a lot of data, which makes it difficult to follow, in particular for the non-specialists. Concerning potential technical issues, the ribosome run-offs in the libraries prepared without CHX may affect the distribution of reads within ORFs, which would require some caution with the data interpretation.Specific points:1. The libraries obtained in the absence of CHX clearly show ribosome run-off, as seen in the metagene profiles (Figure S1) and in the translation profiles of most viral ORFs, especially at the late time points (Figure 2B and S8). This raises the question of whether the reported distribution of reads within ORFs is biased and calls for caution when comparing footprint densities between different regions of the same ORF. This may have implications for several analyses in the paper, including the frameshift efficiency calculations of ORF1ab, and translation from ORF1b body TRS sgRNAs and from heteroclite sgRNAs. This issue should be clarified and the conclusions toned down where appropriate.

For context, we start by clarifying our decision to omit a CHX pre-treatment step in the preparation of the majority of the libraries in this manuscript:

Early profiling papers on *Saccharomyces cerevisiae* and *Schizosaccharomyces pombe* found that pre-treatment with CHX exacerbated a “ramp” in the 5′ ~200 codons of CDSs in the host meta-gene profile (Gerashchenko and Gladyshev 2014; Ingolia et al., 2009), although the ramp was still present to some extent in the absence of CHX pre-treatment (Ingolia et al., 2009). In mammalian cells, the ramp was only detectable after CHX pre-treatment, leading to the conclusion that it is not a physiological feature of translation in mammalian cells (Ingolia, Lareau, and Weissman 2011). Although this conclusion has been challenged by recent work (Sharma et al., 2021), the general best practice in harvesting mammalian cells for ribosome profiling is still to avoid CHX pre-treatment where possible, to avoid such potential artefacts (Ingolia et al., 2012, 2011; Sharma et al., 2021). CHX-induced artefacts are a particular issue when observing stressed cells. In yeast cells harvested with CHX pre-treatment, the 5′ ramp becomes greatly enhanced under stress conditions compared to non-stressed conditions, whereas in yeast cells harvested without CHX pre-treatment, the ramp is unchanged between these conditions (Gerashchenko and Gladyshev 2014). This led to the conclusion that the increase in the ramp under stress was an artefact of CHX pre-treatment and further spoke against its use in harvesting protocols. Unpublished mammalian cell data from our labs and the lab of Dr Nerea Irigoyen (see page 32 of G. M. Cook’s PhD thesis, Figure 1.12, https://doi.org/10.17863/CAM.77982) indicates that the same is true for the murine cell line 17 Cl-1. For the libraries described in the present work, it was particularly pertinent to consider artefacts that might be enhanced under stress conditions, as viral infection exerts considerable stress on the cell. A CHX-induced artefact that affects RPF distribution in stressed (e.g. late-timepoint infected) cells more than unstressed (e.g. mock or early-timepoint infected) cells could confound interpretation of the results. Therefore, we opted to omit the CHX pre-treatment step, judging that any bias introduced by a small amount of ribosome run-off (which should be comparable across all libraries) would be more favourable than the potential bias introduced by an artefactual ramp specifically enhanced in stressed cells. To minimise ribosome run-off, we harvested these cells as rapidly as was feasible, by snap-freezing in liquid nitrogen.

Regarding the extent to which the run-off in our libraries has the potential to affect our conclusions, we believe this to be minimal. Reviewer three is correct in pointing out that there is some run-off observable in Figure 1—figure supplement 1 (formerly Supplementary Figure 1); however, its magnitude is relatively small, and it only noticeably affects the first ~100 nt of host CDSs. It occurred to us while drafting this response that it may not have been clear that the x axis on this host meta-gene profile denotes nucleotides as opposed to codons, so this has been clarified on this and other similar plots. The fact that the visible run-off in our libraries covers such a short region makes it unlikely that its effect is observable in viral ORFs at the scale depicted in Figure 2 and Figure 2—figure supplement 2 (formerly Supplementary Figure 8), as both of these Figures show the full viral genome, which is ~15,000 nt long. Figure 2B displays RNASeq data, to which ribosome run-off is not relevant (and the increasing density towards the 3′ end represents sgRNA transcription); however, we presume the reviewer meant to cite Figure 2C here. Although it is probable that the first ~100 nt of viral CDSs are affected by run-off in a similar way to the host CDSs in the meta-gene profile, we do not believe there is evidence that run-off affects the ribosome profile of viral ORFs beyond this point. Taking ORF1ab as an example (more clearly seen in Figure 10A and Figure 10—figure supplement 1 [formerly Supplementary Figure 17], which show only this ~12,000-nt long ORF), the reviewer may have interpreted the increasing RPF density shortly after the nsp2 frameshift site (originally discussed in lines 700-705 but since relocated to lines 761-769) as ribosome run-off due to the similarity in shape. This could also lead to their conclusion that run-off is more pronounced on viral ORFs at later timepoints, as the density in this region is relatively stable at 6 hpi. However, this region in which density increases at late timepoints begins ~4,000 nt downstream of the ORF1ab start codon, a distance 40× greater than the ~100-nt distance over which run-off is observed in host CDSs, making this an unlikely cause for the observed profile. Furthermore, run-off being more pronounced at later timepoints in viral ORFs would be surprising, as it would be inconsistent with the observed patterns in host CDSs, in which there is no clear trend over time (Figure 1—figure supplement 1 [formerly Supplementary Figure 1]).

We agree with the reviewer that caution is required when comparing RPF densities in different regions of an ORF on which ribosome run-off is evident. However, this is only likely to have a significant effect when one of the regions under comparison is close enough to the initiation codon to be noticeably affected by run-off. This is not the case for any of the regions of ORF1ab that we used to calculate frameshift efficiency or putative translation from ORF1b sgRNAs – the shortest distance between the ORF1ab initiation codon and the beginning of any of these regions was 1,551 nt (all region coordinates given in Supplementary file 2 [formerly Supplementary Table 1]; ORF1ab initiation codon begins at genomic coordinate 191). This is well beyond the ~100 nt region over which run-off has a pronounced effect, and therefore we do not expect ribosome run-off to have a significant impact on our conclusions in these sections. For our analysis of putative heteroclite sgRNA translation (Figure 9D), the “heteroclite” region used in our calculations began at the very beginning of ORF1ab, and therefore does include the first ~100 nt and may be affected by ribosome run-off. In contrast, the downstream region (“before FS” in the original manuscript, “gRNA” in the revised submission) is too far downstream of the ORF1ab initiation codon for run-off to have an observable effect. Therefore, for the heteroclite sgRNA translation analysis, run-off could potentially have a small impact on our results (although we expect this to be minor, as the 100 nt run-off region represents only a small proportion of the 1,550-nt “heteroclite” region). However, any effect that this would have would be to artefactually reduce RPF density in the heteroclite region relative to the gRNA region. If this is indeed occurring, it would mean our analysis under-estimates heteroclite sgRNA translation. Taking run-off into consideration therefore strengthens our conclusion that our data support translation of this non-canonical heteroclite sgRNA, as we still see evidence of its translation in spite of this potential bias towards its under-estimation. However, we do not feel it is necessary to modify the manuscript to clarify this somewhat nuanced point, particularly as the reviews will be made publicly available for any specialist readers who are interested.

2. The non-canonical sgRNA would require extremely high translation efficiency to be detected above translation from the gRNA. This is surprising and the authors should provide more explanations to clarify this point.

Although we appreciate that the translation of the in-frame regions of ORF1ab is slightly less obvious at first glance, the novel ORFs overlapping ORF1b provide compelling evidence that the ORF1b sgRNAs are indeed translated with high enough efficiency to be observable against the gRNA background. These ORFs (for example nsp12-iORF) are highly translated at late timepoints, reaching levels comparable to that of ORF1b itself, and this is very clear in the Figures (e.g. Figure 6B) as the RPF density is in an overlapping phase. It is extremely unlikely that these ORFs are expressed from the canonical PRRSV transcripts (from which the only potential candidate would be gRNA), as this would mean their 5′ UTRs were at least 8,000 nt long and contained over 100 AUGs. By far the most parsimonious explanation is that these novel ORFs are expressed from the ORF1b sgRNAs, indicating that these transcripts are translated efficiently enough to be observable in our analyses against the background of canonical gRNA translation.

As for how this is the case – although the gRNA transcript is more abundant than the non-canonical heteroclite/ORF1b sgRNAs, an important factor to consider is that gRNA contains signals (for example packaging and/or replication signals) which likely cause a large proportion of the gRNA transcript pool to be sequestered and unavailable for translation. The translation efficiency (TE) calculations in our study are not able to distinguish these sequestered transcripts from the transcripts available for translation, and gRNA has very low TE relative to the other viral transcripts in our analysis (as seen in Figure 8—figure supplement 3 [formerly Supplementary Figure 15] and mentioned in the text in lines 552-555). This is consistent with observations for coronaviruses in other studies (Finkel et al., 2021; Irigoyen et al., 2016; Kim et al., 2021). The non-canonical transcripts, however, likely do not contain all the same packaging/replication signals as the genomic RNA – for example, there is no evidence that the ORF1b sgRNAs are packaged into virions – so they likely spend a much higher proportion of time available for translation. Although heteroclite sgRNAs do retain at least the essential signals required to facilitate packaging into the virion (Yuan et al., 2004; Yuan, Murtaugh, and Faaberg 2000), it is possible/likely that sequestration and packaging of these heteroclite transcripts occurs at lower efficiency than for the gRNA. Further, the heteroclite sgRNAs and the non-canonical ORF1b sgRNAs may be less frequently subjected to replication than gRNA due to differences in the presence/absence of replication signals. Differences, such as these, in the proportion of the transcript pool that is available for translation likely give rise to higher TE values for these non-canonical transcripts.

This is a concept we tried to highlight in our original manuscript (lines 399-402); however, we agree with this reviewer that we did not explain this clearly enough. We have moved the explanation of this concept to lines 644-649, so it is grouped together with the main discussion of ORF1b and heteroclite sgRNA translation and appears after we have mentioned the low TE of gRNA. We have also expanded the text to explain some of the above points.

Further, in line with this reviewer’s comment, and also taking into account suggestions from the other reviewers to streamline the manuscript, we have edited Figure 9D to try and make its meaning clearer. We removed the “after FS” region (instead moving the discussion of density variations in this region to the frameshifting section, where they are more relevant). We renamed the “before FS” region to “gRNA” to simplify its interpretation, and we applied the decumulation procedure to the RiboSeq read density as well as the RNASeq, to make these results more easily comparable, re-calculating the TE using the decumulated density. Hopefully these changes to the Figure make it clear that the heteroclite sgRNAs do indeed have considerably higher TE than the gRNA, an observation which we have now also added to the main text (lines 696-700).

3. To test the potential mechanism of changing PRF efficiency, can the authors correlate the PRF efficiency at different time points with the expression of the tRNA isoacceptors (or isodecoders) that are required to read the slippery sequence?

This is a very astute comment and, indeed, such knowledge would potentially be informative about the frameshift process and how the process may be regulated during the viral replication cycle. Unfortunately, the RNASeq method employed is not tailored for tRNAs and whilst the datasets do contain tRNA reads, it is certain that they will have biased representation and we cannot easily determine the range of isoacceptors present. In future work, this might be addressed using tRNA profiling methods (Behrens, Rodschinka and Nedialkova, 2021), but this is beyond the scope of the present work.

4. The paper is written in a dense technical language that limits its readability and is likely to reduce the impact of the study. As not all findings are equally important, the authors should revise the manuscript for clarity, preferably by shortening the main text and leaving less important observations in the Supplements.

We have made changes throughout the body of the manuscript to reduce technical language (allowing the reader to find this in the Materials and methods if they are interested) and streamlining the main text. We also added a schematic of a junction-spanning read to Figure 4 (new panel A) to help clarify the terms “donor” and “acceptor” and to provide a visual aid to non-specialist readers. We have added several new sections to the Results (e.g. “Examining the potential for non-canonical transcripts to modulate non-structural protein stoichiometry”) which break up longer sections into smaller parts and hopefully makes the salient points of each section clearer. Further, we hope many of the specific points in the list below will also improve readability for both a specialist and non-specialist audience.

In addition, the following specific points should be addressed:a. Figure 12, which should summarize the findings of the paper, is very difficult to understand. It should be revised for clarity and explained in Figure legend. All elements of the scheme should be clearly labeled.

We have revised this Figure considerably, adding/expanding labels for all elements of the Figure, and allocating each finding its own panel. Each panel has been given a legend highlighting the key finding, and these have been rearranged so that the order of the panels follows the order of the narrative in the Discussion.

b. All Figure legends should be shortened by moving non-essential parts into Methods.

We have shortened the Figure legends wherever possible, introducing new headings in the Materials and methods section (“Plots of read distributions on viral genomes”; “Scaling ORF1b sgRNA transcript abundance for presentation alongside RiboSeq density plots”) to allow us to relocate this information and make it easy to find.

c. The Discussion should be better structured, e.g. by introducing subheadings.

We have introduced subheadings for each section of the Discussion and stated the key finding of the section in the subheading. As mentioned in (a), we have also reorganised Figure 12 so that the sections of the Discussion correspond to the panels in the Figure, and progress through these in order. We think these changes have significantly improved the readability of the Discussion and will help make it clear to readers which findings are the most important – thank you for these suggestions.

d. The figures are not mentioned in the text in order (e.g. Figure 9 may be mentioned before Figure 7). Since there are many figures, this is confusing, and changing the order of the figures or the text will help clarity. In addition, text links to the figure are missing in several places throughout the manuscript (for example, line 312 Figure 5B, line 380 RNA read length distribution).

Supplementary Table 2 was mentioned in the text before Supplementary Table 1, so the numbers of these two tables have been changed accordingly, as well as changing their names from Supplementary Table to Supplementary file, in line with *eLife*’s publishing conventions. Other than this, the Figures are mentioned in the text in the correct order. As we regularly refer back to Figures previously mentioned, we have now edited the text in some places to make it clear that we are referring back to a Figure that has already been discussed, as opposed to mentioning a new Figure (lines 227 and 631).

We have gone through the text and added Figure citations for each point, where they were absent or unclear. We also added a new panel (C) to Figure 5, with the N-long and N-short body TRS logo plots reproduced from Figure 7, so that the reader can inspect these when their conservation is discussed in lines 343-348 instead of waiting until Figure 7 is cited.

e. Whenever conservation data regarding a specific nucleotide is used to claim it is conserved, it will be useful to present background conservation to demonstrate whether the position is conserved more than expected.

For the uORF initiation codon, it is somewhat difficult to obtain a fair measure of background conservation because this is complicated by the presence of multiple predicted RNA secondary structures in the PRRSV 5′ UTR (Gao et al., 2013), which are expected to be subject to selection pressure. The A and U of the uORF AUG are in the loop of a predicted stem loop, so the effect of selection pressure to maintain the secondary structure is expected to be reduced as there are no base pairs to maintain, and the conservation of these residues may represent selection pressure to maintain the initiation codon. To avoid adding further detail/complexity, we did not include the following analysis in the manuscript, but to address this reviewer’s point more systematically, we compared the conservation of the uORF AU residues to that of other unpaired residues. We made a multiple sequence alignment of the 5′ UTRs of the available NA PRRSV sequences (using MUSCLE with default parameters), after removing sequences with incomplete 5′ UTRs. We used mfold to predict the secondary structure of the SD95-21 5′ UTR, to determine which regions are predicted to be base paired and which are loops/bulges/unpaired regions (see Author response image 1). We mapped these structural predictions to the positions on the multiple sequence alignment (Author response image 1) and calculated the average Shannon entropy for the bases in each unpaired region. The AU of the uORF AUG had lower entropy, indicating greater sequence conservation, than 14 out of the 17 unpaired regions, showing that it is more conserved than comparable regions of the NA PRRSV 5′ UTR.

**Author response image 1. sa2fig1:** Structure and conservation of the NA PRRSV 5′ UTR. (A) The structure of the first 240 nt of the SD95-21 NA PRRSV genome, as predicted by mfold (default parameters). The first base was excluded due to being absent on many NA PRRSV sequences. The uORF AUG and the leader TRS are highlighted in blue. (B) Multiple sequence alignment of the NA PRRSV 5′ UTR (excluding the first base from all sequences and excluding sequences with incomplete UTRs). Regions which are predicted to be unpaired in panel A are indicated by boxes, with the mean Shannon entropy for each of these regions indicated above (as calculated by entropy-one; https://www.hiv.lanl.gov/content/sequence/ENTROPY/entropy_one.html). The uORF AUG is indicated by a purple dashed box, and the average entropy for the A and U were calculated separately from the three upstream bases.

For the body TRSs, we could potentially assess whether the third positions within degenerate codons are conserved as a measure of whether the conservation is greater than expected. However, the body TRSs are in different frames relative to ORF1b, and are only six nucleotides long, so may not all contain highly degenerate codons. This would make this a rather detailed and complicated analysis to include in the manuscript, given it is already very long. Although we do mention the nucleotide conservation of the ORF1b sgRNA body TRSs (particularly the final two Cs; lines 664-665), the more compelling evidence for the conservation of these transcripts is that body TRSs in similar (or sometimes identical) genomic locations are used in isolates from diverse areas of the phylogeny, including different species (lines 667-684). We do not think that adding an assessment of background nucleotide conservation to the manuscript would be worth the increase in length, given the detail it would require to perform this accurately.

f. Two viral species are analyzed in the paper, but their similarities and differences could be further discussed in the results or in the introduction.

We have added some information about this to the Introduction, starting at line 57.

g. In several cases, comparisons between data presented on two separate graphs are performed. In order to allow the reader to properly compare, for example, RNA levels of viral transcripts at different time points, it would be helpful to present them together on the same plot.

Viral replication leads to a great increase in the number of viral reads between 6 and 9 hpi. This means that comparisons of the absolute transcript or RiboSeq read density levels between timepoints are not always the most informative or interesting interpretations of the data, as we expect these to increase due to replication. We do show these large-scale changes in plots where all timepoints are displayed together (for example Figure 2D and E, Figure 8); however, the main focus of our analyses is often on observing the trends between transcripts/ORFs within each timepoint and comparing how these trends change over time. For example, in Figure 9C, the key result is that the overall ribosome distribution in ORF1b is approximately uniform at 6 hpi but becomes more skewed towards higher density at the 3′ end at late timepoints. These changes in the shape of the data are more clearly seen when plotting each timepoint on a separate panel, and the 6 hpi results would be difficult to discern if the bars were rearranged to be plotted without separation by timepoint.

Taking the reviewer’s example of the RNA transcript levels: As mentioned above, we show the different timepoints on the same plot in Figure 8, where the overall trend of increasing read density over time can be seen. In the bar chart in Figure 8—figure supplement 1 (formerly Supplementary Figure 13), we expand on this by including the non-canonical transcripts and separating the results into timepoints. Many of the key points of this analysis (such as showing there is a skew towards gRNA transcription at 6 hpi and then sgRNA transcription at later timepoints, or that the abundance of sgRNAs relative to each other is similar at 9 hpi and 12 hpi), are observations of within-timepoint trends, which are better highlighted when timepoints are separated (see paragraph starting at line 534 for discussion of this Figure). To aid comparisons between timepoints, in Figure 8—figure supplements 1-3 (formerly Supplementary Figures 13-15) the sub-plots are plotted at the same scale if the data is suitable for comparison (noting that, for analyses involving junction-spanning read counts, libraries must have similar RNA insert lengths to be unbiasedly comparable).

h. Multiple sequence alignment in Figure 6E are not very useful, as the color code for amino acids is not assigned; at the minimum, clarify the color code. In Figure 6D, the nucleotide sequence is somewhat sandwiched and is hard to read. The issue is easily solvable by giving it more space on top and the bottom. Sequence logos in Figures6C, F, G and Figure 7 contain text (on axes) that is of unintelligible size. If it cannot be enlarged, it could be simply removed since the logos are easily interpretable without it.

Thank you for these suggestions, we have incorporated all of them into the relevant Figures and increased the space above and below the nucleotide sequence in Figure 10—figure supplement 7 (formerly Supplementary Figure 23) as well as Figure 6D.

i. Please add the canonical body TRS sequence for comparison in Figure 7.

The conservation at the genomic locations corresponding to each of the canonical body TRSs is given to the right of the transcripts in Figure 7, and we have now made a combined logo plot of all these consensuses for each species, which we have added to the plot next to the leader TRS.

j. The western blot in Figure 3F is missing a loading control.

We have repeated the Western blot with a loading control.

k. Figure S23B, the green 6 hpi lines seem to be colored blue.

The 6 hpi libraries have very low read counts at this position compared to the libraries from later timepoints, so the green line is difficult to see in this Figure (now named Figure 10—figure supplement 7B). We have changed the order in which the lines are plotted, so that 6 hpi is plotted last to increase visibility and have slightly decreased the lower y axis limit so the 6 hpi line is no longer obscured by the line separating the WT plot from the KO2 plot.

l. In Figure S23B and C, the read length distribution in the specific position should be compared to the overall library read length distribution.

The host distributions have been added to Figure 10—figure supplement 7C (formerly Supplementary Figure 23C) as dashed grey lines; however, we did not add them to Figure 10—figure supplement 7B (formerly Supplementary Figure 23B) as it would be difficult to discern them amongst the other lines on this plot, and they are in Figure 1C for readers who wish to compare length distributions for libraries not in Figure 10—figure supplement 7C.

m. Which infected time points are analyzed in Figure 11? Showing a time course of the genes of interest may be useful.

Figure 11 shows an analysis of differential gene expression at 12 hpi (lines 893, 898, 921, and 1008). We did not include differential gene expression analyses of other timepoints, but we have included in this response a plot of TXNIP expression over the timecourse covered by our libraries (Author response image 2), which shows that TXNIP is slightly up-regulated in the infected libraries at 9 hpi, a trend which continues in the KO2 libraries but is sharply curbed in the WT libraries. The translation efficiency of TXNIP in the infected samples starts to decrease at 9 hpi, and continues to decrease in the WT libraries but recovers in the KO2 libraries. These profiles are consistent with a model in which TXNIP expression is inhibited by the WT virus but not the KO2 virus.

**Author response image 2. sa2fig2:** TXNIP expression over a timecourse of NA PRRSV infection. Bowtie-aligned positive-sense reads mapping to TXNIP (XM_007977281.1) were filtered to select those with inferred P sites within the CDS (filter applied to RNASeq reads for comparability). Read counts were converted to RPM using a denominator of positive-sense host-mRNA-mapping reads, and TE was calculated by dividing RiboSeq values by RNASeq values.

n. From the analysis of RNA and TE in Figure 11, it is concluded that translation of transcriptionally upregulated genes does not go up. The footprint data can be used to directly show the change in translation, and even an increase in translation that is lower than expected from the increase in RNA levels can point to translation inhibition.

We agree with this reviewer that, even if overall translation levels increased, this could still correspond to translation inhibition if they didn’t increase in line with the increase in transcript levels. However, the main focus of this WT vs mock analysis (paragraphs beginning lines 899 and 1008) is on whether transcription or translation control makes the greater contribution to the host response, a question which is addressed more appropriately by analysing TE as opposed to total RPF counts (which would measure the outcome of these two levels of regulation combined). Although it is possible to use DESeq2 to process solely the RPFs, we feel that the manuscript is already rather long and detailed, and the benefit of adding this extra analysis to confirm a point already evident in the existing analyses would be outweighed by the addition of further length and complexity to the manuscript.

o. Line 911, no AUG codons were found in the 5'UTR of these viruses, but translation may initiate from near-cognate start codons.

We have edited the text (lines 1052-1053) to clarify this.

p. I think References 49,117 and 128 do not support the claim that frameshift efficiency changes with time of infection.

To summarise the parts of these papers to which we were referring when saying there is previous work which suggests an increase in ORF1ab PRF efficiency during coronavirus infection:

Reference 49 is Irigoyen *et al.*, 2016 (a previous study from our laboratories). We calculated ORF1ab PRF efficiency for murine coronavirus in 17 Cl-1 cells at 5 hpi (MOI 10), giving values of 48% and 70% for the first and second replicates, respectively (Figure 8B, RiboSeq bars). Despite both replicates having been harvested at the same timepoint, other analyses in the paper indicated that the second replicate had advanced further in infection than the first (clearest in Figure 3B), hence our suggestion in the present work that these results support an increase in PRF efficiency over time.

Reference 117 is Kim *et al.*, 2021. These authors calculated SARS-CoV-2 ORF1ab PRF efficiency at 4, 12, 16, 24, 36 and 48 hpi in Calu-3 cells (MOI 10). They found a great increase in PRF efficiency, from ~20% to ~70%, between 4 and 12 hpi, after which it remained roughly stable (60-80%) until 48 hpi, when it decreased to ~35% (Supplementary Figure 2j).

Reference 128 is Puray-Chavez *et al.*, 2021. Specifically, we reference version 3 of this bioRxiv pre-print (posted on October 7^th^, 2021, and the most recent version at the time of writing). These authors calculated ORF1ab PRF efficiency for SARS-CoV-2 at 6, 12 and 24 hpi in Vero E6 cells at MOI 2 pfu/cell. They found that the average PRF efficiency at 6 hpi was ~45%, whereas it was ~75% at 12 and 24 hpi, however they have considerable variation between their replicates, particularly at 6 hpi (Figure 1F). This may be due to library quality issues in their first replicate (out of three), for example as indicated by the substantial difference in RPF length distribution for host and viral CDSs (Figure S2A), suggesting viral RNP contamination. The variation at 6 hpi means the range of values from this timepoint overlaps those at the later timepoints, so they would not obtain statistical significance for the differences observed (nor do they do claim to in the text of their paper), however, their average efficiency does increase from the early to late timepoints.

In our submitted manuscript, our intention was to ensure we gave due acknowledgement to previous work in this area. We did not intend to claim that any of these papers stated in their text that there was a temporal increase in frameshift efficiency (as none address this directly), nor to give the impression that any one paper had strong enough evidence to support this conclusion without further investigation. For all three papers, assessment of potential RNP contamination within ORF1b and specific statistical assessment of any observed differences between timepoints (in some cases needing additional replicates) would be required to formally evaluate the hypothesis. To try and soften our statement about previous work supporting temporal modulation of coronavirus ORF1ab PRF, we have edited the text in our manuscript (line 1100), changing “previous results suggest ORF1ab frameshift efficiency may increase over time” to “previous results tentatively suggest ORF1ab frameshift efficiency may increase over time”.

References

Behrens, Andrew, Geraldine Rodschinka, and Danny D. Nedialkova. 2021. “High-Resolution Quantitative Profiling of TRNA Abundance and Modification Status in Eukaryotes by Mim-TRNAseq.” *Molecular Cell* 81(8):1802-1815.e7.

Finkel, Yaara, Orel Mizrahi, Aharon Nachshon, Shira Weingarten-Gabbay, David Morgenstern, Yfat Yahalom-Ronen, Hadas Tamir, Hagit Achdout, Dana Stein, Ofir Israeli, Adi Beth-Din, Sharon Melamed, Shay Weiss, Tomer Israely, Nir Paran, Michal Schwartz, and Noam Stern-Ginossar. 2021. “The Coding Capacity of SARS-CoV-2.” *Nature* 589(7840):125–30.

Gao, Fei, Huochun Yao, Jiaqi Lu, Zuzhang Wei, Haihong Zheng, Jinshan Zhuang, Guangzhi Tong, and Shishan Yuan. 2013. “Replacement of the Heterologous 5 0 Untranslated Region Allows Preservation of the Fully Functional Activities of Type 2 Porcine Reproductive and Respiratory Syndrome Virus.” *Virology* 439:1–12.

Gerashchenko, Maxim V and Vadim N. Gladyshev. 2014. “Translation Inhibitors Cause Abnormalities in Ribosome Profiling Experiments.” *Nucleic Acids Research* 42(17):e134.

Ingolia, Nicholas T., Gloria A. Brar, Silvia Rouskin, Anna M. McGeachy, and Jonathan S. Weissman. 2012. “The Ribosome Profiling Strategy for Monitoring Translation in vivo by Deep Sequencing of Ribosome-Protected MRNA Fragments.” *Nature Protocols* 7(8):1534–50.

Ingolia, Nicholas T., Sina Ghaemmaghami, John R. S. Newman, and Jonathan S. Weissman. 2009. “Genome-Wide Analysis in vivo of Translation with Nucleotide Resolution Using Ribosome Profiling.” *Science* 324(5924):218–23.

Ingolia, Nicholas T., Liana F. Lareau, and Jonathan S. Weissman. 2011. “Ribosome Profiling of Mouse Embryonic Stem Cells Reveals the Complexity and Dynamics of Mammalian Proteomes.” *Cell* 147(4):789–802.

Irigoyen, Nerea, Andrew E. Firth, Joshua D. Jones, Betty Y. W. Chung, Stuart G. Siddell, and Ian Brierley. 2016. “High-Resolution Analysis of Coronavirus Gene Expression by RNA Sequencing and Ribosome Profiling” edited by M. B. Frieman. *PLoS Pathogens* 12(2):e1005473.

Kim, Doyeon, Sukjun Kim, Joori Park, Hee Ryung Chang, Jeeyoon Chang, Junhak Ahn, Heedo Park, Junehee Park, Narae Son, Gihyeon Kang, Jeonghun Kim, Kisoon Kim, Man Seong Park, Yoon Ki Kim, and Daehyun Baek. 2021. “A High-Resolution Temporal Atlas of the SARS-CoV-2 Translatome and Transcriptome.” *Nature Communications* 12(1):1–16.

Sharma, Puneet, Jie Wu, Benedikt S. Nilges, and Sebastian A. Leidel. 2021. “Humans and Other Commonly Used Model Organisms Are Resistant to Cycloheximide-Mediated Biases in Ribosome Profiling Experiments.” *Nature Communications* 12(1):1–13.

Yuan, S., M. P. Murtaugh, and K. S. Faaberg. 2000. “Heteroclite Subgenomic RNAs Are Produced in Porcine Reproductive and Respiratory Syndrome Virus Infection.” *Virology* 275(1):158–69.

Yuan, Shishan, Michael P. Murtaugh, Faith A. Schumann, Dan Mickelson, and Kay S. Faaberg. 2004. “Characterization of Heteroclite Subgenomic RNAs Associated with PRRSV Infection.” *Virus Research* 105(1):75–87.

Zhang, Riteng, Peixin Wang, Xin Ma, Yifan Wu, Chen Luo, Li Qiu, Basit Zeshan, Zengqi Yang, Yefei Zhou, and Xinglong Wang. 2021. “Nanopore-Based Direct RNA-Sequencing Reveals a High-Resolution Transcriptional Landscape of Porcine Reproductive and Respiratory Syndrome Virus.” *Viruses* 13(12):2531.